# Sertm2 is a conserved micropeptide that promotes GDNF-mediated motor neuron subtype specification

Fang-Yu Hsu [ID] [1,2], Ya-Ping Yen [ID] [1], Hung-Chi Fan [ID] [1], Mien Chang[1] & Jun-An Chen [ID] [1,2,3 ✉]

## Abstract

Small open-reading frame-encoded micropeptides within long noncoding RNAs (lncRNAs) are often overlooked due to their small size and low abundance. However, emerging evidence links these micropeptides to various biological pathways, though their roles in neural development and neurodegeneration remain unclear. Here, we investigate the function of murine micropeptide Sertm2, encoded by the lncRNA *A730046J19Rik*, during spinal motor neuron (MN) development. Sertm2 is predicted to be a conserved transmembrane protein found in both mouse and human, with subcellular analysis revealing that it is enriched in the cytoplasm and neurites. By generating C terminally Flag-tagged Sertm2 and expressing it from the *A730046J19Rik* locus, we demonstrate that the Sertm2 micropeptide localizes in spinal MNs in mice. The GDNF signaling-induced Etv4[+] motor pool is impaired in *Sertm2* knockout mice, which display motor nerve arborization defects that culminate in impaired motor coordination and muscle weakness. Similarly, human *SERTM2* knockout iPSC-derived MNs also display reduced ETV4[+] motor pools, highlighting that Sertm2 is a novel, evolutionarily conserved micropeptide essential for maintaining GDNF-induced MN subtype identity.

**Keywords** Motor Neurons; Long Noncoding RNA; Micropeptide; Etv4; GDNF
**Subject Categories** Neuroscience; RNA Biology

## Introduction

Long noncoding RNAs (lncRNAs) are a class of RNA transcripts that exceed 500 nucleotides in length and have traditionally been considered as non-protein-coding (Mattick et al, 2023). However, recent findings indicate that small open-reading frames (sORFs) embedded within these lncRNAs may be translated into functional small proteins, termed micropeptides (Magny et al, 2013; Matsumoto et al, 2017). These micropeptides have likely been overlooked due to genome annotation conventions that set an arbitrary 300-nucleotide (100-codon) threshold for defining coding potential. Moreover, advances in computational and experimental approaches have revealed that a significantly larger proportion of the genome is actively translated than previously recognized (Makarewich and Olson, 2017; Yeasmin et al, 2018). Thus, sORFs that encode evolutionarily conserved micropeptides have been identified within transcripts originally classified as 'non-coding'. These micropeptides have been proven to exert critical roles in various biological processes, functioning both independently and as regulators of larger proteins (Makarewich, 2020). Emerging research now indicates that micropeptides play essential roles in fundamental biological processes, including myoblast fusion, calcium homeostasis, and cardiomyocyte biology (Hassel et al, 2023). These findings on sORF-encoded polypeptides (SEPs) open up new avenues for research into their regulatory potential in cellular and developmental biology (Saghatelian and Couso, 2015). For example, Myoregulin (MLN), a conserved 46-amino-acid micropeptide, was initially classified as a product of an lncRNA. MLN is expressed across all three types of skeletal muscle and interacts with sarcoplasmic reticulum (SR) $Ca^{2+}$-ATPase (SERCA), a membrane pump critical for regulating SR $Ca^{2+}$ uptake. By inhibiting $Ca^{2+}$ uptake into the SR, MLN complexes promote muscle relaxation (Anderson et al, 2015). Another micropeptide originating from an lncRNA is PAR-amplifying and CtIP-maintaining micropeptide (PACMP), a 44-amino-acid peptide highly enriched in breast tumors. PACMP appears to play a role in cancer progression and drug resistance by modulating the DNA damage response, highlighting its therapeutic potential in cancer treatment (Zhang et al, 2022). While biological roles have been confirmed for only a small number of the micropeptides identified to date, this rapidly expanding field highlights the substantial work that remains to fully understand their existence and functions. In particular, the roles of micropeptides during neural development remain largely unclear. This topic is particularly intriguing given that lncRNAs are predominantly expressed in the mammalian nervous system (Briggs et al, 2015; Chen and Chen, 2020), and many neuropeptides, some of which are small secretory peptides, play a crucial role in neuronal physiology (Liau et al, 2023). A recent study discovered that *MALAT1*, traditionally regarded as a nuclear lncRNA, may be exported into the cytoplasm in differentiating neurons and produce a micropeptide, M1. Synaptic stimulation was shown to enhance M1 expression, indicating that *MALAT1* could function as a cytoplasmic coding RNA that modulates

[1]Institute of Molecular Biology, Academia Sinica, Taipei 11529, Taiwan. [2]Genome and Systems Biology Degree Program, Academia Sinica and National Taiwan University, Taipei 10617, Taiwan. [3]Neuroscience Program of Academia Sinica, Academia Sinica, Taipei, Taiwan. ✉E-mail: jac2210@gate.sinica.edu.tw

synaptic activity, opening up a new avenue for investigating the roles of micropeptides in neural tissues (Xiao et al, 2024).

To further explore this topic, we use spinal motor neurons (MNs) as a model to systematically investigate MN differentiation and specification and to test if any MN-derived lncRNAs might have the potential to produce functional micropeptides and thereby act in the MN generation process. Motor neuron development is driven by a well-defined set of spatiotemporal transcription factors (TFs). Specifically, MN differentiation is largely regulated by extrinsic signals that control intrinsic genetic programs, which define and stabilize MN subtype identities, particularly those within MN columns and pools. This organization is most evident at the limb level, where MNs that supply limb muscles located in the lateral motor column (LMC) are subdivided into molecularly distinct motor pools, each innervating specific muscle groups (Chen and Chen, 2019; Dasen and Jessell, 2009). The diversification of MN subtypes is primarily governed by the Hox family of homeodomain-containing TFs (Miller and Dasen, 2024). These factors act in a combinatorial manner, regulating the rostrocaudal expression of Foxp1, a key determinant of LMC MNs in the brachial and lumbar spinal cord (Dasen et al, 2008).

Within the LMC, distinct MN pools are defined by specific TFs, such as the ETS transcription factors Etv4 (Pea3) and Etv1 (Er81) (Arber et al, 2000; Catela et al, 2016; Ladle and Frank, 2002), the runt-related protein Runx1, and the Pou-domain transcription factor Pou3f1 (Scip) (Dasen et al, 2005; Helmbacher et al, 2003). The role of Hox TFs in fine-tuning MN pool differentiation has been studied extensively using selective MN pool markers. Functional experiments altering Hox gene expression in MN pools have revealed corresponding changes in MN pool markers, with the outcomes closely linked to modifications in motor axon trajectories toward their muscle targets (Stifani, 2014). Etv4 and Etv1 are expressed in specific MN pools in the vertebrate spinal cord, and limb ablation in early chick embryos eliminates their expression (Lin et al, 1998). In this context, glial cell line-derived neurotrophic factor (GDNF) has been identified as one of the major limb-producing signals that induces the Etv4⁺ motor pool subsets in the LMC (Haase et al, 2002; Lin et al, 1998; Livet et al, 2002). Striking alterations in MN positioning, muscle target invasion, dendritic development, and sensory-motor connectivity have been observed in Etv4 mutant mice (Vrieseling and Arber, 2006). These findings underscore the significant role of both early intrinsic transcriptional programs and target-derived signals in MN differentiation. However, the mechanistic link between how target-dependent GDNF signaling integrates with the early-established Hox transcriptional networks in MNs remains to be elucidated. Furthermore, how retrograde GDNF signaling impacts specific Etv4-mediated molecular targets to ensure sensory-motor connectivity also needs to be scrutinized.

In this study, we have identified a highly MN-enriched lncRNA A730046J19Rik (4.7 kb) that contains a conserved sORF of 270 base pairs encoding an 89-amino-acid micropeptide, Sertm2 (serine-rich and transmembrane domain containing 2). Strikingly, we demonstrate that Sertm2 is strongly expressed in the Etv4⁺ motor pools. Furthermore, we have discovered that mouse Sertm2 may modulate peripheral GDNF signaling. Mice lacking A730046J19Rik containing Sertm2 exhibit defects in axonal arborization, particularly in the Etv4⁺ motor pools targeting the cutaneous maximus and anterior latissimus dorsi muscles, resulting in impaired walking coordination and motor function. Moreover, loss of the Etv4⁺ motor pools caused by A730046J19Rik deletion could be rescued by administering the micropeptide Sertm2. In line with these observations, depletion of human SERTM2 significantly impairs ETV4 expression, underscoring the critical role of Sertm2 in MN development in both mice and humans. Our study identifies a functional, conserved lncRNA-derived micropeptide in spinal MNs, establishing a foundation for future research into the roles of micropeptides in neural development and neurodegeneration.

## Results

### Identification of a motor neuron-enriched lncRNA, A730046J19Rik

In a previous effort to uncover novel lncRNAs potentially critical for MN development, we differentiated Mnx1::GFP embryonic stem cells (ESCs) into spinal MNs or non-MNs according to established protocols (Fig. 1A). We compared MN progenitors (pMNs, Day 4) with nascent differentiated MNs (Mnx1::GFPᵒⁿ cells) sorted from embryoid bodies, as well as dissociated MNs displaying neurite outgrowth at Day 7. Among the identified lncRNA candidates, A730046J19Rik was prominent as not only does it exhibit MN-specific expression (Fig. 1B,C) (Yen et al, 2018), but it is also highly conserved across mammals (Fig. EV1A). In line with our finding, a previous study also found that A730046J19Rik is enriched in differentiated MNs (Biscarini et al, 2018). Subsequent quantitative polymerase chain reaction (qPCR) analysis confirmed A730046J19Rik expression in spinal MNs, with peak levels observed between postnatal day 1 (P1) and P7 (Fig. 1D). Next, we performed in situ hybridization (ISH) on embryonic spinal cords from embryonic day 9.5 (E9.5) to P14 and found that A730046J19Rik primarily localizes at the ventral marginal zone of the spinal cord (Fig. 1E). In E13.5 brachial spinal cord sections, A730046J19Rik appeared to be enriched in specific MN subtypes, particularly in Foxp1⁺ LMC MNs, which innervate limb muscles, and in Lhx3⁺ medial motor column (MMC) MNs that innervate epaxial muscles (Fig. 1F).

To gain a finer resolution of the MN subtypes associated with A730046J19Rik during mouse spinal cord development, we analyzed its expression using our previously generated single-cell RNA-sequencing (scRNA-seq) dataset on E13.5 Mnx1::GFP mouse spinal MNs (Liau et al, 2023). Based on molecular signatures of known columnar markers in brachial MNs, we observed that A730046J19Rik is differentially expressed in both MMC MNs and LMC MNs (Figs. 1G and EV1B). Notably, A730046J19Rik was expressed in three major MMC MN subtypes (Nr2f2, Satb2, and Bcl11b; Fig. EV1B), but it was highly co-expressed with Etv4 in Etv4⁺ (Pea3⁺) motor pools and displayed clear segregation from Pou3f1⁺ (Scip⁺) populations in the LMC MN subtypes (Figs. 1G,H and EV1D). This expression pattern was further validated using RNAscope-based in situ hybridization, whereby A730046J19Rik was examined alongside the MMC-specific markers Satb2 and Nr2f2 (Fig. EV1C), as well as the LMC markers Etv4 and Pou3f1 (Fig. 1I,J). Co-localization of A730046J19Rik and Etv4 expression, with clear segregation from that of Pou3f1, in the brachial region implies a potential role for A730046J19Rik in modulating Etv4-associated motor functions. Collectively, these findings indicate that A730046J19Rik is exclusively expressed in postmitotic MNs and may

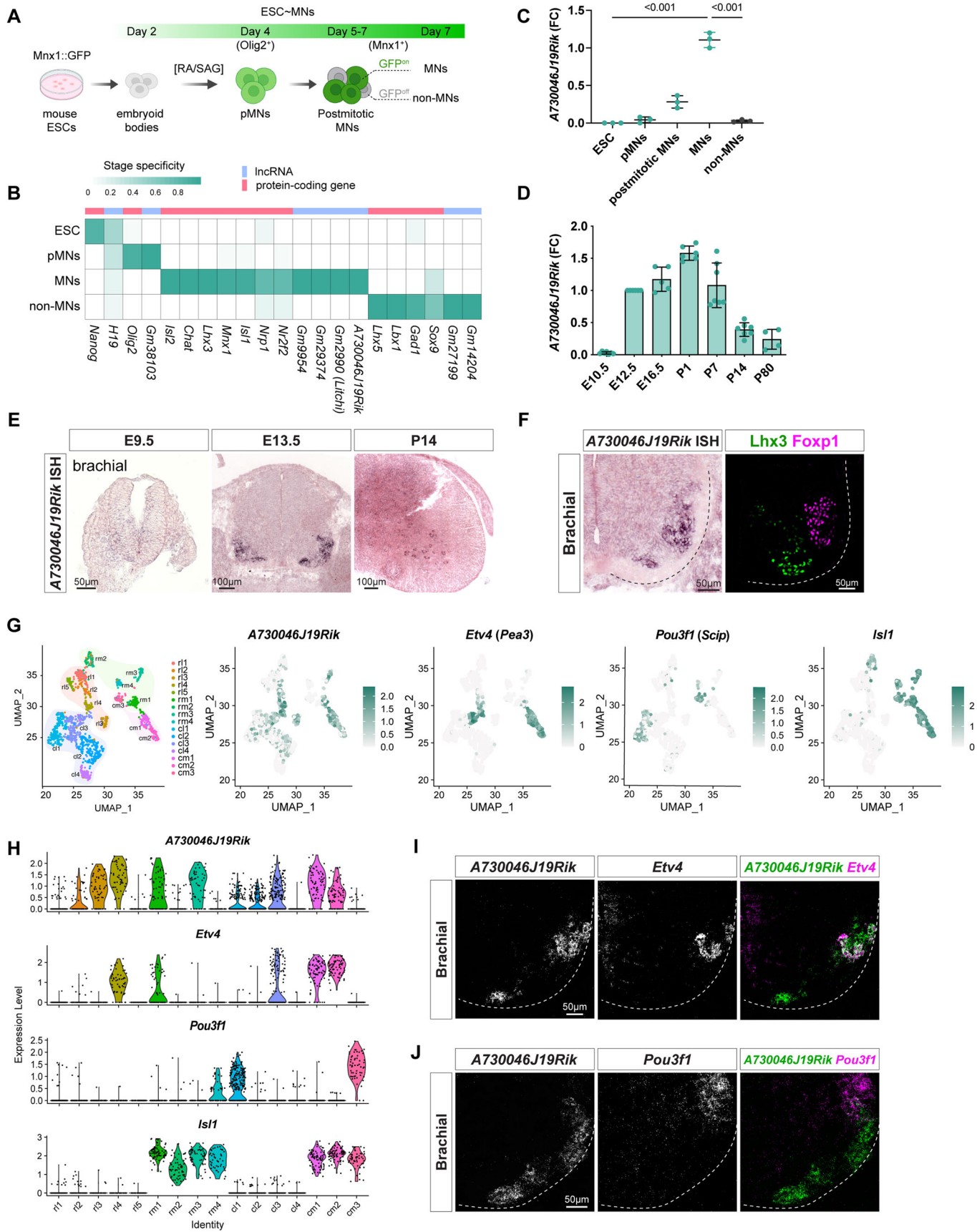

**Figure 1. Spatiotemporal dynamics of *A730046J19Rik* expression during MN development.**

(A) Illustration showing the timeline of motor neurons (MNs) derived from Mnx1::GFP mouse embryonic stem cells (ESCs). RA retinoic acid, SAG smoothened agonist, pMNs motor neuron progenitors, MNs motor neurons. (B) Heatmap showing the gene expression profiling of protein-coding genes [pink] and lncRNAs [blue] during mouse ESC-derived MN differentiation. The green color gradient indicates the stage specificity score across stages from ESCs to MNs or non-MNs. The strand-specific RNA-seq data analyzed in this figure was sourced from Yen et al, 2018. (C) qPCR analyses showing the expression of *A730046J19Rik* during mouse ESC-derived MN differentiation. Data are presented as fold change (FC) relative to MNs at Day 7 with values representing the mean ± SD from $n = 3$ independent experiments and analyzed using ordinary one-way ANOVA. $P < 0.001$ ($P = 2.75 \times 10^{-9}$ for MNs vs. ESC; $P = 4.04 \times 10^{-9}$ for MNs vs. pMNs; $P = 4.8 \times 10^{-8}$ for MNs vs. postmitotic MNs; $P = 3.52 \times 10^{-9}$ for MNs vs. non-MNs). (D) qPCR analyses showing the expression of *A730046J19Rik* in B6 mouse spinal cord at various developmental stages. Data are presented as fold change (FC) relative to the expression at E12.5, with values representing the mean ± SD from $n = 4$–7 independent spinal cord samples, as indicated in the figure. (E) In situ hybridization (ISH) shows that *A730046J19Rik* expression is gradually enriched and restricted to the ventral horn of the developing spinal cord. Scale bars: 50 µm (E9.5), 100 µm (E13.5 and P14). (F) A high-magnification image of *A730046J19Rik* ISH and adjacent immunostaining sections from E13.5 brachial spinal cord demonstrating that *A730046J19Rik* expression is enriched in columnar MN subtypes, as revealed by Foxp1 and Lhx3 staining. Dashed lines outline the spinal cord boundary. Scale bars, 50 µm. (G) (Left) Uniform Manifold Approximation and Projection (UMAP) plot of 16 LMC subclusters from single-cell RNA sequencing (scRNA-seq) of Mnx1::GFP labeled-MNs in E13.5 brachial spinal cord (Rostral C4-T3 segments). The four shaded colors represent distinct subtypes within limb MNs, including rLMCl (rostral lateral, orange), rLMCm (rostral medial, green), cLMCl (caudal lateral, blue), and cLMCm (caudal medial, pink) regions. (Right) UMAP plots displaying the expression patterns of *A730046J19Rik*, *Etv4* (*Pea3*), *Pou3f1*(*Scip*), and *Isl1* in LMC MN subtypes. The green gradient represents the expression levels of the genes of interest. The dataset is derived from Liau et al, 2023. (H) Violin plots reflecting the expression of selected genes (*A730046J19Rik*, *Etv4*, *Pou3f1*, and *Isl1*) within LMC MN clusters. The x-axis represents 16 LMC subclusters, categorized as follows: rl (rostral lateral), rm (rostral medial), cl (caudal lateral), and cm (caudal medial). (I, J) RNAscope-based ISH shows the differential subtype distributions of *A730046J19Rik* transcripts among brachial LMC neurons expressing *Etv4* (I) or *Pou3f1* (J) at transcript level at E13.5. Dashed lines outline the spinal cord boundary. Scale bars, 50 µm.

play a critical role in regulating MN subtype specification, particularly in *Etv4*+ motor pools.

## *A730046J19Rik* encodes a conserved micropeptide, Sertm2

Although most lncRNAs exhibit low conservation relative to protein-coding genes, accumulating evidence has demonstrated that certain lncRNAs contain conserved sORFs across species (Galindo et al, 2007; Patraquim et al, 2022; Pueyo et al, 2016; Ulitsky, 2016). Notably, the human ortholog of murine *A730046J19Rik*, i.e., *LINC00890*, has been re-annotated as encoding the micropeptide SERTM2 (serine-rich and transmembrane domain containing 2) based on peptidomics data (Fig. EV1E, LncPep) (Liu et al, 2022). To determine if murine *A730046J19Rik* also has the potential to produce a micropeptide, we examined its evolutionary conservation. Using the UCSC Genome Browser, we conducted sequence alignments across multiple species. Homologs of *A730046J19Rik* were identified in several mammals, with a specific region demonstrating significant conservation among vertebrates, though a respective homolog is absent from fish, as revealed by a 100-way Multiz vertebrate alignment using the human genome as the reference. Notably, one highly conserved region within *A730046J19Rik* spans a 270-bp sORF, potentially encoding an 89-amino-acid micropeptide, annotated as Sertm2 (Fig. 2A). Further alignment of the predicted protein sequence across species highlighted a conserved transmembrane domain (Fig. 2B).

As most lncRNAs tend to be enriched in the nucleus, we reasoned that if *A730046J19Rik* displays coding potential, it should have a cytoplasmic distribution. To explore the subcellular distribution of *A730046J19Rik* in murine MNs, we harvested ESC-derived MNs on Day 7 and performed single-molecule RNA fluorescence in situ hybridization (smFISH). Quantifications revealed that *A730046J19Rik* is more enriched in the cytoplasm than nuclei (Fig. 2C,D). Additionally, computational tools, including PhyloCSF and the Coding Potential Assessment Tool (CPAT), predicted high coding potential for both mouse *A730046J19Rik* and

human *SERTM2* (Figs. EV1F and EV2A) (Lin et al, 2011; Wang et al, 2013). Moreover, a publicly available ribosome profiling database, combined with mRNA abundance coverage from the GWIPS-viz browser, demonstrated significant translational activity within the sORF region of *A730046J19Rik* (Fig. EV2B) (Michel et al, 2014). To experimentally confirm the translation potential of *Sertm2*, we engineered a construct in which ORF was fused to a C-terminal Flag epitope tag. As a negative control, we also mutated the start codon (ATG → ATT) of ORF to prevent translation. Following overexpression of both constructs in HEK293T cells, western blot analysis revealed that *Sertm2* encodes a ~17 kDa protein (Fig. 2E,F). Furthermore, immunostaining with an anti-Flag antibody detected the Flag-tagged ORF in the cytoplasmic region and extracellular surface of the HEK293T cells, consistent with predictions that Sertm2 might be a transmembrane protein (as supported by predictions from Phobius and AlphaFold) (Figs. 2G and EV2C,D) (Jumper et al, 2021). Thus, we re-annotated the original lncRNA *A730046J19Rik* to *Sertm2* given its conserved micropeptide coding potential.

## Sertm2 is generated in mouse spinal cords

These promising in vitro findings prompted us to examine if Sertm2 is produced endogenously in the mouse spinal cord. Initially, we raised polyclonal antibodies in both rabbit and guinea pig that target two distinct regions of the Sertm2 protein (Fig. EV2E). However, western blot and immunostaining assays revealed that both antibodies demonstrated nonspecific. This result was not unexpected, as previous studies have encountered similar challenges in developing antibodies for small proteins of low abundance and limited epitope availability for custom antibody generation (Hassel et al, 2023). To overcome this obstacle, we generated knock-in (KI) mice in which a 3X-Flag tag was inserted at the C-terminus of Sertm2 (Sertm2-Flag KI) (Choi et al, 2023; Ferrando et al, 2015) (Figs. 3A and EV2F,G). Successful tagging of the *Sertm2* transcript with the Flag epitope was confirmed by qPCR, which exclusively detected the Flag sequence in the spinal cord of KI mice, which was absent from control mice (Fig. 3B,C). Utilizing

**A**

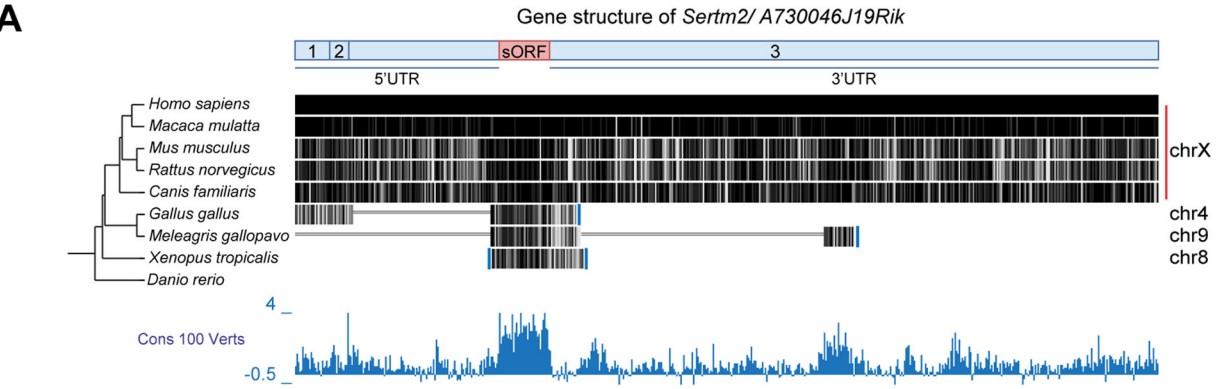

**B**

Transmembrane domain

```
Homo sapiens      MMEAHFKYHGN-LTGRAHFPTLATEVD-TSSDKYSNLYMYVGLFLSLLAILLILLFTMLLRLKHVISPINS--DSTESVPQFTDVEMQSRIPTP
Mus musculus      MTEVLFKYHGN-LTGRAHFPTLATEAD-TTSDKYSNLYMYVGLFLSLLAILLILLFTMLLRLKHVISPINS--DSTESVPQFTDVEMQSRIPTP
Gallus gallus     MTEIYFKFHGN-LTGRVHFPTLATEVD-NTADKYSGLYVYAVLFLTLLAILLILLFTMLLRLKHVVSPVTTPPESTENVQQFTDVEMHFRIPTT
Xenopus tropicalis MTEAYFKYRGNRTIFVQQFPTVTTEATPTTADKYSTLYMYVGLFLGLLAVLLVLLFTMLLRIKHVISPITPNTESTDNMPQFTDLEMQGRPPNN
                   * *   **:**   :***:::**.  .::**** **:. *** ***:**:*********:***:**:.    :**:.: ****:**: * *.
```

**C**

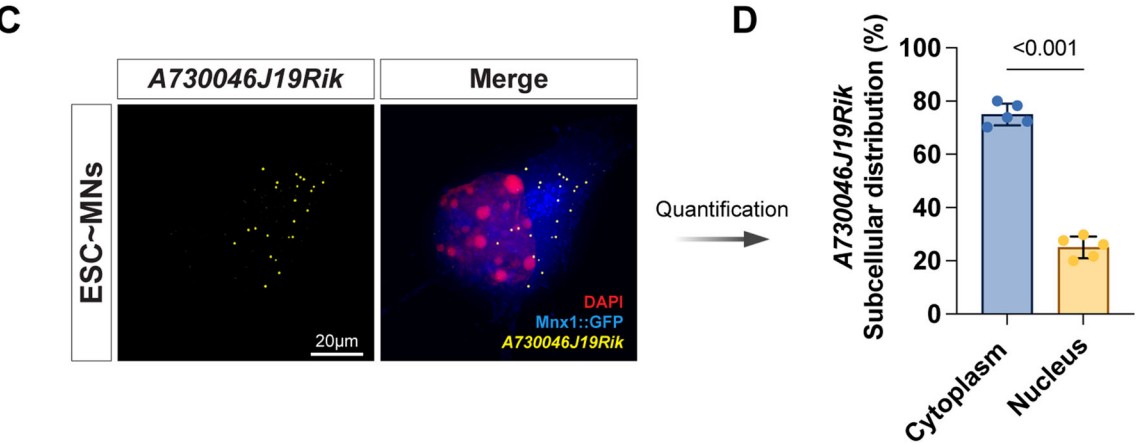

**D**

Quantification →

**E**

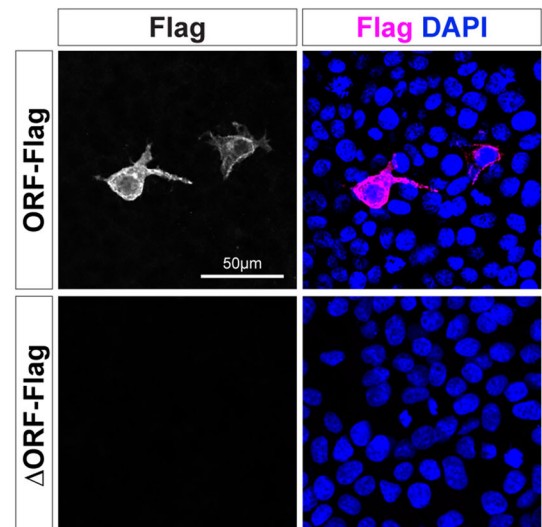

**F**

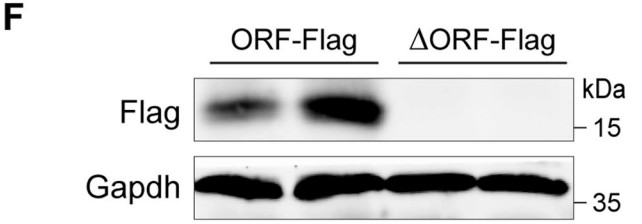

**Figure 2. *A730046J19Rik* encodes a transmembrane micropeptide, Sertm2.**

(A) Comparative sequence analysis illustrating the conservation of *A730046J19Rik* across species. The UCSC Conservation track (Cons 100 Verts) shows conservation scores across 100 vertebrate species, highlighting a highly conserved small open-reading frame (sORF, 270 bp) within exon 3 of the *A730046J19Rik* locus. UTR, untranslated region. (B) Amino acid sequence alignment of the sORFs from the indicated species. The transmembrane domain is highlighted within a colored square. The symbols below the alignment represent sequence similarity: asterisks (*) denote identical residues across all sequences, colons (:) indicate conserved substitutions, and periods (.) suggest similar conservation. (C, D) Single-molecule FISH (smFISH) experiments on mouse Mnx1::GFP ESC-derived MNs reveal the localization patterns of *A730046J19Rik* transcripts. A representative image is shown in (C). MNs were identified by the endogenous fluorescent Mnx1::GFP signal, and DAPI staining labels the nucleus. Scale bar, 20 μm. Additionally, quantitative measurements of *A730046J19Rik* expression across different subcellular compartments are shown in (D). Each dot represents one experimental batch, with results presented as mean ± SD from $n = 5$ independent experiments; unpaired two-tailed $t$ test. $P < 0.001$ ($P = 5.35 \times 10^{-8}$ for Cytoplasm vs. Nucleus). (E–G) The most conserved sORF of *A730046J19Rik* was engineered with a C-terminal Flag tag (ORF-Flag). In parallel, the start codon (ATG) was mutated to ATT to prevent translation, serving as a negative control (△ORF-Flag). HEK293T cells were transiently transfected with these constructs and analyzed by immunoblotting (F) and immunostaining (G). Flag detection confirms sORF translation, with Gapdh as an internal control in (F). Immunostaining (G) shows that the translated sORF is enriched in the cytoplasm, with DAPI labeling the nucleus. Scale bar, 50 μm.

the Sertm2-Flag KI mice, we employed a mouse monoclonal anti-Flag antibody to detect the endogenous Sertm2 protein. Western blotting of various tissues from KI mice revealed that Sertm2 is predominantly enriched in the spinal cord and muscle tissue (Fig. 3D). Interestingly, we found that the detected size of Sertm2 protein in vitro and in vivo did not correspond to the predicted size of ~10 kDa (Figs. 2F and 3C), implying potential post-translational modifications (PTMs). To investigate this possibility, we examined the UniProt database (Fig. EV2H) and used prediction tools such as FindMod to identify potential PTMs in Sertm2. Our analysis revealed a predicted glycosylation site. Strikingly, by using PNGase F treatments to deglycosylate the protein, the molecular weight of Sertm2 more closely matched the predicted value (Fig. EV2I), indicating that Sertm2 is a glycosylated micropeptide. Moreover, immunostaining at high magnification further demonstrated that Sertm2 expression is largely localized to the cytoplasmic region of ChAT⁺ MNs in the spinal cord at P7 (Fig. 3E). These findings confirm the spatial expression pattern of Sertm2 and its presence in spinal MNs, supporting its physiological relevance.

Next, the enrichment of Sertm2 in muscle tissues prompted us to investigate if the unusually long untranslated regions (UTRs) of Sertm2 might facilitate its transport along axons. In support of this hypothesis, we identified numerous AU-rich elements (AREs) in the 3′ UTR of *Sertm2*, which are known to promote mRNA trafficking within axons (Fig. 3F) (Hong and Jeong, 2023; Loedige et al, 2023). To explore this outcome further, we generated Sertm2-Flag KI ESCs and differentiated them into MNs with long axons. Remarkably, we observed an abundant distribution of the *Sertm2* transcript within the MN axons (Fig. EV2J). Given that *Sertm2* was highly co-expressed with *Etv4* motor pools that are induced by peripheral glial cell line-derived neurotrophic factor (GDNF) signaling from muscles (Haase et al, 2002), we further tested if GDNF can affect Sertm2 distributions. Strikingly, upon addition of GDNF, levels of the Sertm2 protein were significantly upregulated in the MN axons, indicating that the Sertm2 micropeptide might be produced in response to GDNF signaling (Fig. 3G). Based on these results, we postulate that Sertm2 is a GDNF-induced micropeptide in spinal MNs.

### *Sertm2* impairment leads to a reduction in Etv4⁺ motor pools

To explore the potential functions of *Sertm2* in differentiated MNs, we utilized a CRISPR/Cas9 approach to delete the entire *Sertm2*

locus in mouse ESCs (Fig. 4A), enabling us to analyze the resulting phenotypes during MN differentiation (Fig. 4B). Successful deletion of the *Sertm2* locus was confirmed by DNA sequencing (Fig. EV3A,B), and absence of its encoded transcript was further verified through qPCR (Fig. 4C). Furthermore, expression of two genes neighboring *Sertm2*, i.e., *Dcx* and *Alg13*, remained unchanged, indicating that *Sertm2* exerts no obvious *cis*-regulatory role (Fig. EV3C,D). Unlike the regular MN differentiation protocols we have adopted previously (Chen et al, 2023; Tung et al, 2015; Tung et al, 2019), we supplemented GDNF following dissociation, as *Etv4* expression is driven by peripheral GDNF signaling (Dasen et al, 2005; Haase et al, 2002). Doing so allowed us to investigate if *Etv4* expression is affected by *Sertm2* deletion. Under these conditions, we observed comparable gene expression of *Mnx1*⁺ cells between *Sertm2* knockout (KO) MNs and control (Ctrl) MNs (wild-type ESC line) (Fig. 4D). Whereas GDNF significantly induced *Etv4* expression in the Ctrl MNs, *Etv4* expression was notably reduced in the *Sertm2* KO ESC-derived MNs (Fig. 4F). Interestingly, expression of the upstream regulator *Hoxc8* remained unchanged, indicating that Sertm2 exerts a specific role in regulating *Etv4* pool identity through GDNF signaling (Fig. 4E). These findings prompted us to investigate the function of Sertm2 further by generating *Sertm2* knockout mice. Consistent with our results from ESC-derived MNs, we found that deletion of *Sertm2* did not impact the generation of motor neuron progenitors (Olig2⁺) or generic MNs (Isl1/2⁺ and Mnx1⁺) (Fig. EV3E). Furthermore, *Sertm2* knockout did not significantly affect the development of most columnar MMC (Lhx3⁺) and LMC (Foxp1⁺) neurons (Fig. 4G). The MMC subtypes in the spinal cord of *Sertm2* mutant mice showed no significant alterations (Fig. EV3F,G). However, the Etv4⁺ motor neuron pools were drastically reduced among Foxp1⁺ MNs, including both the Isl1⁺ pools, which innervate the cutaneous maximus (CM) muscle, and the Isl1⁻ pools that innervate the latissimus dorsi (LD) muscle (Fig. 4H–L). Taken together, our in vitro differentiation and in vivo embryo analyses indicate that the primary function of *Sertm2* in spinal MNs is to specify Etv4⁺ MN populations through peripheral GDNF signaling.

### Loss of *Sertm2* prompts Etv4⁺ motor nerve erosion and a coordinated motor deficit

To investigate further the impact of *Sertm2* loss-of-function on Etv4 expression within the LMC population in vivo, we analyzed the motor axon trajectory patterns of the Etv4⁺ motor pool at E13.5

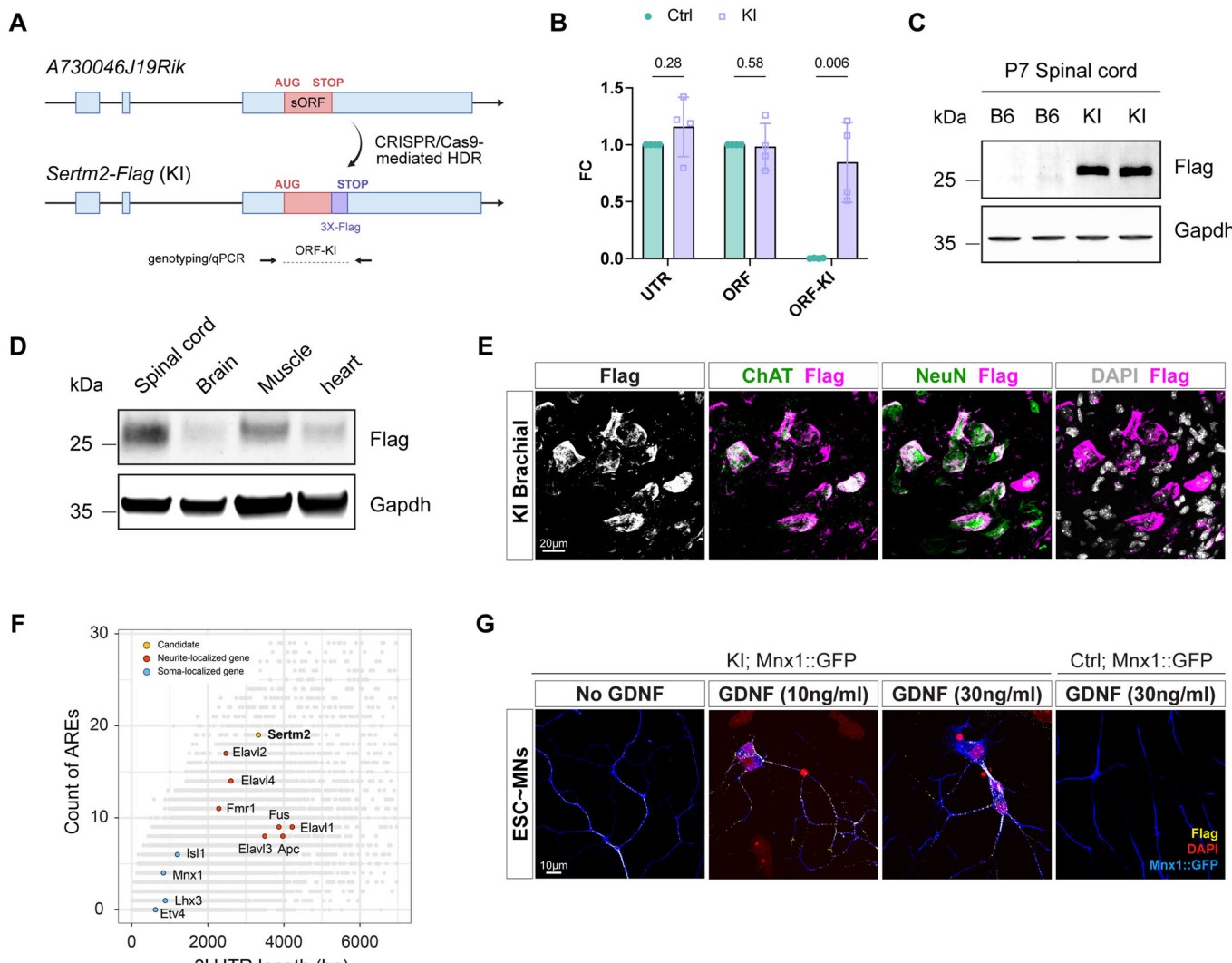

**Figure 3. Endogenous expression and function of Sertm2.**

(A) Schematic of the genome engineered for CRISPR/HDR-mediated knock-in (KI) of a sequence encoding a C-terminal 3X-Flag tag into the sORF region (light red rectangle) of the *A730046J19Rik* locus in a C57BL/6 (B6) mouse model. The 3X-Flag tag is highlighted in pale purple. HDR, Homology-directed Repair. Genotyping or qPCR primers for [ORF-KI] amplify the ORF-3X-Flag fusion region, spanning the light red to pale purple regions. (B) Verification of Flag insertion in Sertm2 by qPCR analysis using various primer pairs targeting specific regions. The x-axis indicates primers specific to regions within *A730046J19Rik* or Sertm2-Flag KI. Data are presented as fold change (FC) relative to the control expression level and are shown as mean ± SD, with $n = 4$ independent biological samples per group. The primers for each group are shown in (A) and Fig. 4A. Normalization: [UTR] expression in KI samples was normalized to control [UTR], while [ORF] and [ORF-KI] expression were normalized to control [ORF]. Statistical analysis was conducted using multiple unpaired *t* tests. *P* values for *UTR*, *ORF*, and *ORF-KI* (Ctrl vs. KI) were 0.28, 0.58, and 0.006, respectively. (C, D) Immunoblot analysis of Sertm2-Flag in the P7 spinal cord of B6 and Sertm2-Flag KI mice (C). Characterization of Sertm2-Flag protein expression in various tissues from P7 Sertm2-Flag KI mice (D). Anti-Flag detects the Sertm2 endogenous protein. Gapdh was used as a loading control. (E) Immunostaining for Flag in the P7 Sertm2-Flag KI brachial spinal cord reveals enrichment in neurons labeled with ChAT and NeuN. Nuclei are highlighted by DAPI staining. Scale bar, 20 μm. (F) Scatter plot showing the number of AREs in the 3′ UTRs of the indicated genes. Known neurite-localized (red) and soma-localized (blue) genes are plotted. AREs, AU-rich elements. The data was derived from Loedige et al, 2023. (G) Representative images showing the distribution of Flag-tagged Sertm2 protein in the axons of Ctrl; Mnx1::GFP and KI; Mnx1::GFP mouse ESC-derived MNs, with varying levels of GDNF supplementation. The Flag tag indicates Sertm2 protein expression, with motor neuron axons labeled by endogenous Mnx1::GFP and nuclei highlighted by DAPI staining. Scale bar, 10 μm.

by crossing *Sertm2* KO mice with the Mnx1::GFP reporter line (Chang et al, 2021; Chen et al, 2023). Examination of the forelimb nerves innervating the CM and LD muscles—located in the subcutaneous tissues of the trunk and back, respectively—revealed significant motor axon arborization defects in the *Sertm2* mutant mice (Fig. 5). Specifically, we observed reduced nerve branching in the LD muscle and diminished coverage in the CM muscle

(Fig. 5A,B). We reasoned that impaired specification of the Etv4⁺ motor pool and erosion of the associated motor nerves could elicit deficits in motor coordination, particularly affecting delicate, limb-driven behaviors. Accordingly, we assessed basal motor activity and locomotor coordination in control and mutant mouse lines by means of rotarod, beam walking, grip strength, and treadmill tests. In line with our predictions, *Sertm2* KO mice at approximately P30

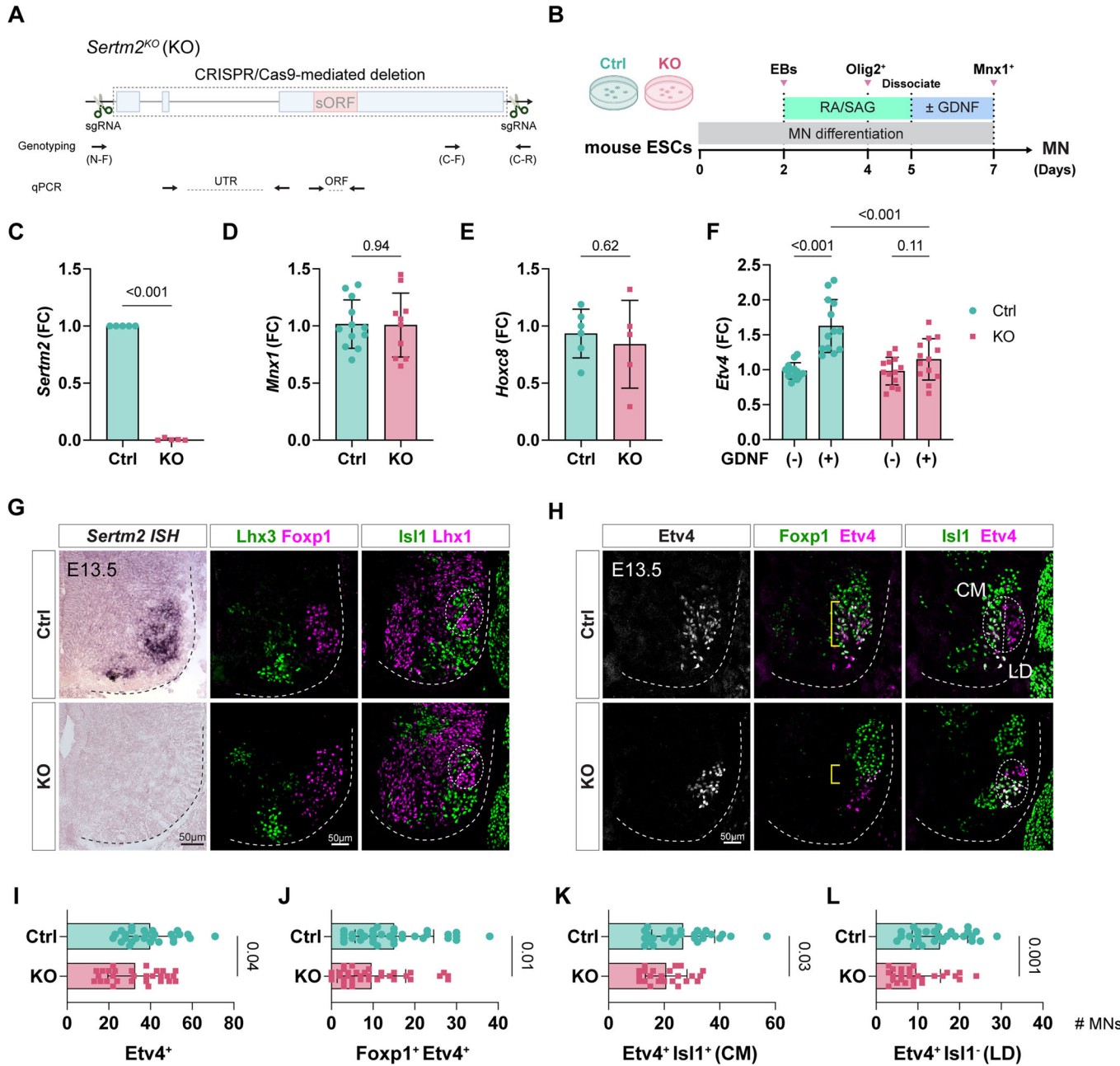

exhibited a significant increase in slippage rate during the beam walking test (Figs. 5C,D and EV4A; Movie EV1). To further assess fine motor skills, we employed a treadmill to examine the detailed kinematics of limb coordination and gait patterns (Chang et al, 2021). Remarkably, we observed widespread motor skill impairments across the *Sertm2* KO mice (Fig. 5E–G; Movie EV2 and see Methods for details). Phase analyses of interlimb coordination revealed that although homolateral limb and synchronous diagonal limb gaits—indicative of spinal interneuron circuit integrity—were unaffected, alternating homologous gaits that reflect spinal motor neuron output were significantly compromised in our *Sertm2* KO mice (Fig. 5H,I). These findings indicate that the ventral motor circuitry is disrupted in the absence of *Sertm2*. Moreover, these

coordination deficits persisted into adulthood (P90–120) (Fig. EV4B,C). Collectively, these results underscore the indispensable role of *Sertm2* in maintaining the proper physiological function of the Etv4+ motor pool.

## Sertm2 administration rescues *Etv4* downregulation in mouse *Sertm2* KO ESC-derived MNs

To investigate further if the Sertm2 micropeptide, or its encoding *Sertm2* RNA, regulates *Etv4* expression during MN differentiation, we used CRISPR/Cas9 to delete a 10-bp region from the middle of the Sertm2 micropeptide in a mouse ESC line (frameshift, FS) (Figs. 6A and EV5A). This approach allowed us to confirm that the

◄  **Figure 4.  Corroborating the MN phenotype elicited by *Sertm2* knockout.**

(A) Generation of the *Sertm2* knockout (KO) mouse cell line and mice using CRISPR/Cas9 gene editing. The paired sgRNAs targeted the entire *Sertm2* locus (dashed rectangle) on the X chromosome. Primers designed for genotyping and qPCR analyses are indicated in the figure. qPCR primers for [UTR] amplify the region spanning exon 2 to exon 3, and [ORF] primers target the sORF region. (B) Experimental timeline for cultured mouse control (Ctrl) and KO ESC-derived MNs, incubated with or without GDNF. (C–E) qPCR analyses of *Sertm2*, *Mnx1*, and *Hoxc8* expression at Day 7 in Ctrl and *Sertm2* KO ESC-derived MNs cultured without GDNF. Data are presented as fold change (FC) relative to Ctrl and shown as mean ± SD from $n = 5$–12 independent experiments; unpaired two-tailed $t$ test. For *Sertm2* (C), *Mnx1* (D), and *Hoxc8* (E), the $P$ values (Ctrl vs. KO) were <0.001 ($P = 1 \times 10^{-15}$), 0.94, and 0.62, respectively. (F) GDNF supplementation significantly increases *Etv4* transcript levels in Ctrl ESC-derived MNs but not *Sertm2* KO MNs. Data are shown as fold change (FC) relative to Ctrl [GDNF (−)] and represent mean ± SD from $n = 13$ independent experiments; ordinary one-way ANOVA. $P < 0.001$ ($P = 1.57 \times 10^{-7}$) for GDNF (−) vs. (+) in Ctrl, $P = 0.11$ for GDNF (−) vs. (+) in KO, and $P < 0.001$ ($P = 3.57 \times 10^{-5}$) for GDNF (+) between Ctrl and KO. (G) E13.5 brachial spinal cord sections from Ctrl and KO mice were analyzed by in situ hybridization for *Sertm2*, combined with immunostaining for MN markers Lhx3, Foxp1, Isl1, and Lhx1 to identify columnar MN subtypes. Circles demarcate LMC subtype (LMCl and LMCm), and dashed lines outline the spinal cord boundary. Scale bars: 50 μm. (H) Immunostaining of the Etv4-expressing MN populations in E13.5 brachial spinal cord sections from Ctrl and KO mice. Brackets in the middle panel highlight co-expressing cells, while circles in the right panel indicate Etv4$^+$ MN subtypes within LMC. Dashed lines outline the spinal cord boundary. Cutaneous maximus (CM), Etv4$^+$Isl1$^+$. Latissimus dorsi (LD), Etv4$^+$Isl1$^-$. Scale bar: 50 μm. (I–L) Quantification of subsets of Etv4-expressing MNs in Ctrl and KO brachial spinal cords, as shown in (H). Data represent mean ± SD from $n = 10$ independent biological samples, with three rostral-caudal sections per sample. Statistical analysis was performed using an unpaired two-tailed $t$ test. For Etv4$^+$ (I), Foxp1$^+$ Etv4$^+$ (J), Etv4$^+$ Isl1$^+$ (K), and Etv4$^+$ Isl1$^-$ (L) MNs, the $P$ values (Ctrl vs. KO) were 0.04, 0.01, 0.03, and 0.001, respectively.

major *Sertm2* transcript (*Sertm2*_UTR) and Mnx1$^+$ cells remain stable despite the resulting frameshift in *Sertm2* (*Sertm2*_ORF) (Fig. 6B–D). Notably, whereas GDNF significantly induced *Etv4* expression in our control MNs, *Etv4* expression was markedly reduced in the FS-MNs (Fig. 6E), indicating that *Etv4* is primarily mediated through the Sertm2 micropeptide.

To determine if Sertm2 administration could rescue *Etv4* downregulation in the mouse *Sertm2* (full length 4.7 kb deletion) KO ESC-derived MNs, we utilized a lentivirus (LV) system to deliver either wild-type Sertm2 (270 bp) or ATG-mutated Sertm2 (as a control) into mouse *Sertm2* KO ESCs, which were then differentiated into MNs (Fig. 6F). Overexpression of Sertm2 micropeptide in the mouse *Sertm2* KO cells was confirmed by western blot and qPCR, verifying both protein and RNA levels (Fig. 6G,H). We observed that ectopic expression of wild-type Sertm2 protein, but not the mutant version, significantly rescued *Etv4* expression in the mouse KO cells (Fig. 6I), substantiating the evidence that Sertm2 is a critical and functional micropeptide that modulates the GDNF-Etv4 signaling pathway in MNs.

## The function of human SERTM2 is conserved with that of murine Sertm2

Given that *Sertm2* is highly conserved among vertebrates (Fig. 2A), we sought to establish if its human ortholog, human *SERTM2* (a putative micropeptide derived from the lncRNA *LINC00890*), is conserved in terms of sequence, subcellular localization, and function. First, we assessed conservation of their exonic structures, sequences, and genomic positions (Guo et al, 2020; Ulitsky, 2016). According to GENCODE annotations, there are two *SERTM2* isoforms; the major isoform contains three exons, i.e., similar to mouse *Sertm2*, whereas the other isoform is shorter and only has two exons. Sequence analysis revealed that mouse *Sertm2* shares 91% similarity with the major human isoform (Fig. 7A). In addition, both genes have conserved upstream and downstream neighboring genes, namely *ALG13* and *DCX*, indicating that human *SERTM2* is conserved with murine *Sertm2* in terms of both sequence and genomic contexts (Fig. EV3C).

Next, we examined the expression pattern of human *SERTM2* during MN differentiation. To do so, we differentiated a human ESC line harboring the MN reporter (HuES3 MNX1::GFP) into

spinal MNs (Fig. 7B) (Di Giorgio et al, 2008), and then performed strand-specific RNA sequencing at three stages of differentiation, i.e., motor neuron progenitors (pMNs, Day 8), nascent postmitotic MNs (Day 16), and mature MNs with neurites (Day 31) (Figs. 7C and EV5B,C). Consistent with our findings in mouse MNs (Fig. 1B,C), we observed that human *SERTM2* was highly and specifically enriched in the postmitotic MNs. Notably, *GDNF* and its receptor *RET* were also strongly expressed in the mature MNs. These results were further validated by qPCR on MNs derived from another human induced pluripotent stem cell (iPSC) line harboring the MNX1-tdTomato (MNX1-tdT) reporter (Fig. 7D) (Garcia-Diaz et al, 2020).

Similar to mouse *Sertm2*, we found that human *SERTM2* predominantly localized in the cytoplasm of iPSC-derived MNs, using *GAPDH* and *U1* as controls for cytoplasmic and nuclear fractions, respectively (Fig. 7E). Moreover, to establish if the function of the GDNF-SERTM2-ETV4 axis is conserved in humans, we mutated the entire locus of *SERTM2* in the MNX1-tdT iPSC line (Figs. 7F and EV5D,E), and verified the absence of *SERTM2* expression from the resulting *SERTM2* mutant iPSC~MNs (KO) (Fig. 7G). Similarly, we also observed significant downregulation of *ETV4* expression, whereas columnar *FOXP1* expression was comparable to control (Fig. 7H,I). This result recapitulates the expression patterns we observed in our mouse *Sertm2* KO ESC-derived MNs and embryos (Fig. 4). Notably, we also observed a reduction in neurites, as revealed by the SMI-32 marker, in our *SERTM2* KO MNs (Fig. 7J–L). Together, these findings indicate that human *SERTM2* exerts a conserved role in *ETV4*$^+$ MN subtype development, similar to *Sertm2* in mice.

## Discussion

Recent research has unveiled that certain lncRNAs encode micropeptides, small proteins with crucial regulatory roles. These lncRNA-derived micropeptides are gaining recognition for their involvement in neural development, where they modulate key signaling pathways, influence neuronal differentiation, and contribute to axonal guidance and synaptic function (Duffy et al, 2022). These findings indicate that micropeptides may play previously unappreciated but essential roles in neural development and

   

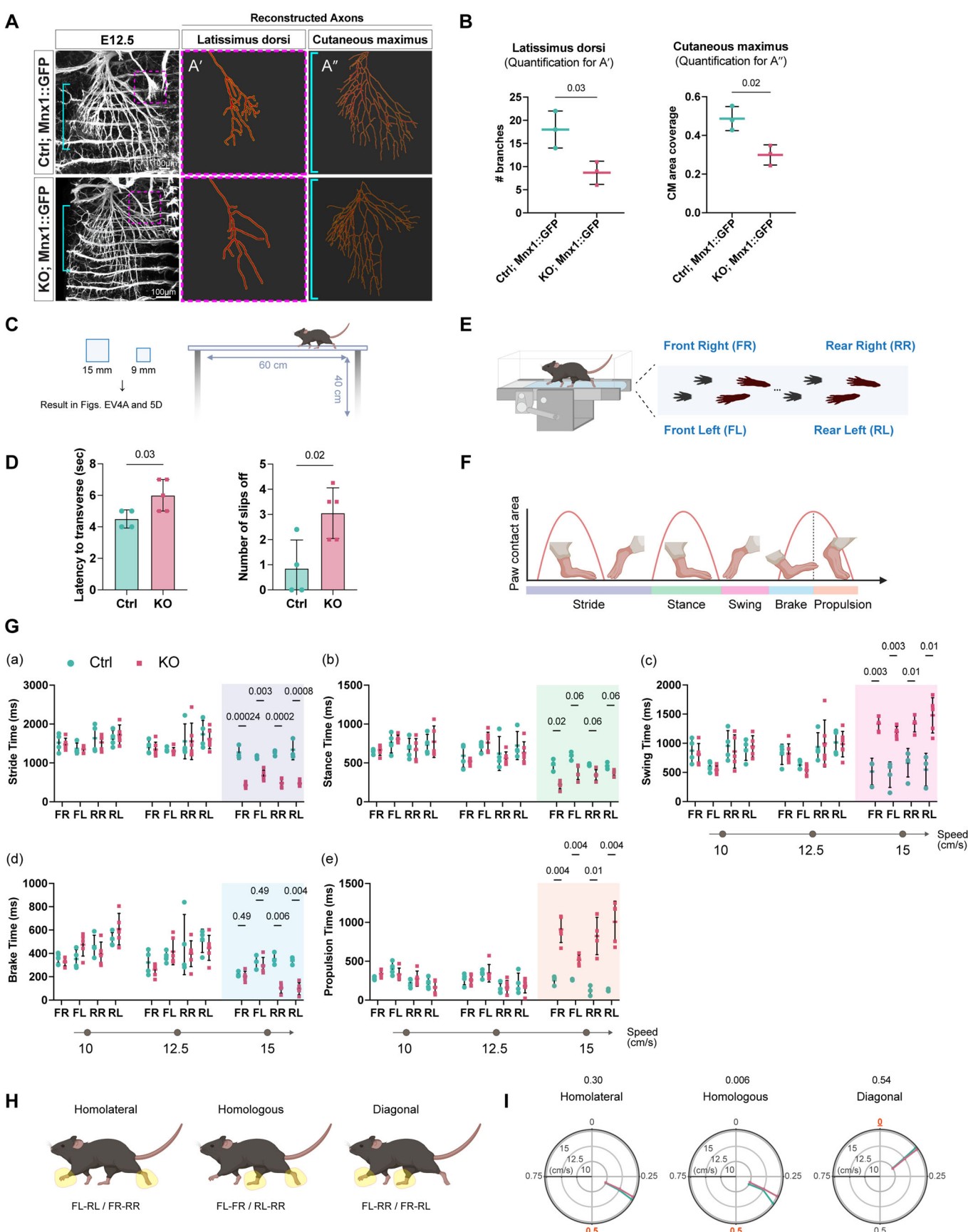

◀

**Figure 5. Impact of impaired Etv4⁺ motor pools in *Sertm2* mutant mice.**

(A, B) Neurite arborization in the latissimus dorsi (LD) muscle and cutaneous maximus (CM) muscle of E12.5 Ctrl; Mnx1::GFP and *Sertm2* KO; Mnx1::GFP (KO; Mnx1::GFP) mice, respectively. The magenta dashed squares in (A) indicate LD muscle regions, with axonal reconstructions displayed in (A′). Similarly, cyan brackets in (A) highlighted the CM muscle, with corresponding reconstruction in (A′′). Scale bars: 100 μm. (B) Presents the quantification of axonal branches in the LD muscle from (A′, left), and innervation area of CM axons normalized to body length (right). Data represent mean ± SD from $n = 3$ independent biological samples, with statistical analysis conducted using unpaired two-tailed $t$ tests. $P = 0.03$ (LD) and $P = 0.02$ (CM) for Ctrl; Mnx1::GFP vs. KO; Mnx1::GFP. (C, D) A schematic of the balance beam test is shown in (C), featuring two beam diameters: 9 mm and 15 mm. (D) Presents the results for the 9-mm beam, comparing Ctrl and KO mice at P30 in terms of traversal time (left) and the number of foot slips (right). Data represent mean ± SD from $n = 4$–5 independent biological samples, with statistical analysis performed using an unpaired two-tailed $t$ test. $P = 0.03$ (Latency to transverse) and $P = 0.02$ (Number of slips off) for Ctrl vs. KO. Results for the 15-mm beam are provided in Fig. EV4A. (E) Schematic representation of gait spatial characteristics in treadmill analysis. FR front right paw, FL front left paw, RR rear right paw, RL rear left paw. (F) Illustration of gait cycle paw contact area plots (red lines) along with gait parameters for each limb, including stride, stance, swing, break, and propulsion times, as assessed by treadmill analysis. (G) Treadmill walking parameters measured include (a) stride time, (b) stance time, (c) swing time, (d) brake time, and (e) propulsion time at three different walking speeds (10, 12.5, 15 cm/s) in P30 Ctrl and KO mice. Significant changes at a speed of 15 cm/s are highlighted in color in each panel. Data are presented as mean ± SD from $n = 4$–5 independent biological samples, with statistical analysis conducted using multiple unpaired $t$ tests. All $P$ values shown in the figures represent comparisons between Ctrl and KO. (H) Illustration of temporal interlimb coordination during walking, with phase intervals for pairs of limbs described below and examples highlighted in yellow. (I) Limb coordination (homolateral, homologous, and diagonal phase coupling) is represented as polar plots at different walking speeds. Phase values of 0 (or 1) correspond to perfect synchronization, while a phase value of 0.5 indicates strict alternation. The ideal phase values for each coordination type are highlighted in red and underlined. Statistical results for the speed of 15 cm/s are shown above each plot. Data are from $n = 4$–5 independent biological samples at P30, analyzed using an unpaired two-tailed $t$ test. $P = 0.30$ for homolateral, $P = 0.006$ for homologous, and $P = 0.54$ for diagonal comparisons between Ctrl and KO.

potentially in the progression of neurodegenerative diseases (Liaci et al, 2022; Salvatori et al, 2020). Notably, lncRNAs are highly enriched in the central nervous system (CNS), a pattern likely reflecting their critical functions in regulating gene expression, particularly during the complex processes of brain development and function (Srinivas et al, 2023). The tissue-specific and developmental stage-specific nature of lncRNA expression implies that they contribute to the fine-tuning of gene networks essential for neural development, synapse formation, and neural plasticity (Chen and Chen, 2020; Yen et al, 2018). This scenario raises a fascinating question, i.e., do CNS-enriched lncRNAs exert their functions primarily through their RNA molecules or do they act via encoded micropeptides? In this study, we have identified Sertm2, a lncRNA-derived micropeptide highly conserved across vertebrates, and rigorously demonstrated its functional role in neural development using genetic knockout and knock-in strategies. Furthermore, we employed micropeptide rescue experiments, widely regarded as the gold standard for validating the physiological functions of neuropeptides in vivo. Our findings establish a role for Sertm2 in spinal MN differentiation and specification, providing compelling evidence for its significance in neural development.

Why do Etv4⁺ motor pools specifically require the function of the Sertm2 micropeptide? In the spinal cord, MNs are organized along the rostrocaudal axis into distinct columnar types (Dasen et al, 2003). Within the limb LMCs, MNs are further grouped into more than sixty motor pools, each innervating a specific muscle. This organization is largely controlled by intrinsic mechanisms, particularly via a Hox-mediated clustering process, with most MN diversity being established early on, i.e., before MNs reach their final positions (Chen and Chen, 2019; Kania and Jessell, 2003; Sockanathan and Jessell, 1998). Interestingly, extrinsic signals from the periphery also play a critical role in motor pool identity. One such extrinsic factor is GDNF, initially known for its role in MN survival, but now recognized for its early presence in the developing forelimb plexus and later localization in the cutaneous maximus and latissimus dorsi muscles (Livet et al, 2002). In the absence of GDNF signaling, MNs that normally innervate these muscles are mispositioned within the spinal cord, and their axonal invasion of target muscles is significantly impaired. Etv4, a member of the ETS

family of transcription factors that is typically expressed by these MNs, fails to be induced in the absence of GDNF signaling, resulting in aberrant MN positioning and defective muscle innervation (Arber et al, 2000). Thus, GDNF acts as a peripheral signal to induce Etv4 expression in specific MN pools, regulating both the positioning of MN cell bodies and their ability to innervate target muscles (Arber et al, 2000; Haase et al, 2002; Koo and Pfaff, 2002). Recent studies have indicated that Hoxc8, a key Hox protein expressed in the brachial spinal cord, regulates the expression of *Ret* and *Gfrα* genes, which encode the receptor tyrosine kinases and GPI-anchored co-receptors, respectively, that mediate the effects of GDNF family ligands (Catela et al, 2016). Hoxc8 confers sensitivity to peripheral GDNF signaling in a subset of MNs, enabling the induction of Etv4 expression within specific motor pools (Catela et al, 2016). This mechanism ensures the precise fidelity of synaptic specificity within proprioceptive circuits (Shin et al, 2020). Given the role of GDNF in regulating Etv4 expression and MN function, as a potential modulator of GDNF signaling, the Sertm2 micropeptide may be essential for the proper differentiation and function of Etv4⁺ motor pools. In this study, we have identified Sertm2 as a critical factor that modulates GDNF signaling to induce Etv4 expression. *Sertm2* knockout results in significant impairment of the GDNF-induced Etv4⁺ motor pool, highlighting its essential role in this process. Moreover, we observed that the 3′ UTR of Sertm2 is enriched with ARE motifs, known to enable axonal trafficking (Loedige et al, 2023). This observation is supported by the enriched distribution of *Sertm2* mRNA in MN neurites, indicating a potential role for Sertm2 in axonal transport. Notably, we observed that GDNF stimulation drastically increased Sertm2 protein expression in both cell bodies and neurites, indicating that GDNF may promote Sertm2 activity to autoregulate the GDNF-Etv4 signaling axis. Further investigations, including deletion of the 5′ and 3′ UTRs of Sertm2, are warranted to identify the precise motifs responsible for induction of this micropeptide. Moreover, using Sertm2-Flag to pull down the mass spectrometry-determined Sertm2 interactome in the spinal cord would enable the identification of possible candidates responsible for how Sertm2 modulates the GDNF signaling pathway. Doing so would also clarify if Sertm2 is locally induced at neuromuscular junctions or within MN cell

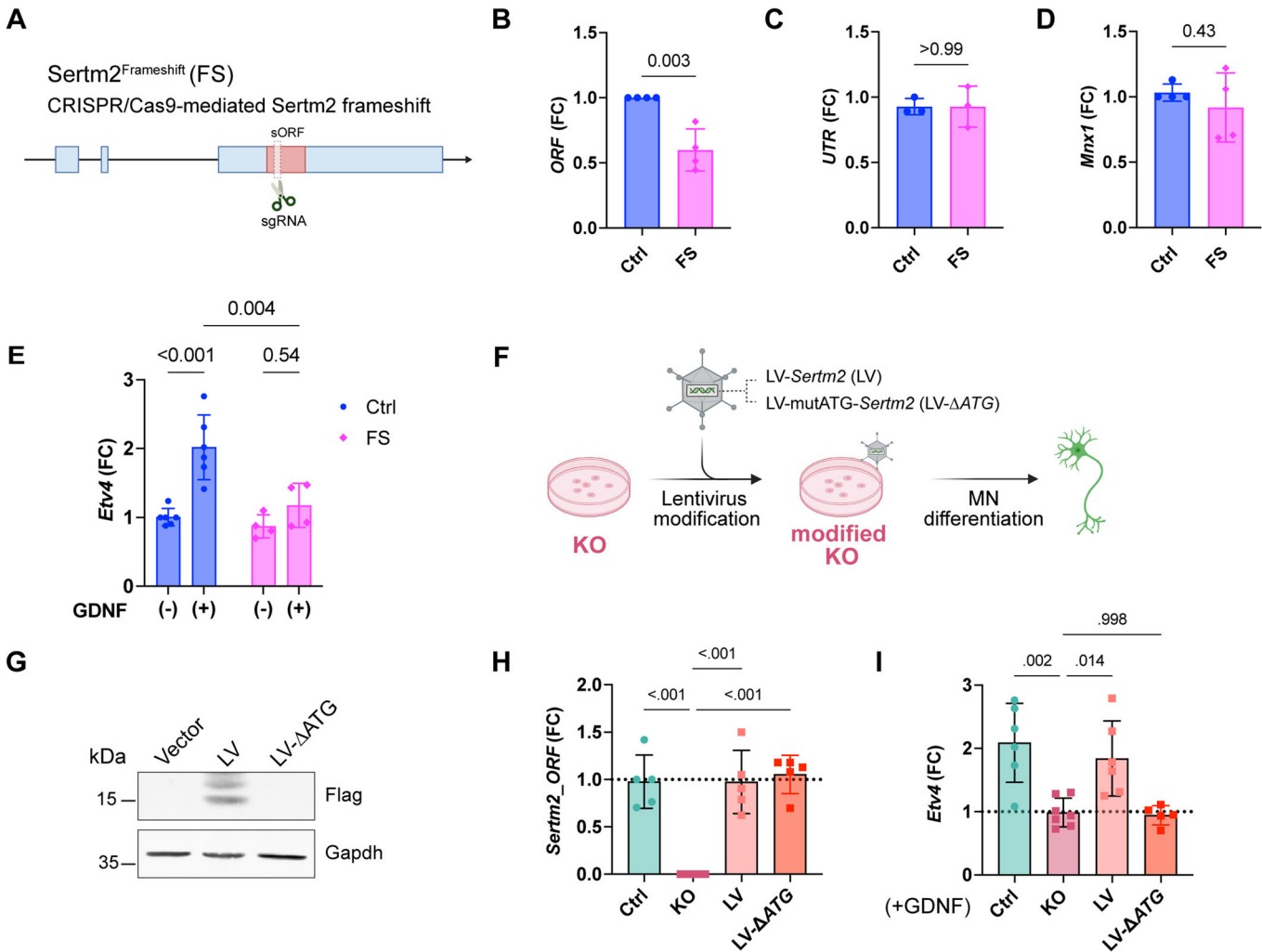

**Figure 6. Sertm2 locus plays a role in regulating Etv4 expression via the micropeptide Sertm2.**

(A) CRISPR/Cas9-induced frameshift mutation in the sORF of *Sertm2* in a mouse cell line carrying the Mnx1::GFP transgenic reporter, Sertm2^Frameshift (FS). The sgRNA targets the *Sertm2* sORF, resulting in a partial deletion. (B–D) qPCR analyses of the *ORF* and *UTR* regions of *Sertm2* (see Fig. 4A), and *Mnx1* in Ctrl and FS ESC-derived MNs. Data are presented as fold change (FC) relative to Ctrl and shown as mean ± SD from n = 3–4 independent experiments; unpaired two-tailed t test. P values for *ORF*, *UTR*, and *Mnx1* (Ctrl vs. FS) were 0.003, >0.99, and 0.43, respectively. (E) Consistent with mouse *Sertm2* KO MNs, *Etv4* RNA levels are reduced in FS ESC-derived MNs following the treatment shown in Fig. 4B. Data are presented as fold change (FC) relative to Ctrl [GDNF (−)] and shown as mean ± SD from n = 4–6 independent experiments; ordinary one-way ANOVA. $P < 0.001$ ($P = 2.22 \times 10^{-4}$) for GDNF (−) vs. (+) in Ctrl, $P = 0.54$ for GDNF (-) vs. (+) in FS, and $P = 0.004$ for GDNF (+) between Ctrl and FS. (F) Schematic illustration of lentivirus-mediated overexpression in *Sertm2* KO ESC-derived MNs. Plasmids containing either the wild-type (LV) or ATG-mutated sORF (LV-ΔATG) of Sertm2, tagged with Flag, were packaged into lentiviruses and delivered into mouse *Sertm2* KO ESCs for subsequent MN differentiation. (G) Western blot analysis of lentivirus-mediated overexpression of Sertm2 protein following transfection. Flag indicates Sertm2 protein, and Gapdh serves as a loading control. (H, I) qPCR analyses of transduced MN culture reveal rescued expression of *Sertm2_ORF* (H) and *Etv4* RNA (I) in mouse *Sertm2* mutant ESC-MNs supplemented with GDNF. Data are shown as fold change (FC) relative to Ctrl, representing mean ± SD from n = 5–7 independent experiments; ordinary one-way ANOVA. $P < 0.001$ ($P = 4.58 \times 10^{-5}$ for Ctrl vs. KO; $P = 4.82 \times 10^{-5}$ for KO vs. LV; $P = 1.88 \times 10^{-5}$ for KO vs. LV-ΔATG) in (H), while $P = 0.002$ for Ctrl vs. KO; $P = 0.014$ for KO vs. LV; $P = 0.998$ for KO vs. LV-ΔATG in (I).

bodies, providing deeper insights into its regulatory role in GDNF-mediated MN function. While our study demonstrates that Sertm2 plays a crucial role in the specification of Etv4⁺ MN pools, two intriguing questions remain. First, while our findings demonstrate the bona fide protein-coding potential of *Sertm2* as a conserved micropeptide, we cannot entirely rule out the possibility that *Sertm2* may also have additional regulatory functions within motor neurons. Interestingly, given that the host lncRNA, *A730046J19Rik*, is also significantly expressed in MN nuclei, it is plausible that

*A730046J19Rik* could possess both RNA- and protein-mediated functions (Mattick et al, 2023; Wright et al, 2022), which remain to be fully explored. Additionally, a previous study showed that the *Tug1* locus exerts three distinct regulatory functions: acting in *cis* to regulate neighboring genes; in *trans* to modulate distant gene expression; and as a peptide to influence mitochondrial membrane potential, all contributing to its essential role in male fertility (Lewandowski et al, 2020). It is tantalizing to probe the full spectrum of functions of the *A730046J19Rik* locus by deleting

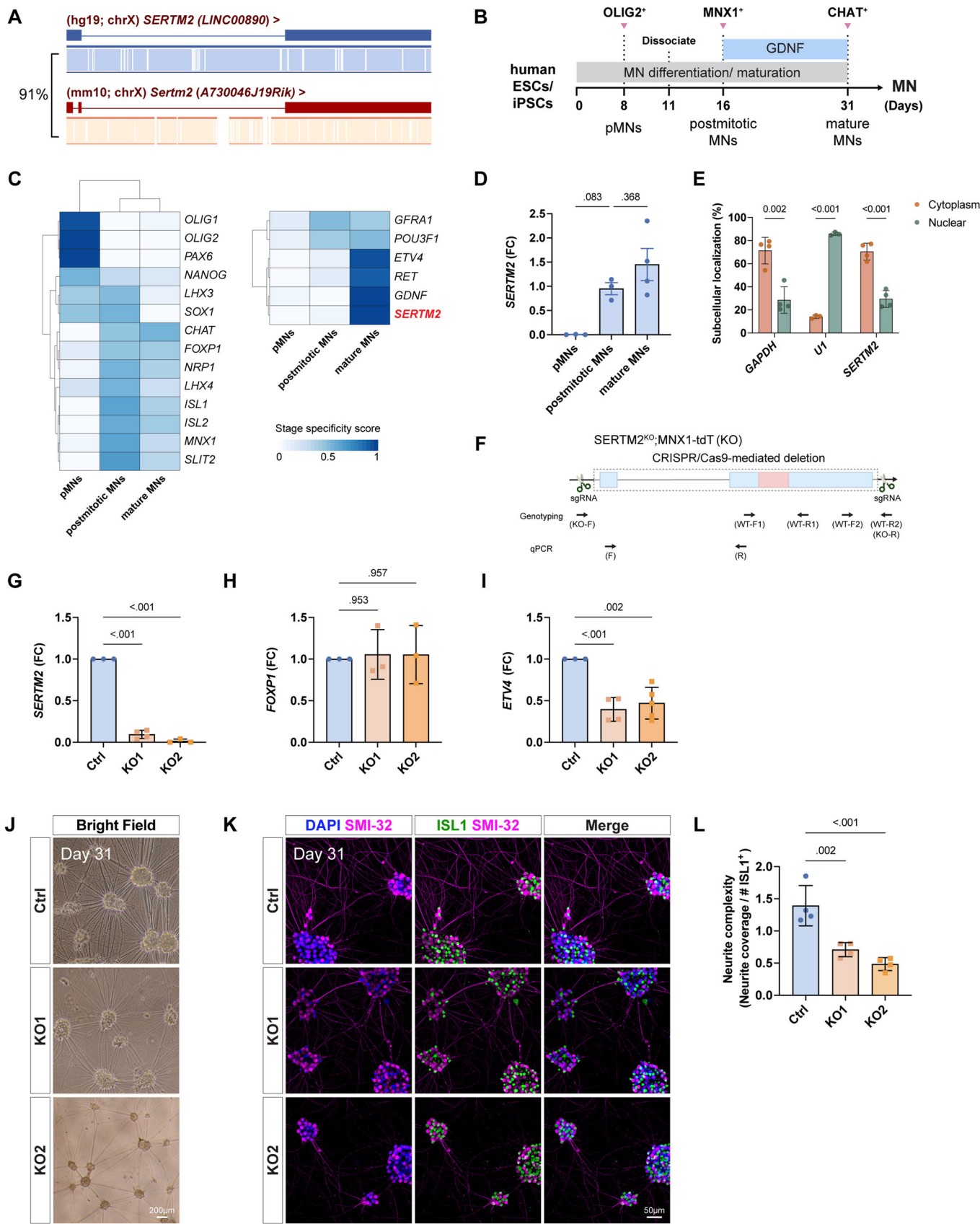

**Figure 7. Sequence and functional conservation of the human homolog *SERTM2*.**

(A) Sequence comparison between human *SERTM2* and mouse *Sertm2* reveals that the homologous genes, located on the X chromosome, share 91% sequence similarity. (B) Timeline of differentiation of human ESCs or iPSCs into spinal MNs. Key markers for the indicated stages are shown in the figure. pMNs, motor neuron progenitors. (C) Heatmap showing the abundance of known transcription factors and genes of interest across the stages of differentiation of the human ESC line (HuES3 MNX1::GFP) into spinal MNs. The blue gradient represents the stage specificity score. The strand-specific RNA-seq data analyzed in this figure have been deposited in GEO under accession number GSE275447. (D) Expression profile of the *SERTM2* transcript in human iPSC (MNX1-tdT)-derived MNs. Data are presented as fold change (FC) relative to postmitotic MNs (Day 16), with mean ± SD from $n = 3–4$ independent experiments; statistical analysis was performed using an unpaired two-tailed $t$ test. $P = 0.083$ (postmitotic MNs vs pMNs) and $P = 0.368$ (postmitotic MNs vs mature MNs). (E) Subcellular fractionation of MNX1-tdT iPSC-derived MNs at Day 31 reveals significant *SERTM2* transcripts in the cytoplasm. *GAPDH* and *U1* serve as markers for the cytoplasmic and nuclear fractions, respectively. Data are presented as mean ± SD from $n = 3–4$ independent experiments; multiple unpaired $t$ tests. $P$ values for *GAPDH*, *U1*, and *SERTM2* (Cytoplasm vs. Nuclear) were 0.002, <0.001 ($P = 1.37 \times 10^{-6}$), and <0.001 ($P = 3.24 \times 10^{-4}$), respectively. (F) Illustration of CRISPR/Cas9-mediated *SERTM2* deletion in the MNX1-tdT iPSC line referred to as KO. The paired sgRNAs targeted the entire *SERTM2* locus (dashed rectangle) on the X chromosome. Primers designed for genotyping and qPCR analyses are indicated in the figure. (G–I) qPCR analyses of Ctrl and KO iPSC-derived MNs at Day 31 reveal the downregulation of *SERTM2* and *ETV4* in the absence of *SERTM2*, whereas *FOXP1* expression remains unaffected. Data are presented as fold change (FC) relative to Ctrl, with mean ± SD from $n = 3–5$ independent experiments; ordinary one-way ANOVA. Here, Ctrl refers to the MNX1-tdT iPSC line, whereas KO represents two CRISPR-mediated *SERTM2* mutant iPSCs (KO1 and KO2) used for qPCR analyses. $P$ values for Ctrl vs. KO1 and Ctrl vs. KO2 in *SERTM2* (G) were $P < 0.001$ ($8.52 \times 10^{-9}$ and $7.49 \times 10^{-9}$); in *FOXP1* (H), they were $P = 0.953$ and 0.957; and in *ETV4* (I), they were $P < 0.001$ ($9.88 \times 10^{-4}$) and 0.002. (J) Brightfield microscopy-derived image of Ctrl and KO iPSC-derived MNs at Day 31. Scale bar, 200 μm. (K, L) Validation of MN identity and morphology by immunostaining for ISL1 and SMI-32 at Day 31 of iPSC-MN differentiation (K). SMI-32 highlights neurite formation, while DAPI staining labels the nuclei. Scale bar, 50 μm. Neurite complexity, quantified using ilastik, was normalized to the number of ISL1$^+$ neurons and is shown in (L). Data are presented as mean ± SD from $n = 4$ independent experiments; ordinary one-way ANOVA. $P$ values for Ctrl vs. KO1 and Ctrl vs. KO2 were $P = 0.002$ and <0.001 ($P = 2.31 \times 10^{-4}$).

specific regions and examining the resulting physiological consequences will provide critical insights. Future studies employing more systematic approaches are necessary to investigate these potential multifaceted roles and further elucidate the broader regulatory functions of Sertm2 and *A730046J19Rik* in MNs. Second, *Sertm2* is also expressed in MMC MNs, yet we did not observe a significant impact on MMC MN subtypes (i.e., Nr2f2 and Satb2) in the spinal cords of *Sertm2* KO mice. Given that the protein size of Sertm2 appears to vary in different cell contexts, it is likely that LMCs and MMCs might enforce distinct PTMs, thereby modifying Sertm2 to have differential functions in MN subtypes, representing an uncharted domain for further research. Our findings highlight the conserved function of human SERTM2 in iPSC-MNs, providing valuable insights into its evolutionary and functional conservation. However, the expression of the SERTM2 micropeptide in specific human MN subtypes in vivo has yet to be confirmed. Recent advancements in spinal cord organoid technology (Gribaudo et al, 2024), which may better capture MN diversity in a human context, could help address this question and facilitate functional studies of SERTM2 in human organoids.

The interplay between lncRNAs and their encoded peptides adds another layer of complexity to gene regulation in the CNS, indicating that a deeper investigation of micropeptides could yield insights into both development and disease. Our results open up new avenues for understanding the regulatory mechanisms at play during neural differentiation and raise intriguing possibilities for Sertm2's involvement in neurodegenerative conditions. A recent study uncovered that the micropeptide SHMOOSE might be linked to Alzheimer's disease (Miller et al, 2023), but whether micropeptides exert functional roles during MN neurodegeneration remains largely unknown. Micropeptides are small and usually secreted, so deciphering their biological roles may reveal potential applications as disease biomarkers and pave the way for developing new drugs for neurodegenerative diseases, such as amyotrophic lateral sclerosis (ALS). Given that GDNF is one of the critical targets for ongoing ALS clinical trials (Baloh et al, 2022), further study of the role of Sertm2 in spinal MNs and if it may be exploited as a

biomarker and a new therapeutic target for ALS could prove illuminating. Overall, our study underscores the importance of considering both the RNA and peptide functions of lncRNAs in future research on neural development and neurodegeneration.

# Methods

**Reagents and tools table**

| Reagent/resource | Reference or source | Identifier or catalog number |
|---|---|---|
| **Experimental models** | | |
| Mouse: *Wild-Type C57BL/6 (B6) Mice* | National Laboratory Animal Center | N/A |
| Mouse: *Sertm2* KO | This paper | N/A |
| Mouse: Sertm2-Flag (Sertm2 KI) | This paper | N/A |
| Mouse: *Mnx1::GFP^{Tg/Tg}* | PMID:12176325 | Dr. Hynek Wichterle (Columbia University) |
| Mouse: *Sertm2* KO; Mnx1::GFP (KO; Mnx1::GFP) | This paper | N/A |
| Mouse: Sertm2-Flag KI; Mnx1::GFP (KI; Mnx1::GFP) | This paper | N/A |
| Mouse: *C57BL/6 (B6)* ESC line | This paper | N/A |
| Mouse: *Sertm2 KO* ESC line | This paper | N/A |
| Mouse: Mnx1::GFP ESC line | PMID:12176325 | Dr. Hynek Wichterle (Columbia University) |
| Mouse: Sertm2-Flag; Mnx1::GFP ESC line (KI; Mnx1::GFP) | This paper | N/A |
| Human: *MNX1-tdTomato iPSC line* (*MNX1-tdT*) | PMID: 32615233 | Dr. Hynek Wichterle (Columbia University) |
| Human: *SERTM2* KO; MNX1-tdTomato iPSC line (SERTM2 KO; MNX1-tdT) | This paper | N/A |

| Reagent/resource | Reference or source | Identifier or catalog number |
|---|---|---|
| Human: HEK293T | Sigma-Aldrich | Cat# 12022001 |
| **Recombinant DNA** | | |
| pHRST-A730046J19Rik-ORF-Flag-IRES-eGFP | This paper | N/A |
| pHRST-mutATG-A730046J19Rik-ORF-Flag-IRES-eGFP | This paper | N/A |
| pENTR_D-TOPO | Invitrogen | K2400-20 |
| pENTR/D-TOPO-Sertm2-Flag | This paper | N/A |
| pENTR/D-TOPO-mut-Sertm2-Flag | This paper | N/A |
| pLX301 | Addgene | Plasmid #25895 |
| pLX301-Sertm2-Flag | This paper | N/A |
| pLX301-mutATG-Sertm2-Flag | This paper | N/A |
| **Antibodies** | | |
| Rabbit polyclonal anti-Lhx3 (1:1000) | Abcam | ab14555; RRID:AB_301332 |
| Rabbit polyclonal anti-Foxp1 (1:20000) | Abcam | ab16645; RRID:AB_732428 |
| Goat anti-Foxp1(1:100) | R&D systems | Cat# AF4534, RRID:AB_2107102 |
| Goat polyclonal anti-Isl1 (1:1000) | Neuromics | GT15051 ;RRID:AB_2126323 |
| Rabbit polyclonal anti-Etv4 (1:2000) | Thomas Jessell (Columbia University) | Cat# C115, RRID:AB_2631446 |
| Rabbit anti-Lhx1 (1:20000) | Thomas Jessell (Columbia University) | Cat# CU453, RRID:AB_2827967 |
| Goat polyclonal anti-ChAT (1:100) | Millispore | AB144P #NG1752017; RRID:AB_2079751 |
| Mouse monoclonal anti-NeuN (1:500) | Millipore | Cat# MAB377, RRID:AB_2298772 |
| Rabbit anti-Satb2 (1:1000) | Abcam | Cat# ab92446, RRID:AB_10563678 |
| Rat anti-Bcl11b/Ctip2 (1:1000) | Abcam | Cat# ab18465, RRID:AB_2064130 |
| Mouse anti-COUP-TF2/NR2F2 (1:200) | R&D systems | Cat# PP-H7147-00, RRID:AB_2155627 |
| Mouse monoclonal anti-Neurofilament H, Nonphosphorylated (SMI-32) (1:1000) | BioLegend | 801701 (clone SMI-32P); RRID:AB_2564642 |
| Sheep anti-GFP (1:1000) | AbD Serotec/Bio-Rad | Cat# 4745–1051, RRID:AB_619712 |
| Donkey anti-Sheep IgG (H + L) Cross-Adsorbed Secondary Antibody, Alexa Fluor™ 488 | Thermo Fisher Scientific | Cat # A-11015, RRID:AB_2534082 |
| Donkey Anti-Goat IgG (H + L), Alexa Fluor™ 488 | Thermo Fisher Scientific | Cat# A-11055, RRID: AB_2534102 |
| 488-AffiniPure Donkey Anti-Rat IgG (H + L) | Jackson ImmunoResearch Labs | Cat# 712-005-153, RRID:AB_2340631 |
| Cy3-AffiniPure Donkey AntiGuinea Pig IgG (H + L) | Jackson ImmunoResearch Labs | Cat# 706-165-148, RRID:AB_2340460 |
| Cy3-AffiniPure Donkey Anti-Mouse IgG (H + L) | Jackson ImmunoResearch Labs | Cat# 715-165-150, RRID:AB_2340813 |
| Cy3-AffiniPure Donkey Anti-Rabbit IgG (H + L) | Jackson ImmunoResearch Labs | Cat# 711-165-152, RRID:AB_2307443 |
| Cy5-AffiniPure Donkey AntiGuinea Pig IgG (H + L) | Jackson ImmunoResearch Labs | Cat# 706-175-148, RRID:AB_2340462 |
| Cy5-AffiniPure Donkey Anti-Mouse IgG (H + L) | Jackson ImmunoResearch Labs | Cat# 715-175-150, RRID:AB_2340819 |
| Cy5-AffiniPure Donkey Anti-Rabbit IgG (H + L) | Jackson ImmunoResearch Labs | Cat# 711-175-152, RRID:AB_2340607 |
| Anti-Glyceraldehyde-3-Phosphate Dehydrogenase Antibody, clone 6C5 (GAPDH; 1:1000) | Millipore | Cat# MAB374, RRID:AB_2107445 |
| DYKDDDDK Tag Monoclonal Antibody (FG4R) (Flag; 1:1000) | Thermo Fisher Scientific | Cat# MA1-91878, RRID:AB_1957945 |
| DYKDDDDK tag Polyclonal antibody (Flag Rb; 1:2000) | ProteinTech | Cat# 20543-1-AP, RRID:AB_11232216 |
| IRDye 800CW Goat anti-Rabbit IgG | LI-COR Biosciences | Cat# 926-32211, RRID:AB_621843 |
| IRDye 680RD Goat anti-Mouse IgG | LI-COR Biosciences | Cat# 926-68070, RRID:AB_10956588 |
| **Oligonucleotides and other sequence-based reagents** | | |
| Primer for Genotyping | | Table EV1 |
| Primer for CRISPR-Cas9 | | Table EV1 |
| Primer for construct | | Table EV1 |
| Primer for ISH probe | | Table EV1 |
| Primer for RT-qPCR | | Table EV1 |
| **Chemicals, enzymes, and other reagents** | | |
| Retinoic acid (RA) | Sigma-Aldrich | Cat#R2625, CAS 302-79-4 |
| Smoothened agonist (SAG) | Merck | Cat#566660, CAS 364590-63-6 |
| Y-27632 | Stemgent | Cat# 04-0012-10, CAS 129830-38-2 |
| GSK-3 Inhibitor XVI (CHIR-99021) | Merck | Cat#361559, CAS 252917-06-9 |
| SB431542 | Merck | Cat#616461, CAS 301836-41-9 |
| LDN 193189 | Sigma-Aldrich | Cat#SML0559, CAS 1062368-24-4 |
| γ-Secretase Inhibitor IX (DAPT) | Merck | Cat#565770, CAS 208255-80-5 |
| Human BDNF | Peprotech | Cat#450-02 |
| Rat GDNF | Peprotech | Cat#450-51 |
| Human FGFb | Peprotech | Cat#100-18 |
| L-Ascorbic acid | Sigma-Aldrich | Cat#A4544, CAS 50-81-7 |
| Hexadimethrine bromide (polybrene) | Sigma-Aldrich | Cat#H9268, CAS 28728-55-4 |
| PD98059 | Merck | Cat#513000, CAS 167869-21-8 |
| B27™ Supplement (50X), serum free | Gibco | Cat#17504044 |
| N2 Supplement (100X) | Gibco | Cat#17502048 |

| Reagent/resource | Reference or source | Identifier or catalog number |
|---|---|---|
| mPAGE™ 4-20% Bis-Tris Protein Gels | Millipore | Cat# MP42G10 |
| **Software** | | |
| Adobe Illustrator | Adobe | https://www.adobe.com |
| Adobe Photoshop | Adobe | https://www.adobe.com |
| GraphPad Prism version 10.0 | GraphPad | https://www.graphpad.com |
| Zen Lite | Carl Zeiss | N/A |
| ImageJ | NIH | https://imagej.net/ |
| R and RStudio | The R Foundation | N/A |
| University of California Santa Cruz (UCSC) genome browser | PMID: 36420891 (Nassar et al, 2023) | https://genome.ucsc.edu/ |
| GWIPS-viz | PMID: 29927076 (Kiniry et al, 2018) | https://gwips.ucc.ie |
| PhyloCSF | PMID: 21685081 (Lin et al, 2011) | http://compbio.mit.edu/PhyloCSF |
| Phobius | PMID: 17483518 (Kall et al, 2007) | https://phobius.sbc.su.se |
| SignalP-6.0 | PMID: 34980915 (Teufel et al, 2022) | https://services.healthtech.dtu.dk/service.php?SignalP-6.0 |
| AlphaFold | Jumper et al, 2021; Varadi et al, 2024 | https://alphafold.ebi.ac.uk/ |
| **smFISH and RNAScope** | | |
| Stellaris® FISH Probes | LGC Biosearch | |
| RNAscope® Multiplex Fluorescent Reagent Kit v2 | ACD Bio | Cat. # 323100 |
| RNAscope® Probe- Mm-A730046J19Rik-O1 | ACD Bio | 1136231-C1 |
| RNAscope® Probe- Mm-Etv4-C2 | ACD Bio | 458121-C2 |
| RNAscope® Probe- Mm-Nr2f2-C2 | ACD Bio | 480301-C2 |
| RNAscope® Probe- Mm-Satb2-C3 | ACD Bio | 413261-C3 |
| RNAscope® Probe- Mm-Isl1-C3 | ACD Bio | 451931-C3 |
| Opal™ 520 Reagent Pack | ACD Bio | FP1487001KT |
| Opal™ 570 Reagent Pack | ACD Bio | FP1488001KT |
| Opal™ 690 Reagent Pack | ACD Bio | FP1497001KT |

## Mouse husbandry

All mice experimental procedures were approved and conducted according to the Institutional Animal Care and Use Committee (IACUC) guidelines at Academia Sinica (Protocol number 23-03-1991). *Sertm2* knockout and Sertm2-Flag KI mice were generated in-house. The strains of mice used in this study are summarized in the Reagents and Tools Table. All mice were maintained in conformity with specific pathogen-free (SPF) status. The animal housing environment was strictly controlled and monitored: 12-h light/dark cycle and 19–23 °C temperature with 40–60% humidity. We employed an equal number of male and female mice, matched for age and developmental stage as specified in the figure legends, and found no sex differences in this study.

## Generation of CRISPR/Cas9-mediated *Sertm2* deletion and 3X-Flag epitope-tagged knock-in mice

Both *Sertm2* knockout, *Sertm2* frameshift, and Sertm2-Flag KI mice were generated in the C57BL/6 J background using the CRISPR/Cas9 system. A pair of single-guide RNA (sgRNA) was designed to target full-length *Sertm2* to induce *Sertm2* loss-of-function. To examine endogenous Sertm2 micropeptide in the mouse model, a 3X-Flag tag was introduced at the C-terminus of Sertm2 (corresponding to a conserved sORF within *A730046J19Rik*). The single-strand oligodeoxynucleotide (ssODN) was used for knock-in manipulation of the sORF within *Sertm2* via homology-directed repair (HDR) (Cong et al, 2013).

All designs of sgRNA sequences in this study were conducted using Breaking-Cas (https://bioinfogp.cnb.csic.es/tools/breakingcas/). In addition, we assessed the off-target effects created by CRISPR/Cas9 via various computational tools to increase efficiency and mitigate the impact of unintended editing mutations. With the help of the Transgenic Core Facility (Institute of Molecular Biology, Academia Sinica), these CRISPR-modified mice were generated by delivery of Cas9 and the designed sgRNAs into the fertilized embryos of C57BL/6 mice. We then selected founder mice from the CRISPR/Cas9 strategy, and the manipulated region was verified using PCR and Sanger sequencing. A list of sgRNA targeting sequences, donor sequences, and genotyping primers is available in Table EV1.

## Derivation and maintenance of mouse ESCs

To acquire the *Sertm2^KO* or Sertm2-Flag KI; Mnx1::GFP (KI; Mnx1::GFP) ESC lines, ESCs were directly derived from E3.5 blastocysts of designated mice. We adopted a mouse ESC derivation approach described previously (Tung et al, 2019). The derived blastocysts were cultured in KOSR ES (KES) medium: EmbryoMax DMEM (Millipore) containing Knockout-SR (Invitrogen), 1× EmbryoMax MEM Non-essential Amino Acids (Millipore), 1× Nucleosides (Millipore), 2 mM L-Glutamine (Invitrogen), 1% Penicillin/Streptomycin (Invitrogen), 0.00072% 2-mercaptoethanol (β-ME, Sigma), 0.02% leukemia inhibitory factor (LIF) (Millipore), 50 μM MEK1/MAPK inhibitors PD98059 (Merck), and 3 μM GSK-3β inhibitor CHIR-99021 (Merck). Maintenance of ESCs required co-culture with feeder cells in the ES medium [1× EmbryoMax DMEM (Millipore) supplemented with 15% Hyclone Fetal Bovine Serum (FBS) (GIBCO, 16000-044), 1× EmbryoMax MEM Non-essential Amino Acids, 1× Nucleosides, 2 mM L-Glutamine, 1% Penicillin/Streptomycin, 0.00072% β-ME, and LIF/ESGRO] and subculturing using 0.05% Trypsin-EDTA (Gibco) every three days. All cell lines were maintained at 37 °C in a 5% $CO_2$ cell culture incubator and routinely tested for mycoplasma contamination by genotyping with the respective primers listed in Table EV1.

## Mouse motor neuron differentiation

Stable mouse ESCs were cultured and differentiated into spinal MNs as described previously (Wichterle et al, 2002; Wichterle et al, 2009; Yen et al, 2018). At Day 2 of differentiation, cells were cultured in ADFNK medium supplemented with 1 μM of retinoic acid (RA, Sigma-Aldrich) and 500 nM of smoothened agonist (SAG, Merck) until Day 5 when cells reached the postmitotic state. Afterward, embryoid bodies (EBs) were dissociated by Accutase

(Gibco) and plated onto poly-L-orithine/laminin-coated four-well plates with coverslips. From Day 5 onward, the cells were either left untreated or induced with recombinant GDNF (10–30 ng/mL, Peprotech) in MN culture medium [ADFNK medium containing N2 (Life Technologies) and B27 (Life Technologies)] (Peljto et al, 2010).

## Culture and differentiation of human ESC or iPSC-derived MNs

The human iPSC line MNX1-tdTomato (iPSC MNX1-tdT, SalvaRED) was acquired from Dr. Hynek Wichterle (Garcia-Diaz et al, 2020). The *SERTM2* knockout cell line (*SERTM2^{KO}*) was generated in-house by the CRISPR/Cas9 gene editing system (Ran et al, 2013) in the MNX1-tdT background. In brief, MNX1-tdT cells were nucleofected using a 4D-Nucleofector (Lonza) with targeting sgRNAs (Table EV1). After 24 h of nucleofection, drug selection was performed with puromycin (0.3 µg/mL) for three days. The selected cells were isolated and then subjected to PCR-based genotyping, with primer sequences provided in Table EV1.

Human ESCs (HuES3 MNX1::GFP) (Di Giorgio et al, 2008) or iPSCs (MNX1-tdT [Ctrl] and *SERTM2^{KO}*; MNX1-tdT [KO]) were maintained on vitronectin (Gibco) at 37 °C and 5% CO$_2$ in Essential 8 medium (Life Science) and subcultured using 0.5 mM EDTA (Invitrogen) every 3 days. All cell lines used in this study are shown in the Reagents and Tools Table.

All human cell lines were dissociated with Accutase and differentiated towards MNs according to a method described previously (Maury et al, 2015; Tung et al, 2019). In brief, cells were cultured in N2/B27 medium consisting of a 1:1 mixture of DMEM-F12 and Neurobasal medium containing N2, B27, 1% Penicillin/Streptomycin, 2 mM Glutamax, 0.2 mM β-ME, 0.5 µM ascorbic acid (Sigma-Aldrich), 10 µM Y-27632 (STemGent), 20 µM SB431542 (Merck), 0.1 µM LDN 193189 (Sigma-Aldrich), 3 µM CHIR-99021 (Merck), and 10 ng/mL bFGF (Peprotech). The formation of EBs occurred after 2 days of differentiation. SB431542 and LDN 193189 were supplemented for 4 days, and 100 nM RA and 0.5 µM SAG were added from Day 2 to 16. BDNF (10 ng/mL, Peprotech) was added from Day 7, and 10 µM of DAPT (Calbiochem) was included from Day 9 to Day 16. At Day 11 of differentiation, EBs were dissociated by Accutase and then plated on poly-L-ornithine/laminin-coated four-well plates at a density of $5 \times 10^4$ cells per well. Dissociated MNs were cultured in MN culture medium [CultureOne Supplement medium (Gibco) containing 10 µM Y-27632, 10 ng/mL BDNF, 10 ng/mL GDNF, and 10 µM 5-fluoro-2′-deoxyuridine (FdU)/Uridine (to inhibit proliferating cells, Merck)] from Day 16 until Day 31 when the cultures were analyzed. The medium was replenished every 3–5 days.

Dissociated MNs (human iPSC-derived MNs) were fixed in 4% paraformaldehyde (PFA) for 30 min at 4 °C and rinsed with PBS. Characterization of iPSC-derived MNs was performed using immunostaining, with images acquired using an ImageXpress® Micro XLS High-Content Imaging System (Molecular Devices) or LSM 780 upright confocal microscope (ZEISS) (Chen et al, 2023; Tung et al, 2019). The neurite complexity was assessed using automated image analysis with ilastik (Berg et al, 2019) and ImageJ/Fiji plugins.

## cDNA library construction and sequencing

For low-input stranded RNA-seq, 100 ng of total RNA was harvested from human ESC-derived MNs (HuES3 MNX1::GFP) at the indicated time points (Day 8, Day 16, and Day 31) during MN differentiation, using two, three, and two biological replicates, respectively, for cDNA library construction. cDNA libraries were prepared with a TruSeq Stranded Total RNA Ribo-zero Plus kit and a TruSeq mRNA Stranded Library Prep kit. The seven libraries were sequenced on a Next-seq 500 system (Illumina) using a 75 pair-ended approach. Approximately, 80–100 million reads were generated from each library. The raw sequence reads in FASTQ format were generated for further analysis.

## Stranded RNA-seq analysis

Adapter contamination in the paired-end reads was removed using PEAT (Li et al, 2015), and the trimmed reads were aligned to GRCh38 using STAR aligner (Dobin et al, 2013). Standard GTF-formatted transcript annotation was defined according to GRCh38 (Harrow et al, 2012), which includes many evidence-based lncRNAs. We used this annotation to aid in junction read alignment in STAR, the output of which was submitted to Cufflinks (Trapnell et al, 2010) for de novo transcript assembly with the option 'library type; first-strand' to allow strand-specific alignments. We calculated the specificity score of each transcript among the samples at different stages using the Jensen–Shannon definition for tissue specificity scores (Cabili et al, 2011; Trapnell et al, 2010; Yen et al, 2018), for which specificity score distributions were plotted and compared.

## Mouse tissue collection

For spinal cord collection from mouse embryos, mice were crossed at the age of 8 ~ 12 weeks to obtain embryos of desired genotypes. When a vaginal plug was observed, the embryos were considered as embryonic day (E) E0.5. At E9.5-E13.5, pregnant mice were euthanized by cervical dislocation, and then embryos were isolated for further study. After removing the head and internal organs, the embryos were immersed in 4% PFA for ~1–2 h (for immunostaining), 4 h (for in situ hybridization), or overnight (for RNAscope) at 4 °C, followed by a PBS wash. Spinal cords were cryoprotected in 30% sucrose and embedded in FSC 22 frozen section media (Leica) before sectioning. Cryostat slices of 20 µm (for immunostaining and in situ hybridization) or 12 µm (for RNAscope) were prepared using a CM3050S Cryosectioning System (Leica). Samples were stored at −80 °C prior to downstream processing.

Neonatal mice (less than postnatal day (P) 10 in age) were anesthetized using hypothermia and then transcardially perfused with cold PBS, followed by 4% PFA post-fixation overnight. The same procedures described above for cryosectioning were used for neonatal samples.

## Immunohistochemistry

Immunohistochemistry of mouse embryos was performed on 20-µm cryostat sections as described previously (Li et al, 2017; Liau et al, 2023). During experimental procedures, sections were

permeabilized and blocked with 10% FBS plus 0.5% Triton X-100 for 1 h at room temperature. Primary antibodies were applied and incubated at 4 °C overnight. Subsequently, sections were washed frequently with 0.01% Triton X-100/PBS. Species-specific Alexa Fluor secondary antibodies (Jackson Immunoresearch) were applied at room temperature for 2 h of incubation. Finally, samples were washed with PBS and mounted with Aqua-Poly/Mount (Polysciences Inc.).

Spinal cord sections from neonatal mice were permeabilized in 0.3% Triton X-100/PBS for 1 h and blocked in 3% bovine serum albumin (BSA) in 0.3% Triton X-100/PBS for 30 min at room temperature. The sections were then incubated with indicated primary antibodies in the blocking solution at 4 °C over two nights. After PBS washes, the sections were immersed with secondary antibodies and DAPI in the permeabilized buffer for 1 h at room temperature. After further PBS washes, the sections were mounted.

Commercially available primary antibodies and antibodies gifted by H. Wichterle and TM Jessell used at respective titers are described in the Reagents and Tools Table. All immunostaining samples were subsequently imaged with a ZEISS LSM 780 upright confocal microscope.

## In situ hybridization (ISH)

Samples were prepared for in situ hybridization as described previously (Hsu et al, 2024). In brief, samples were post-fixed with 4% PFA for 15 min and rinsed with 1× PBS at room temperature. Proteinase K (2 µg/mL, Roche) treatment was applied for 10 min, followed by acetylation in acetic anhydride/triethanolamine (Sigma-Aldrich) for 10 min. After pre-hybridization in hybridization solution [50% formamide, 5× saline-sodium citrate (SSC), 5× Denhardt's solution (Fisher), 250 µg/mL yeast tRNA (Ambion), 0.5 mg/mL salmon sperm DNA (Thermo Fisher Scientific), 2% Roche blocking reagent] for at least 3 h, a digoxigenin (DIG)-labeled RNA probe (100–150 ng) was hybridized into sections overnight at 58 °C. Next, the slides were frequently washed in 2× SSC and then in 0.2× SSC at 55 °C. After blocking with blocking solution [1% BMB (1,4-bismaleimidobutane) and 10% fetal bovine serum in maleic acid buffer (MAB; 100 mM maleic acid, 150 mM sodium chloride)], the sections were incubated with anti-digoxigenin-AP, Fab fragments overnight at 4 °C. Finally, the slides were processed with NBT/BCIP solution (Roche), allowing a color reaction. The sequence for ISH riboprobe generation is provided in Table EV1 and the template DNA was amplified by PCR. Images were captured from projections of a Zeiss AxioImager Z1 upright microscope.

## Single-molecule RNA fluorescence in situ hybridization (smFISH)

Mouse ESC-derived MNs were cultured on poly-L-ornithine/laminin-coated coverslips in MN culture medium. Cells were harvested on the designated day and fixed in 3.7% formaldehyde in DEPC-treated PBS for 10 min at room temperature, permeabilized for 5 min on ice in PBS with 0.5% Triton X-100, followed by immersing in 70% EtOH for subsequent RNA FISH (Yen et al, 2018). Samples were washed in wash buffer [10% deionized formamide in 2× SSC] for 5 min and then incubated with a 1:100 mouse *A730046J19Rik* smFISH probe (Stellaris®, see Reagents and

Tools Table) in hybridization buffer [10% deionized formamide, 0.1 g/mL dextran sulfate (Millipore) in 2X SSC] in the dark within a humidity chamber at 37 °C for 4–16 h. The sections were mounted with VECTASHIELD Antifade Mounting Medium (VECTOR) before undergoing imaging with a Delta Vision microscopy system.

For RNA quantification, we extracted reliable information from the images by generating maximum intensity projections of the z-stacks using ImageJ. Following the guidelines outlined in a previous study (Patel et al, 2021), we quantified the number of spots in different cellular regions and calculated their enrichment across various fractions.

## RNAscope

RNAscope was performed on 12-µm spinal cord sections or mouse ESC-derived MNs using a RNAscope Multiplex Fluorescent Reagent Kit v2 (Advanced Cell Diagnostics, Cat. No. 323100), and probes were detected with Akoya Biosciences Opal 520 (Cat. No. FP1487001KT), 570 (FP1488001KT), and 690 (FP1497001KT). Sample preparation and staining procedures were performed according to the manufacturer's guidelines. In brief, RNAscope probes were hybridized for 2 h at 40 °C. Next, signal amplification and channel development were conducted sequentially, before counterstaining with DAPI and mounting with VECTASHIELD Antifade Mounting Medium (VECTOR). Further details of the probes are presented in the Reagents and Tools Table. Samples were examined using a ZEISS LSM 780 upright confocal microscope.

## Innervation analysis

Experimental procedures to visualize axon arborization patterns in mouse embryos were performed as described previously (Liau et al, 2018; Yen et al, 2018). The *Sertm2* mutant mice were outcrossed with transgenic Mnx1::GFP mice to investigate the possible defects in motor axon arborization. The embryos (E12.5) were fixed in 4% PFA for 1 h at 4 °C, permeabilized in 0.5% PBS-Triton X-100 (PBST), blocked with 10% FBS, and then processed for whole-mount GFP staining at 4 °C with constant shaking. After several washes, embryos were cut in half along the spinal cord. Finally, the preparations were incubated in RapiClear® (RI = 1.49; SUNJIN LAB) to render the embryo transparent for imaging. Maximum intensity projections of z-stack images were collected on Zeiss LSM980 with Airyscan confocal microscopes. Motor innervation patterns of the cutaneous maximus (CM) and latissimus dorsi (LD) muscles of each genotype were processed and quantified using Imaris Microscopy Image Analysis Software (Oxford).

## RNA isolation and quantitative real-time PCR (qPCR)

Trizol (Thermo Fisher Scientific) was used to harvest total RNA from samples. We extracted RNA by conventional chloroform phase separation, followed by isopropanol precipitation. The resulting white pellet was washed in a cold 70% ethanol solution, air-dried, and resuspended in DEPC-treated double distilled water.

Total RNA (0.5–1.0 µg) from each sample was reverse-transcribed into cDNA using Superscript III (Invitrogen). The product of the reverse transcription reaction was applied to subsequent qPCR reactions, which were conducted in technical

replicates and underwent a melt curve analysis on a LightCycler480 Real-Time PCR instrument (Roche) using SYBR Green PCR mix (TOOLS, FPT-BB05) for each gene of interest. The geometric mean of the reference gene (*Gapdh* and *GAPDH* for mouse and human, respectively) Ct values were determined and expression levels of the genes of interest were normalized using the $2^{-\Delta\Delta CT}$ formula. Primers for quantitative PCR analysis are listed in Table EV1.

## Subcellular RNA fractionation

Isolation of subcellular fractions was performed as described previously (Gagnon et al, 2014; Yen et al, 2018). In brief, EBs at desired time points were collected and subcellular components were separated. The pellets of dissociated EBs were resuspended in 200 μl hypotonic lysis buffer [10 mM Tris (pH 7.5), 10 mM NaCl, 3 mM MgCl$_2$, 0.3% NP-40, 10% glycerol], incubated on ice for 10 min, and spun at $1000 \times g$ for 5 min. The supernatant was collected as the cytoplasmic fraction, and the remaining pellets were further processed to extract the nuclear/chromatin fraction. Modified Wuarin-Schibler (200 μl, MWS) buffer supplemented with RNase-OUT and DTT were added to pellets, before incubating on ice for 10 min. After centrifuging, the supernatant and pellet were saved as the nuclear and chromatin fractions, respectively. Trizol (Thermo Fisher Scientific) was then used to extract RNA, which was subjected to reverse transcription with hexamer primers. Finally, to assess the qualities of the different subcellular fractions, we determined expression levels by qPCR of *U1* (*RNU1*, a snRNA in the nucleus) and *GAPDH* as markers of the nucleus and cytoplasm, respectively. The relevant primers are listed in Table EV1.

## Western blotting

Cell and tissue lysates were prepared by lysing samples in RIPA lysis buffer [0.1% SDS, 1% NP-40, 0.5% sodium deoxycholate, 5 mM EDTA, 150 mM NaCl, 50 mM Tris-HCl pH 8.0] with freshly added protease inhibitor cocktail (Roche) and phenylmethylsulfonyl fluoride (PMSF). After incubating on ice for 15 min, the lysates were clarified by centrifugation at $12,000 \times g$ for 15 min at 4 °C. Protein concentration was determined using a BCA Protein Assay kit (Thermo Fisher).

Proteins were loaded onto mPAGE™ Bis-Tris Precast SDS-PAGE Gels (Merck) and run in 1× MES SDS running buffer [Tris-Base, MES, SDS, EDTA] at 180 V for ~30 min, followed by 1 h at 100 V in transfer buffer [25 mM Tris-Base, 25 mM Bicine, 10% Methanol]. Next, we subjected the membranes to blocking with 5% silk milk for 1 h at room temperature, before incubating them with primary antibodies at 4 °C overnight. Appropriate IRDye® secondary antibodies (LiCOR) corresponding to the species of the primary antibody were then applied. Membranes were imaged using an Amersham Biosciences Typhoon biomolecular laser scanner (GE Healthcare). The antibodies used in this study are described in the Reagents and Tools Table.

## HEK293T cell culture and transfection

Human embryonic kidney (HEK293T) cells were grown in DMEM (Gibco) supplemented with 2 mM L-Glutamine (Invitrogen), 1% Penicillin/Streptomycin (Invitrogen), and 10% FBS (Gibco) in a 5% CO$_2$ and 37 °C incubator.

HEK293T cells were plated at a density of $0.5–1 \times 10^5$ per well on a four-well plate (with or without coverslips), cultured for the indicated times, and co-transfected with different combinations of plasmids using the commercially available transfection reagent Lipofectamine™ LTX (Invitrogen). Experimental procedures were conducted according to the manufacturer's recommendations. The use of pEGFP-C1 expressing green fluorescent protein (GFP) allowed us to determine transfection success. Cells were harvested 24–48 h after transfection and either lysed in RIPA lysis buffer or fixed with 4% PFA for further experimentation. Plasmids used in this study are detailed in the Reagents and Tools Table.

## Lentivirus preparation and transduction of mESCs-derived MNs

Sertm2 (corresponding to a conserved sORF within *A730046J19Rik*) and the ATG mutant of Sertm2 (ΔATG) were subcloned into pENTR/D-TOPO entry vector. Gateway cloning technology was applied to exchange the target sequence into the Gateway-compatible pLX301 lentiviral plasmid (a gift from David Root; Addgene plasmid # 25895). *Sertm2* KO ESCs were transduced in the presence of 8 μg/mL polybrene. Transduced cells were subsequently subjected to antibiotic selection with puromycin to generate stable cell lines carrying Sertm2 or mutated Sertm2. LV-Sertm2 (LV) and LV-mutATG-Sertm2 (LV-ΔATG) were packaged by the National RNAi Core Facility (Academia Sinica, Taiwan).

## Mouse motor assessment

Animals were assessed for baseline weight and subjected to behavioral tests at different ages. To minimize the impact of the confounding variable of animal handling, mice were transferred to and habituated in the test room before experiments. Littermate controls or age-matched wild-type (C57BL/6) mice served as a reasonable control for experimental comparisons. Treadmill and beam walking tests were conducted on P30 mice, whereas grip strength and rotarod tests were performed on P90 and P120 mice. At least three measurements were taken from each mouse to get an average result for statistical analysis. All behavioral assays were conducted by experimenters who were blinded to the mouse genotypes.

## Rotarod test

Motor coordination and performance were measured using a Rota-Rod (47650 Rota-Rod, Ugo Basile). All mice were subjected to a training course with a constant speed of 4 rpm before the assay. A steadily accelerating rotation speed, from 4 to 40 rpm over 300 s, was applied to measure the latency for mice to fall off the apparatus. The time to fall off or grasp the rod without running was considered the falling latency.

## Grip strength

The Grip Strength Test (Bio-GS3, BIOSEB) determined the gripping strength of the forelimbs of mice. While placed on the grid, mice were pulled back by the tail, keeping the torso horizontal and allowing only the forepaws to attach to the test grid. The maximal grip strength of the mouse was displayed on the screen of

the Bio-GS3 apparatus. This procedure was repeated for at least three consecutive trials, and results were recorded for further analysis. To mitigate concerns that differential body weight could account for any differences, grip strength values were normalized against mouse body weight.

## Beam walking assay

The beam walking test was conducted using a narrow wooden beam 60 cm long and positioned 40 cm above a bench. Prior to testing, mice were trained to transverse a 15-mm beam 3–5 times. Subsequently, mice were subjected to two trials (15 mm and 9 mm rectangular beams) with a cut-off period of 3–5 min per trial and an intertrial interval of 20–30 min. Each trial was video-recorded from a lateral view to capture performance. Manual analysis included measuring and analyzing the number of segments crossed, beam crossing latencies, and slippage events.

## Treadmill locomotion analysis

A TreadScan apparatus (CleverSys, Reston, VA) was used to analyze gait. Mice were placed on a stationary treadmill for acclimation and trained at a speed of 8 cm/s for 5 min before testing. Three test speeds were analyzed (10, 12.5, and 15 cm/s) for each trial, which were recorded at 79 frames/s for 10 s using the TreadScan software. For data analyses, successful trials in which a mouse could maintain the treadmill speed with continuous locomotion for each of the 10-s recordings were selected and further analyzed using TreadScan software. The gait parameters for each limb—including stride, stance, swing, break, and propulsion time—were automatically and unbiasedly calculated, and average values were used for statistical analysis.

The parameters for limb coordination (phase coupling) were also calculated using TreadScan software, and the average values for homologous, homolateral, and diagonal coupling for each hindlimb from all trials were statistically analyzed. The phase coupling parameter is graphically displayed as a circular plot with phase values of 0 or 1 corresponding to perfect synchronization, whereas a phase value of 0.5 represents strict alternation. The mean phase value is indicated by the direction of the vector, and vector length represents the concentration of phase values around the mean (Chang et al, 2021; Crone et al, 2009). Data visualization was performed in R.

## Statistical analysis and graphical representations

GraphPad Prism 10.0 (GraphPad Software) was used to perform all statistical analyses, i.e., $t$ test or one-way ANOVA, as indicated in the figure legends. Quantitative data are presented as mean ± SD (standard deviation) of three or more independent biological replicates. The respective $P$ values from statistical analyses are shown in the figures.

## Graphics

BioRender (https://www.biorender.com) was used to create the schematics shown in Figs. 1A, 5C,E,H, and 6F, and the synopsis for this study.

## Data availability

The datasets and computer code produced in this study are available in the following databases: RNA-seq data: Gene Expression Omnibus GSE275447. The public datasets utilized in this study are listed as follows: RNA-seq data: Gene Expression Omnibus GSE114285 (Yen et al, 2018). Single-cell RNA-sequencing data: Gene Expression Omnibus GSE183759 (Liau et al, 2023).

The source data of this paper are collected in the following database record: biostudies:S-SCDT-10_1038-S44319-025-00400-0.

## Peer review information

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

## Acknowledgements

The authors thank past and present members of the JAC lab for their valuable comments, support, and proofreading. In particular, we appreciate Dr. Ho-Chiang Hsu for his instrumental role in initiating this project, Dr. Ee Shan Liau for acquiring axon arborization in E12.5 mouse embryos, Dr. Ji-Dung Luo for his analysis of publicly available single-cell RNA-seq data, Yi-Ching Su for mouse behavior tests, Gitartha Das for examining the MMC populations, and Chuan-Che Wu for analyzing neurite complexity. The Transgenics Core Facility of the Institute of Molecular Biology (IMB) at Academia Sinica generated the *Sertm2* KO and Sertm2-Flag KI mice. The authors acknowledge the excellent technical assistance from the FACS, Genomics, Bioinformatics, and Imaging core facilities in IMB. We also appreciate the National RNAi Core Facility at Academia Sinica for providing technical support for lentivirus vector packaging and Dr. John O'Brien for reviewing the manuscript. The human MNX1-tdTomato (MNX1-tdT) iPSC line was a kind gift from Prof. Hynek Wichterle

(University of Columbia). We also thank Prof. Jui-Hung Hung (National Yang Ming Chiao Tung University) and his group for analyzing the human ESC–MNs RNA-seq dataset. F-YH is supported by a National Science and Technology Council (NSTC) research fellowship, and Y-PY is supported by a NHRI postdoctoral fellowship (NHRI-EX113-11330NI). This work is funded by the National Health Research Institutes (NHRI-EX113-11330NI), Academia Sinica (AS-GCP-113-L02 & AS-BRPT-113-01), and the National Science and Technology Council (113-2326-B-001-001).

## Author contributions

**Fang-Yu Hsu**: Conceptualization; Formal analysis; Investigation; Visualization; Methodology; Writing—original draft; Writing—review and editing. **Ya-Ping Yen**: Methodology. **Hung-Chi Fan**: Methodology. **Mien Chang**: Methodology. **Jun-An Chen**: Conceptualization; Formal analysis; Supervision; Funding acquisition; Investigation; Methodology; Writing—original draft; Writing—review and editing.

Source data underlying figure panels in this paper may have individual authorship assigned. Where available, figure panel/source data authorship is listed in the following database record: biostudies:S-SCDT-10_1038-S44319-025-00400-0.

## Disclosure and competing interests statement

The authors declare no competing interests.

# Expanded View Figures

**Figure EV1. Features and expression of *A730046J19Rik* in the MMC and LMC of the brachial spinal cord, along with protein-coding potential and peptide validation of human SERTM2.**

(Related to Fig. 1). (**A**) Multiple sequence alignments of *A730046J19Rik* reveal high conservation across four vertebrate species. The 20-way El track represents conserved elements among 17 primates and three other mammals. The conservation tracks were sourced from the UCSC Genome Browser. (**B**) UMAP visualization displaying *A730046J19Rik, Nr2f2, Satb2*, and *Bcl11b* expression in MMC MN subtypes. The dataset is derived from Liau et al, 2023. (**C**) RNAscope-based ISH of *A730046J19Rik, Satb2, and Nr2f2* in the E13.5 B6 brachial spinal cord. Dashed lines outline the spinal cord boundary. Scale bar, 20 μm. (**D**) Scatter plots depicting correlations of gene expression in brachial LMC neurons, with the Pearson correlation coefficient indicated above each plot. (**E**) Mass spectrometry identified one unique peptide specific to human SERTM2 in human tissue. The data was downloaded from Liu et al, 2022 (LncPep). (**F**) Protein-coding potential of human *SERTM2*, as predicted by the PhyloCSF database.

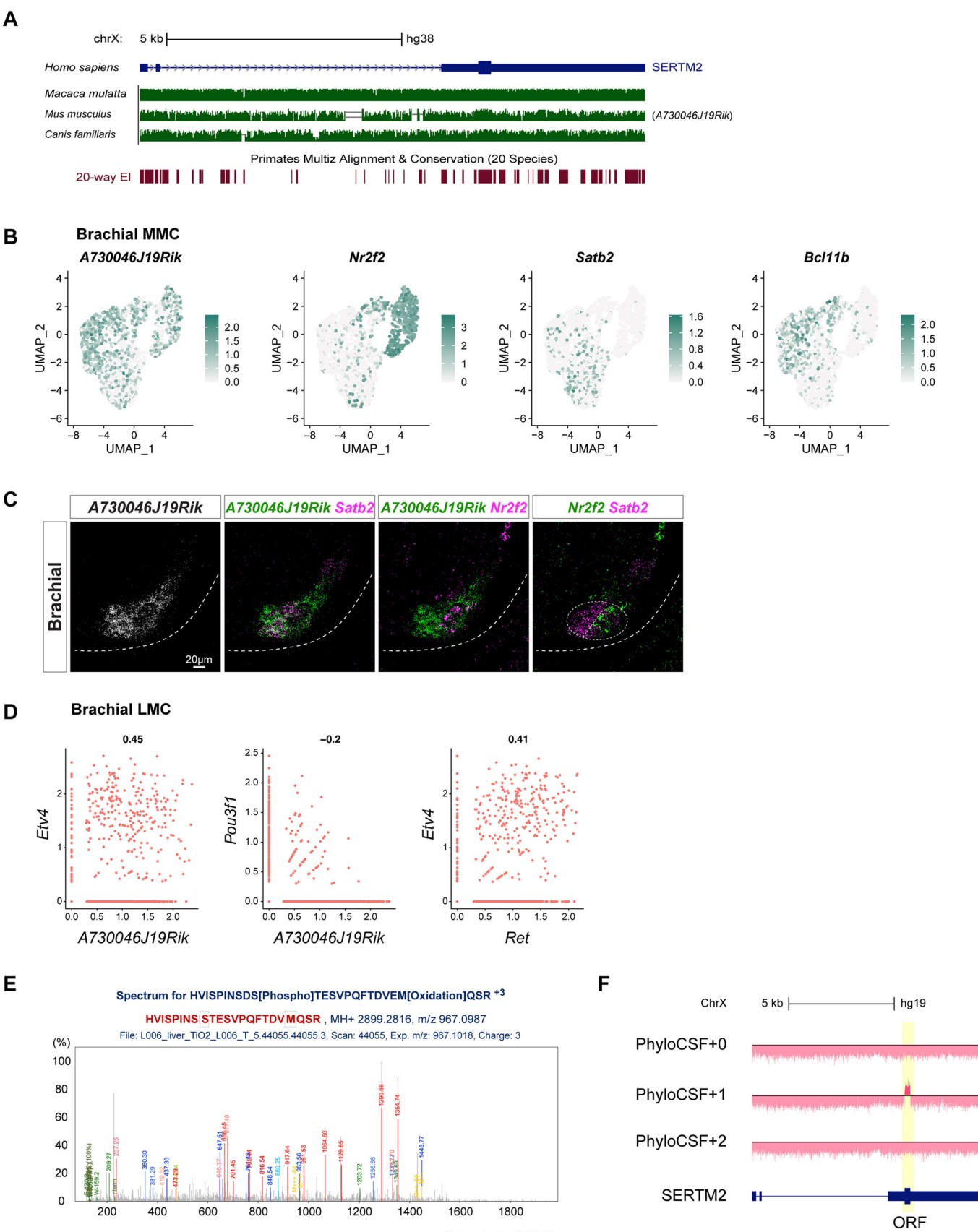

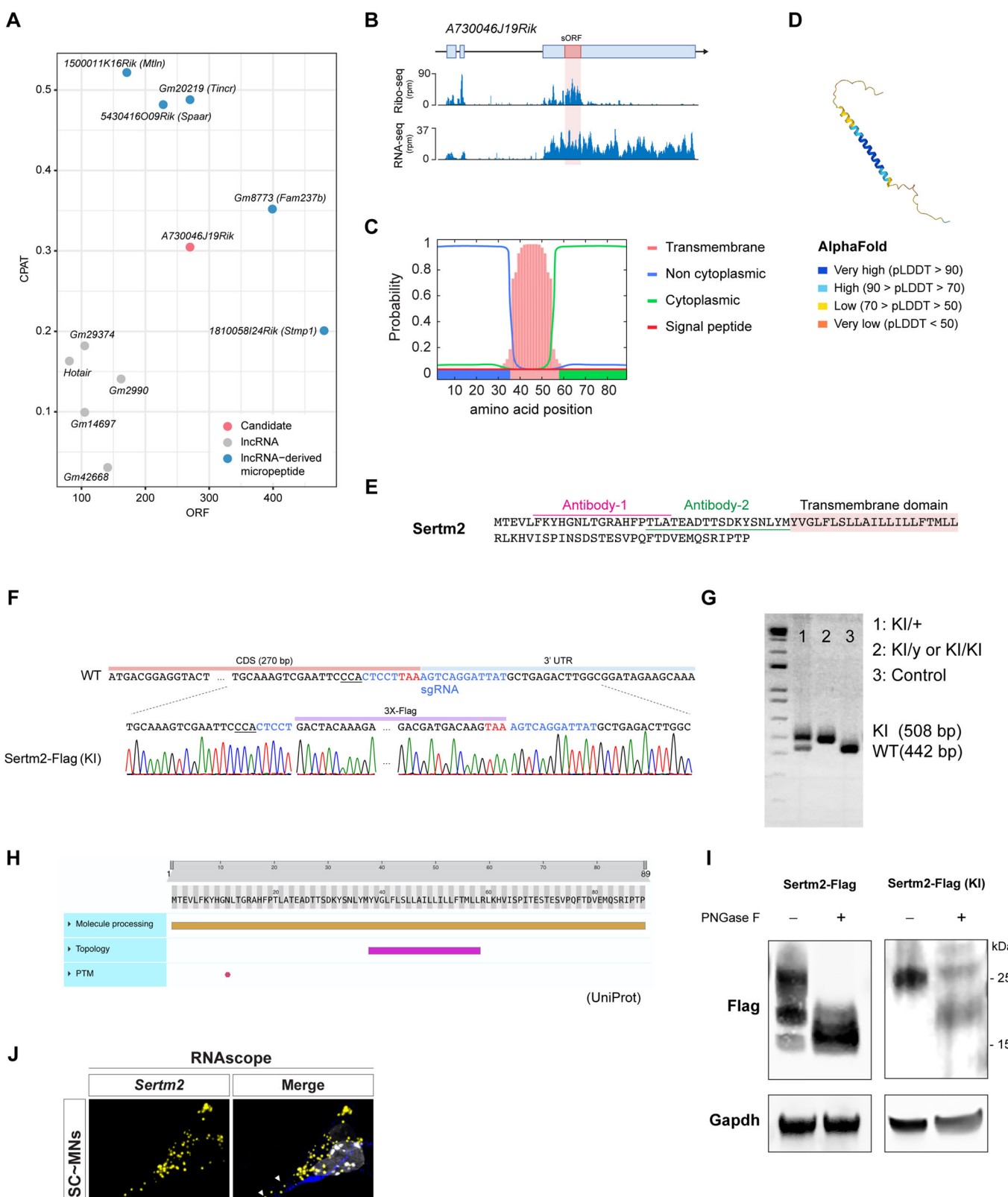

**Figure EV2. Investigation of coding potential and functional analysis of mouse *Sertm2*.**

(Related to Figs. 2 and 3). (**A**) Protein-coding potential of the indicated lncRNAs, as determined in CPAT (Wang et al, 2013). Scatter plot showing the evidence-supported lncRNA-derived micropeptides (blue), lncRNAs with low coding potential (gray), and *A730046J19Rik* (red). (**B**) Ribosome footprint and mRNA fragment densities for *A730046J19Rik* transcripts, as provided by the GWIPS-viz browser. (**C** and **D**) Protein distribution and structure, as predicted by Phobius (**C**) and AlphaFold (**D**), respectively. (**E**) Design of two epitopes for Sertm2-targeting antibodies. (**F**) Validation of CRISPR/Cas9-mediated Sertm2-Flag KI mice by Sanger sequencing. (**G**) Genotyping of Sertm2-Flag KI mice. (**H**) The UniProt database identifies a glycosylation site on the N-terminal region of Sertm2, represented by a pink hexagon. The pink rectangle indicates the transmembrane domain within Sertm2. (**I**) Sertm2 undergoes extensive N-glycosylation, as demonstrated by PNGase F treatment, which shifts its SDS-PAGE migration from 15–25 kDa to approximately 15 kDa. Sertm2-Flag from in vitro overexpression in HEK293T cells (left) and Sertm2-Flag KI from endogenous expression in the spinal cord (right), with both representing evidence of post-translational modifications (PTMs) of the Sertm2 protein. Flag indicates Sertm2 protein, and Gapdh serves as a loading control. (**J**) RNAscope-based ISH reveals the distribution of *Sertm2* in the axon during ESC~MNs. Motor axons are identified by endogenous Mnx1::GFP, while nuclei are highlighted by DAPI staining. White arrows show the localization of *Sertm2* within motor axons. Scale bar: 5 μm.

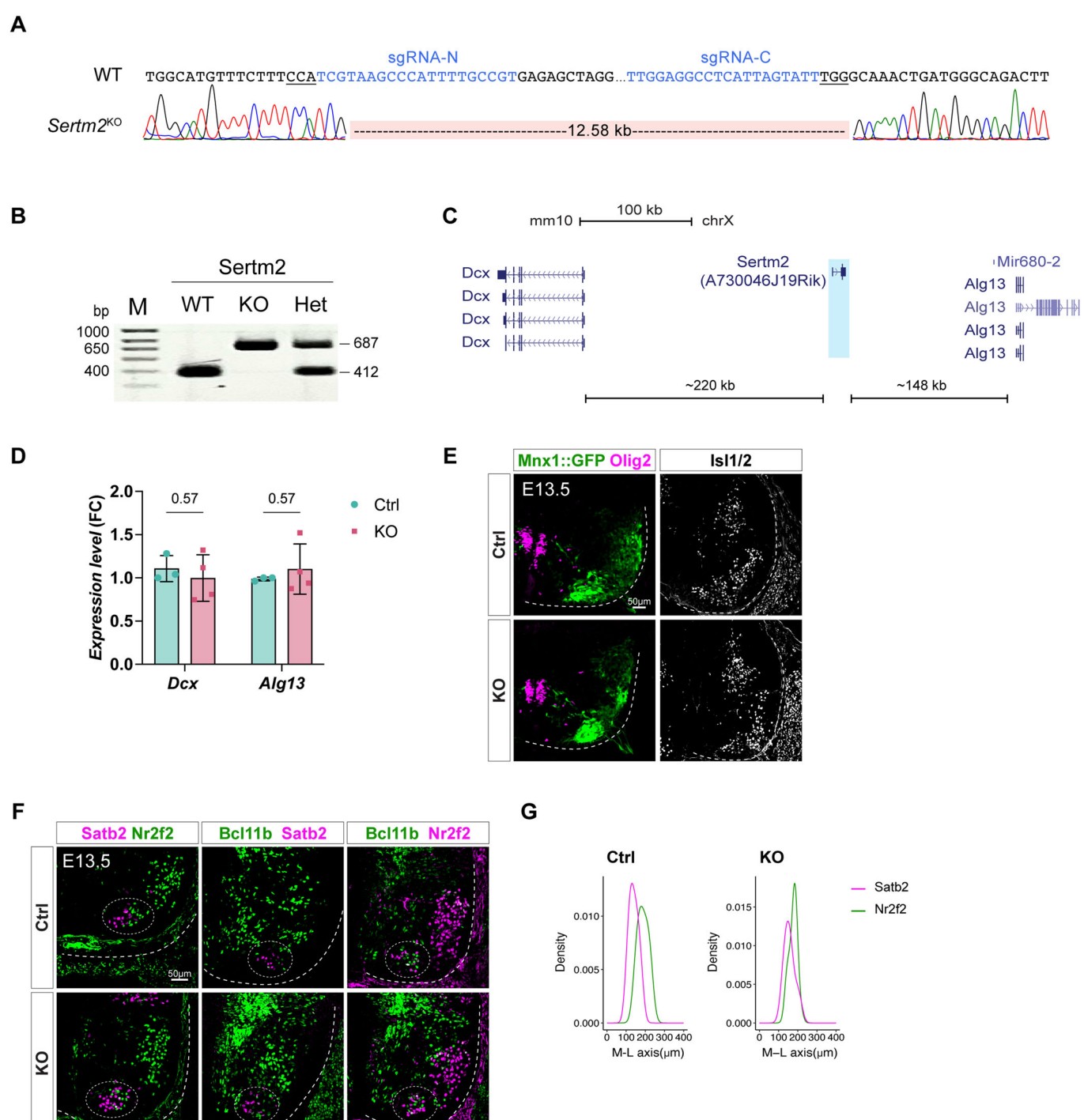

**Figure EV3. Analysis of *Sertm2* depletion effects on neighboring gene expression and the MMC population in the spinal cord.**

(Related to Fig. 4). (**A**) Accurate CRISPR/Cas9 genome editing of the *Sertm2* locus in ESCs, as confirmed by Sanger sequencing. The design of sgRNAs is highlighted in blue. (**B**) Genotyping of the *Sertm2* knockout cell and mouse lines. (**C**) Graphical representation of the mouse *Sertm2* locus and its neighboring genes, *Dcx* and *Alg13* (mm10). (**D**) Depletion of *Sertm2* does not influence gene expression in *cis*. Data from *n* = 3–4 independent experiments; unpaired two-tailed *t* test. *P* values for *Dcx* and *Alg13* (Ctrl vs. KO) were 0.57 and 0.57. (**E**) Immunodetection of Olig2, Mnx1::GFP, and Isl1/2 in E13.5 brachial spinal cord of Ctrl and KO mice. Dashed lines outline the spinal cord. Scale bar: 50 μm. (**F**) Immunostaining of the MMC subtype markers Bcl11b, Satb2, and Nr2f2 in E13.5 brachial spinal cord. Dashed lines outline the spinal cord boundary, and dashed circles demarcate MMC MNs. Scale bar: 50 μm. (**G**) Mediolateral (M–L) density plot of the Satb2⁺ and Nr2f2⁺ subtypes in the E13.5 brachial spinal cord of Ctrl and *Sertm2* mutant mice (KO) in (**F**). Data from *n* = 4 independent biological samples.

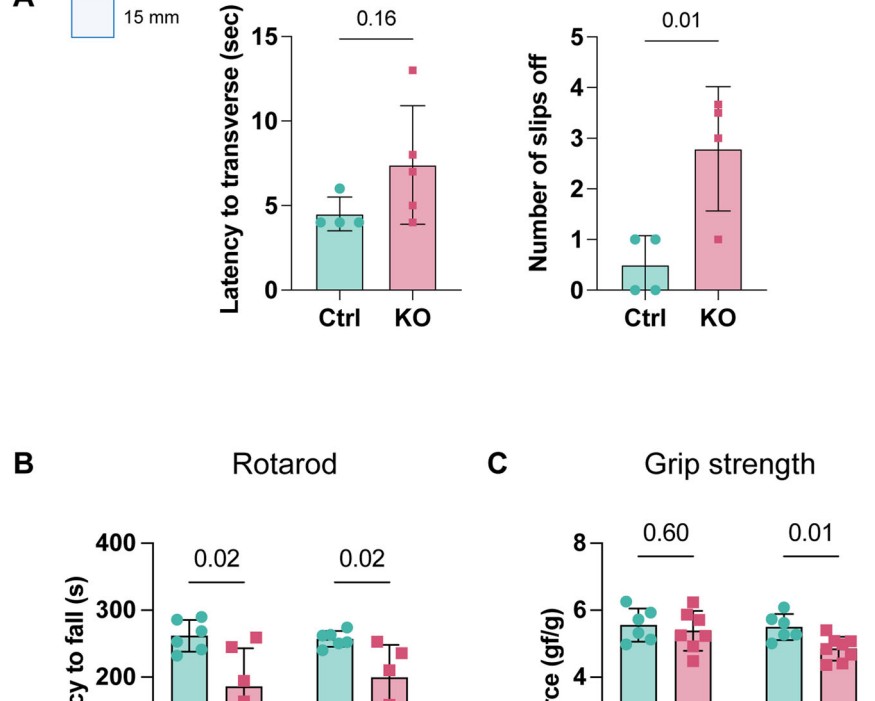

**Figure EV4. Analysis of motor behavior in age-matched Ctrl and *Sertm2* mutant mice.**

(Related to Fig. 5). (**A**) Time to transverse the beam (left) and the number of foot slips (right) on a 15-mm beam for Ctrl and *Sertm2* mutant (KO) mice. Data represent mean ± SD from $n$ = 4–5 independent biological samples; unpaired two-tailed $t$ test. $P$ = 0.16 (Latency to transverse) and $P$ = 0.01 (number of slips off) for Ctrl vs. KO. (**B**) Postnatal day (P) 90 and P120 Ctrl and KO mice were subjected to an accelerating rotarod test to measure motor performance. Latency to fall is shown. Data represent mean ± SD from $n$ = 5–6 independent biological samples; unpaired two-tailed $t$ test. $P$ values for P90 and P120 (Ctrl vs. KO) were 0.02 and 0.02. (**C**) Motor strength was assayed according to grip strength of P90 and P120 Ctrl and KO mice. Grip strength measured in grams is displayed. Data represent mean ± SD from $n$ = 6–7 independent biological samples; unpaired two-tailed $t$ test. $P$ values for P90 and P120 (Ctrl vs. KO) were 0.60 and 0.01.

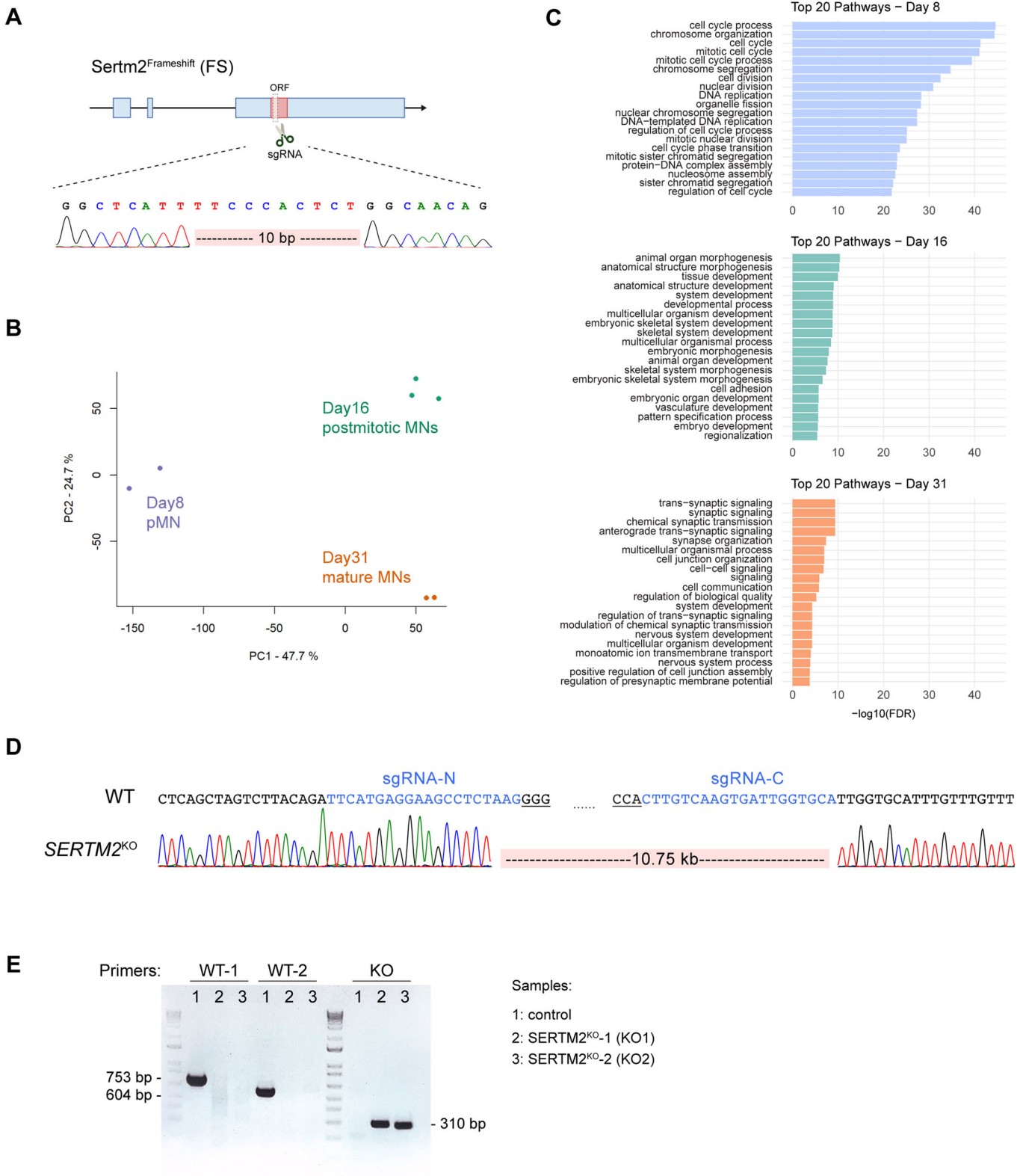

◀ **Figure EV5.  Verification of Sertm2-FS ESC lines and characterization of SERTM2 in a human context, and verification of *SERTM2* knockout in a human iPSC line.**

(Related to Figs. 6, 7). (**A**) Sanger sequencing confirmed the frameshift mutation in mouse Sertm2-FS ESCs. (**B**) The PCA plot of RNA-seq data from human ESC-derived MNs (MNX1::GFP). Samples represent specific time points during MN differentiation: Day 8 ($n = 2$), Day 16 ($n = 3$), and Day 31 ($n = 2$). (**C**) GO analysis was performed for the differentially expressed genes (DEGs) associated across stages of MN differentiation. The criteria for identifying DEGs are as follows: log2 fold change $\geq$ 1 and a false-discovery rate (FDR) < 0.05. (**D**) Sanger sequencing of the CRISPR/Cas9-mediated human *SERTM2* knockout human iPSC line (MNX1-tdT). The design of sgRNAs is highlighted in blue. (**E**) Genotyping of the human *SERTM2* KO line was performed using three distinct primer sets, as illustrated in Fig. 7F.

