## [Peer Review File · EMBO Reports]

Sertm2 is a Conserved Micropeptide that Promotes GDNF-mediated Motor Neuron Subtype Specification

Fang-Yu Hsu, Ya-Ping Yen, Hung-Chi Fan, Mien Chang, and Jun-An Chen

Corresponding author(s): Jun-An Chen (jac2210@gate.sinica.edu.tw)

Review Timeline:

Submission Date:	22nd Aug 24
Editorial Decision:	24th Oct 24
Revision Received:	14th Dec 24
Editorial Decision:	20th Jan 25
Revision Received:	4th Feb 25
Accepted:	7th Feb 25

Editor: *Esther Schnapp*

Transaction Report:

Dear Dr. Chen,

Thank you for your patience while your manuscript was peer-reviewed at EMBO reports. We have finally received the full set of referee reports that is pasted below.

As you will see, all referees acknowledge that the findings are interesting. In fact, all of them rate novelty, interest and technical quality of the ms "High", which is rare and great !

As expected, the referees also have several suggestions for how the study could be strengthened, and I think all suggestions are good and should be addressed. Please let me know in case you disagree, and we can discuss the exact revision requirements further, also in a video chat, if you like.

I would thus like to invite you to revise your manuscript with the understanding that the referee concerns must be fully addressed and their suggestions taken on board. Please address all referee concerns in a complete point-by-point response. Acceptance of the manuscript will depend on a positive outcome of a second round of review. It is EMBO reports policy to allow a single round of major revision only and acceptance or rejection of the manuscript will therefore depend on the completeness of your responses included in the next, final version of the manuscript.

We realize that it is difficult to revise to a specific deadline. In the interest of protecting the conceptual advance provided by the work, we recommend a revision within 3 months (24th Jan 2025). Please discuss the revision progress ahead of this time with the editor if you require more time to complete the revisions.

- 1) A data availability section providing access to data deposited in public databases is missing. If you have not deposited any data, please add a sentence to the data availability section that explains that.
- 2) Your manuscript contains statistics and error bars based on $n=2$. Please use scatter blots in these cases. No statistics should be calculated if $n=2$.

5) a complete author checklist, which you can download from our author guidelines . Please insert information in the checklist that is also reflected in the manuscript. The completed author checklist will also be part of the RPF.

6) Please note that all corresponding authors are required to supply an ORCID ID for their name upon submission of a revised manuscript (). Please find instructions on how to link your ORCID ID to your account in our manuscript tracking system in our Author guidelines

- the name of the statistical test used to generate error bars and P values,
- the number (n) of independent experiments (please specify technical or biological replicates) underlying each data point,
- the nature of the bars and error bars (s.d., s.e.m.),
- If the data are obtained from n {less than or equal to} 2, use scatter blots showing the individual data points.

12) All Materials and Methods need to be described in the main text using our 'Structured Methods' format, which is required for all research articles. According to this format, the Methods section includes a separate file Reagents and Tools Table (listing key reagents, experimental models, software and relevant equipment and including their sources and relevant identifiers) and a Methods and Protocols section describing the methods using a step-by-step protocol format. The aim is to facilitate adoption of the methodologies across labs. More information on how to adhere to this format as well as a downloadable template (.docx) for the Reagents and Tools Table can be found in our author guidelines:

An example of a Method paper with Structured Methods can be found here: <https://www.embopress.org/doi/full/10.1038/s44320-024-00037-6#sec-4>.

You are able to opt out of this by letting the editorial office know (emboreports@embo.org). If you do opt out, the Review Process File link will point to the following statement: "No Review Process File is available with this article, as the authors have

chosen not to make the review process public in this case."

I look forward to seeing a revised form of your manuscript when it is ready.

Yours sincerely,

Referee #1:

In the manuscript, "Sertm2 is a Conserved Micropeptide that Modulates GDNF-mediated Motor Neuron Subtype Specification," Hsu and colleagues present compelling evidence that the lncRNA A730046J19Rik encodes a conserved micropeptide Sertm2, which is relevant for spinal motor neuron development by modulating GDNF signaling. The loss of Sertm2 disrupts motor pool specification in both mouse and human motor neurons, causing motor deficits that are reversible with Sertm2 protein reintroduction.

Overall, the study is thorough and the results are robust, and I do not have any major experiments to suggest. However, based on the provided evidence, I found the classification of A730046J19Rik as a lncRNA-derived microprotein problematic. The manuscript suggests dual functions for this RNA, but no supporting data or citations establish any non-coding roles for A730046J19Rik. Given its high conservation, protein-coding potential, cytoplasmic localization, axonal shuttling, and the presence of AU-rich elements, it appears more likely to be a protein-coding RNA that has been misclassified in the past. To prevent confusion in the field, I suggest the authors clearly textually address whether this is indeed a misclassified protein-coding RNA or if there is support for dual functionality.

1. The authors should discuss the classification of A730046J19Rik as a lncRNA versus a protein-coding RNA. If there are specific data or literature supporting any non-coding roles, it should be included. Otherwise, a clearer statement is needed on why it might be more appropriate to classify this gene as a protein-coding RNA. This is also critical in light that current annotation labels this gene as protein coding.
2. Some of the very compelling data supporting the protein-coding nature of A730046J19Rik are presented later in the manuscript (e.g., Figures 5 and 6). I recommend reordering the results to highlight these findings earlier.
3. Does the detected protein size correspond to predicted protein size? Can the authors comment on this?
4. smRNA FISH shows beautiful redistribution of the mRNA to the axons. Together with the AU-rich elements, that is very strong evidence this is a pcRNA. I would highlight this further and perhaps consider mentioning this earlier in the manuscript.
5. In figure 6D, GDNF 30 ng/ml, it looks like one outlined nucleus in the middle cell has very high levels of the RNA. Could this actually be the cytoplasm?
6. Figures 5C and 5D: the smRNA FISH indicates that the RNA is predominantly cytoplasmic.
7. lncRNA A730046J19Rik was shown previously to be specifically upregulated in mES-derived motor neurons by Biscarini and colleagues in 2018. This should be cited. Some other work relevant to the topic wasn't cited in the context of lncRNA-derived proteins, such as MALAT1 or TUG1.
8. To make the article easily accessible and avoid confusions in the literature, I suggest naming the frequent synonyms used for this gene.

Referee #2:

Fang-Yu et al disrobe the discovery of Sertm2, a novel micro protein encoded in a lncRNA. They show that it is expressed in motor neurons, and is important for motor neuron function. They use CRISPR KO and KI experiments to demonstrate that this micro protein is indeed expressed in human and mouse cells and is functional in vivo.

New microproteins and their function is a very interesting topic as they add new proteins to the catalog of known genes and they can teach us about how biology fine-tunes physiological functions.

The manuscript is well-written and a pleasure to read.

Here are my minor comments:

Figure 4 - RNAseq is missing details - how many replicates per condition, what does the PCA look like, and which GO categories are enriched in the differential genes.

Figure 4e - statistics are missing.

Figure 4 f - one CRISPR clone was used.

Referee #3:

In this study, Hsu and colleagues elegantly demonstrate the role of an lncRNA-derived micropeptide Sertm2 in LMC ETV4+ MN differentiation and function during development. Mechanistically, they further show that peripheral, muscle-released GDNF induces the expression of ETV4 in a Sertm2-dependent manner in those specific MN pools contributing to their specification. Not only has this study exposed the relevance of investigating the potential functions of lncRNAs and their encoded micropeptides in neural development, but also interestingly discusses how lncRNA-derived micropeptides might be linked to neurodegenerative diseases and potentially also serve as biomarkers.

I enjoyed reading this study and have a few minor comments to improve the clarity of the message.

- Probably the most important of them is: is the SERTM2 micropeptide expressed in human MNs? Unless I missed it because of the incomplete figure legend descriptions, all data shown in human cells seems to be transcript expression levels, but no protein. Since the authors seem to be proficient at generating genome edited human iPSC/ESC lines, FLAG-tagging one of these lines to try to detect the protein in human MNs would increase the relevance of SERTM2's role in human MN specification and function. Any alternative approach to prove it would also suffice.

- Often times the text does not explicitly state the approach/technique used to collect the results shown in a certain graph, nor does the figure legend. For example, it should be specified that whatever type of RNAseq was used in fig 1B. By the way, the authors claim that that panel also shows that A730046J19Rik is "also highly conserved across mammals (Figure 1B)", however, I don't see how that information is revealed in that panel.

Along the same lines, to improve clarity, the Fig 1I legend should indicate what clusters are indicated in the violin plots (X axis), especially bc those clusters are not indicated in the UMAPs either.

Similarly, it is not clear how the authors determine SERTM2 cytoplasmic/nuclear expression in Fig. 4E, and therefore it's not clear whether they are referring to mRNA or the micropeptide, which is key information. One has to go to the M&M section to infer that it was done by qPCR upon cytoplasmic/nuclear RNA fractionation. The fig legend again omits this.

Generally, the figure legends are missing important information to fully explain what the figures show. Please, revise.

- The authors show that A730046J19Rik is not only expressed in the LMC MNs in mouse spinal cord, but also in some MMC populations (Satb2+, NR2f2+). While I understand that this study specifically focuses on the role of this lncRNA in LMC ETV4+ MNs, I would have welcomed some discussion on what role this lncRNA might be exerting in MMC MNs.

- The authors seem to attribute the defective motor axon arborization in A730046J19Rik KO mice to MN degeneration (line 191). However, this could just be a developmental defect. The fig legend (3A-B) again does not specify the age of the mice studied, but to claim a degenerative phenotype, the authors should perform the same study across multiple times (prenatal, neonatal and different postnatal stages) and show that degeneration develops over time. Otherwise, the degeneration claim is only an unsubstantiated assumption.

- It is unclear why the authors need to generate a HB9:TdT hiPSC reporter SERTM2 KO line when they already had a hESC SERTM2 KO line (Fig. 4F- L?). From that point on, the text seems to go back-and-forth between both human lines and a second layer of confusion is added when they refer to the KO ESC-derived MNs, as it's unclear whether they are referring to the human or the mouse embryonic line. Again, not specified in the figure legends. Please, revise the text in this regard, since, especially for this study, it is critical that the reader understands whether the text and referred figure pertains to human or mouse in vitro neurons and to transcripts or protein.

- Based on the results shown in Fig 4 on human iPSC/ESC-derived MNs, the authors conclude that SERTM2 expression patterns recapitulate what they've previously shown in mouse embryos. However, the data shown in fig. 4 does not say anything about in which MN pool SERTM2 is expressed in the human spinal cord. Since gaining access to human post mortem samples isn't easy, to make that claim, the authors should check whether recently published sc/snRNAseq studies on human embryonic and adult SC report the expression of this lncRNA and if it's specific to any MN subtype.

- In Fig. 4J-L, to reliably show that human SERTM2 KO MNs have shorter/less neurites, the area occupied by SMI-32+ neurites should be relative to the number of neurons present in the quantified images. Also, is this data collected from the hESC line or the HB9:TdT hiPSC line? Text and fig legend contradictory.

- Fig 5C explores the subcellular distribution of A730046J19Rik in murine MNs, however, what does the area circled by the dotted line indicate?, a nucleus? The legend says "MNs" and that DAPI staining makes it look like it's multiple nuclei. Also not described what the white dots indicate. And what do the 4 points inside the bars in the quantification in 5D represent? The legend should indicate it too.

- According to the text, Figs 2E and 2F seem to be swamped.

Responses to reviewers

We sincerely thank the reviewers for their appreciation of the importance of our study and for their insightful comments. Our point-by-point responses to their comments and descriptions of the new experiments and computational analyses that we have now conducted are provided in this letter. The original reviewers' comments are in **black**, followed by our responses in **blue**. We highlight changes made to the text and figures in **yellow**, whereas **cyan** indicates the paragraph line numbers in the revised text. All changes are highlighted in the revised text. We have made changes to the figure numbering to address the reviewers' comments. To avoid confusion about the numbering, we provide a conversion table here for the reviewers' and editor's reference.

Table R1. Changes to figure numbering and notes for revision.

Old Numbering	New Numbering	Content	Note for changes during revision	In response to
Figure 1	Figure 1	Expression profile of A730046J19Rik during MN development	Removed panel E and changed new panel G	Reviewer 1, 3
Figure 5	Figure 2	In vitro characterization of mouse Sertm2 micropeptide	Changed the layout of panels C and D	Reviewer 1, 3
Figure 6	Figure 3	In vivo validation of mouse Sertm2 micropeptide	Modified panels A, B, F and G	Reviewer 1
Figure 2	Figure 4	Functional analyses of Sertm2 KO in vitro and in vivo	Unchanged	Reviewer 1, 3
Figure 3	Figure 5	Analysis of the impact of Sertm2 deletion on Etv4 -associated behaviors	Changed the layout of panel A (Right)	Reviewer 1, 3
Figure 7	Figure 6	Functional significance of Sertm2 micropeptide in Etv4 -expressing population	Removed panel J	Reviewer 1
Figure 4	Figure 7	Conserved role of human SERTM2	Modified panels E and G-L	Reviewer 1, 2, 3
Figure S1	Figure EV1	Differential expression of A730046J19Rik in MMC and LMC MNs	Reordered the layout	Reviewer 1, 3
Figure S5	Figure EV2	Verification of mouse Sertm2 micropeptide in vitro and in vivo	Added new panels H-J	Reviewer 1

Figure S2	Figure EV3	Characterization of Sertm2 KO mice and frameshift mutant in mouse ESCs	Added panel H	Reviewer 1
Figure S3	Figure EV4	Behavioral assays in Sertm2 mutant mice	Unchanged	Reviewer 1
Figure S4	Figure EV5	1) Evidence for the presence of human SERTM2 micropeptide 2) Investigation of human SERTM2 expression during MN development	Added new panels C and D	Reviewer 1, 2, 3

In addition to the 7 main figures and 5 Expanded View figures shown in the revised manuscript, we have also attached two specific figures herein to address the reviewers' comments (see below). We would be happy to incorporate these additional figures into the revised manuscript if it is felt it is warranted.

Additional Revision Figures:

R1: Western blot analysis using a custom SERTM2 antibody in Day 31 iPSC-derived MNs.

R2: Investigating *SERTM2* expression in human scRNA-seq datasets.

Reviewers' comments:

Referee #1:

In the manuscript, "Sertm2 is a Conserved Micropeptide that Modulates GDNF-mediated Motor Neuron Subtype Specification," Hsu and colleagues present compelling evidence that the lncRNA A730046J19Rik encodes a conserved micropeptide Sertm2, which is relevant for spinal motor neuron development by modulating GDNF signaling. The loss of Sertm2 disrupts motor pool specification in both mouse and human motor neurons, causing motor deficits that are reversible with Sertm2 protein reintroduction.

Overall, the study is thorough and the results are robust, and I do not have any major experiments to suggest. However, based on the provided evidence, I found the classification of A730046J19Rik as a lncRNA-derived microprotein problematic. The manuscript suggests dual functions for this RNA, but no supporting data or citations establish any non-coding roles for A730046J19Rik. Given its high conservation, protein-coding potential, cytoplasmic localization, axonal shuttling, and the presence of AU-rich elements, it appears more likely to be a protein-coding RNA that has been misclassified in the past. To prevent confusion in the field, I suggest the authors clearly textually address whether this is indeed a misclassified protein-coding RNA or if there is support for dual functionality.

We thank the reviewer for his/her encouragements and have addressed each comment in detail below.

1. The authors should discuss the classification of A730046J19Rik as a lncRNA versus a protein-coding RNA. If there are specific data or literature supporting any non-coding roles, it should be included. Otherwise, a clearer statement is needed on why it might be more appropriate to classify this gene as a protein-coding RNA. This is also critical in light that current annotation labels this gene as protein coding.

Response:

We thank the reviewer for raising this query. *A730046J19Rik* is currently annotated in Ensembl databases as a protein-coding gene in mouse (GRCm39) based on its human homolog *LINC00890*, which was originally annotated as a lncRNA, yet this latter has now been reannotated as encoding the micropeptide SERTM2 (serine-rich and transmembrane domain containing 2) based on peptidomics data. However, there is still no literature to verify the presence of the Sertm2 micropeptide *in vivo*. Our study is the first to rigorously demonstrate a functional role for Sertm2 in neural development using genetic knockout and knock-in strategies. Furthermore, we employed micropeptide rescue experiments, widely regarded as the gold standard for validating the

physiological functions of neuropeptides *in vivo*. Our findings establish a role for Sertm2 in spinal MN differentiation and specification, providing compelling evidence for its significance in neural development.

Notably, certain lncRNAs are recognized for their dual roles, functioning both as protein-coding RNAs and as non-coding regulatory RNAs (Nam *et al*, 2016). For instance, the *Tug1* locus demonstrates multifaceted regulatory functions, including acting *in cis* to regulate neighboring genes, *in trans* to influence distant gene expression, and as a peptide that modulates mitochondrial membrane potential, collectively contributing to its critical role in male fertility (Lewandowski *et al*, 2020). Thus, in addition to encoding Sertm2, the *A730046J19Rik* locus may similarly exert diverse regulatory functions. To address this possibility, we have incorporated additional discussion into the revised manuscript to highlight this perspective (Lines 414-427):

“While our study demonstrates that Sertm2 plays a crucial role in the specification of Etv4⁺ MN pools, two intriguing questions remain. First, we cannot entirely exclude the possibility that Sertm2 may exert additional regulatory functions within MNs. Moreover, given that the host lncRNA, *A730046J19Rik*, is also significantly expressed in MN nuclei, it is plausible that *A730046J19Rik* could possess both RNA- and protein-mediated functions (Mattick *et al*, 2023; Wright *et al*, 2022), which remain to be fully explored. Additionally, a previous study showed that the *Tug1* locus exerts three distinct regulatory functions: acting *in cis* to regulate neighboring genes; *in trans* to modulate distant gene expression; and as a peptide to influence mitochondrial membrane potential, all contributing to its essential role in male fertility (Lewandowski *et al.*, 2020). It is tantalizing to probe the full spectrum of functions of the *A730046J19Rik* locus by deleting different regions and examining the physiological consequences. Future studies employing more systematic approaches are necessary to investigate these potential multifaceted roles and further elucidate the broader regulatory functions of Sertm2 and *A730046J19Rik* in MNs.”

2. Some of the very compelling data supporting the protein-coding nature of *A730046J19Rik* are presented later in the manuscript (e.g., Figures 5 and 6). I recommend reordering the results to highlight these findings earlier.

Response:

We sincerely thank the reviewer for this valuable suggestion. After carefully revisiting the manuscript, we agree that reorganizing the layout is a fantastic idea to improve clarity and enhance the narrative. In the revised manuscript, we have restructured the content to provide a more logical and coherent flow. Specifically, we begin by identifying *A730046J19Rik* as a novel locus expressed in spinal motor neurons (MNs) (new Fig. 1). We then establish that *A730046J19Rik* encodes a conserved micropeptide, Sertm2 (new Fig. 2), and confirm its production in mouse spinal cords using a knock-in approach (new Fig. 3). Building on this solid foundation, we

demonstrate that *Sertm2* impairment leads to a significant reduction in *Etv4*⁺ motor neuron pools, accompanied by the erosion of *Etv4*⁺ motor nerves and associated motor deficits (new Figs. 4 and 5). Furthermore, we show that the *Sertm2* micropeptide coding sequence alone (270 bp) is sufficient to rescue *Etv4* downregulation in the full-length *Sertm2* (4.7 kb) knockout ESC-derived motor neurons (new Fig. 6). Finally, we present evidence that the GDNF-SERTM2-ETV4 axis is conserved in humans, demonstrating that human SERTM2 plays a similar role in the development of ETV4⁺ motor neuron subtypes (new Fig. 7).

We believe that this revised structure significantly enhances the clarity and logical progression of our findings, strengthening the case for the critical role of the *Sertm2* micropeptide in spinal motor neuron development.

3. Does the detected protein size correspond to predicted protein size? Can the authors comment on this?

Response:

We thank the reviewer for this insightful observation. The predicted size of the *Sertm2* protein is approximately 10 kDa. For overexpression in HEK293 cells, we included a 1×Flag tag, estimating a size of ~11 kDa, while the 3×Flag tag we used in *Sertm2* knock-in mice resulted in a predicted size of ~13 kDa. However, we observed a ~17 kDa band for *Sertm2*:1×Flag in HEK293 cells and a ~25 kDa band for *Sertm2*:3×Flag in knock-in mice, implying potential post-translational modifications (PTMs). To investigate this possibility, we analyzed UniProt data (new Fig. EV2H) and used prediction tools such as FindMod to identify potential PTMs. Our analysis revealed a predicted glycosylation site in *Sertm2*. Strikingly, after PNGase F treatment to remove glycosylation, the molecular weight of *Sertm2* matched the predicted size (new Fig. EV2I), confirming that *Sertm2* is a glycosylated micropeptide. Furthermore, high-magnification immunostaining revealed that *Sertm2* is predominantly localized in the cytoplasm of ChAT⁺ MNs in the spinal cord at P7 (new Fig. 3E). These findings confirm both the spatial expression pattern of *Sertm2* and its presence in spinal MNs, underscoring its physiological relevance.

We have now added these observations to the Results section (Lines 211-218), and discuss the possible role of glycosylation in the Discussion section (Lines 427-432) of the revised manuscript.

“Interestingly, we found that the detected size of *Sertm2* protein *in vitro* and *in vivo* did not correspond to the predicted size of ~10 kDa (Figs. 2F and 3C), implying potential post-translational modifications (PTMs). To investigate this possibility, we examined the UniProt database (Fig. EV2H) and used prediction tools such as FindMod to identify potential PTMs in *Sertm2*. Our analysis revealed a predicted glycosylation site. Strikingly, by using PNGase F treatments to

deglycosylate the protein, the molecular weight of Sertm2 more closely matched the predicted value (Fig. EV2I), indicating that Sertm2 is a glycosylated micropeptide.”

“Second, *Sertm2* is also expressed in MMC MNs, yet we did not observe a significant impact on MMC MN subtypes (i.e., *Nr2f2* and *Satb2*) in the spinal cords of *Sertm2* KO mice. Given that the protein size of Sertm2 appears to vary in different cell contexts, it is likely that LMCs and MMCs might enforce distinct PTMs, thereby modifying Sertm2 to have differential functions in MN subtypes, representing an uncharted domain for further research.”

4. smRNA FISH shows beautiful redistribution of the mRNA to the axons. Together with the AU-rich elements, that is very strong evidence this is a pcRNA. I would highlight this further and perhaps consider mentioning this earlier in the manuscript.

Response:

We thank the reviewer for this positive feedback on the coding properties of Sertm2 and the thoughtful suggestion. We have now revised and reordered the manuscript to reinforce these findings, emphasizing this point as a key aspect of our study.

5. In figure 6D, GDNF 30 ng/ml, it looks like one outlined nucleus in the middle cell has very high levels of the RNA. Could this actually be the cytoplasm?

Response:

We appreciate the reviewer’s comment and have now clarified the statement in our figure legend (new layout as Fig. 3G). We apologize for the misleading information in the original figure legend. This experiment was actually conducted by immunostaining, not smFISH, and the Flag signal represents the Sertm2 protein. Imaging acquisition was performed across multiple planes along the z-axis and subsequently deconvoluted into a maximum intensity z-projection, which may result in signal overlap. To address this concern, we have merged the original DAPI signal into the revised figure to accurately depict the nucleus. In revised Fig. 3G, the distribution of endogenous Sertm2 protein, as indicated by Flag signal, is enriched in the cytoplasm and distributed along the neurites/axons. The revised sentences are located in the Results section and figure legends.

(Lines 229-236)

“Remarkably, we observed an abundant distribution of the *Sertm2* transcript within the MN axons (Fig. EV2J). Given that *Sertm2* was highly co-expressed with *Etv4* motor pools that are induced by peripheral glial cell line-derived neurotrophic factor (GDNF) signaling from muscles (Haase et al., 2002), we further tested if GDNF can affect Sertm2 distributions. Strikingly, upon addition of

GDNF, levels of the Sertm2 protein were significantly upregulated in the MN axons, indicating that the Sertm2 micropeptide might be produced in response to GDNF signaling (Fig. 3G). Based on these results, we postulate that Sertm2 is a GDNF-induced micropeptide in spinal MNs.”

(Lines 951-954)

“Representative images showing the distribution of Sertm2-Flag protein in the neurites of Ctrl; Mnx1::GFP and KI; Mnx1::GFP mouse ESC-derived MNs, with varying levels of GDNF supplementation. The Flag tag indicates Sertm2 protein expression, with motor neuron axons labeled by endogenous Mnx1::GFP and nuclei highlighted by DAPI staining. Scale bar, 10 μ m.”

6. Figures 5C and 5D: the smRNA FISH indicates that the RNA is predominantly cytoplasmic.

Response:

We employed single-molecule RNA FISH to visualize and quantify endogenous *A730046J19Rik* RNA molecules in fixed cells (new Fig. 2C). Our analysis revealed a prominent cytoplasmic distribution of *A730046J19Rik* RNA. In response to Reviewer #3's concerns regarding this figure, we have altered our presentation of the results in the revised manuscript to enhance clarity and accuracy. First, **to improve visualization of the nuclei**, we now display the DAPI signal as a real image, rendered in red in the merged figure, to avoid potential confusion with the previously used dotted line representation of nuclei. Second, **to clearly identify motor neurons**, we label them with endogenous Mnx1::GFP signal, now depicted in blue for better distinction. Third, **to enhance our depiction of RNA localization**, *A730046J19Rik* RNA molecules are displayed in yellow, highlighting their cytoplasmic enrichment. Fourth, **for quantitative validation**, we have conducted additional independent biological experiments to confirm the subcellular distribution of *A730046J19Rik*, with each dot in the analysis representing an experimental batch (new Fig. 2D). We feel these improvements address the reviewer's concerns, while providing a more robust and precise presentation of the data. The new revised figure legend is presented as follows (Lines 907-913):

“Single-molecule FISH (smFISH) experiments on mouse Mnx1::GFP ESC-derived MNs reveal the localization patterns of *A730046J19Rik* transcripts. A representative image is shown in (C). MNs were identified by endogenous fluorescent Mnx1::GFP signal, while DAPI staining labels the nucleus. Scale bar, 20 μ m. Additionally, quantitative measurements of *A730046J19Rik* expression across different subcellular compartments are shown in (D). Each dot represents one experimental batch, with results presented as mean \pm SD from n = 5 independent experiments; unpaired two-tailed t-test.”

7. lncRNA *A730046J19Rik* was shown previously to be specifically upregulated in mES-derived

motor neurons by Biscarini and colleagues in 2018. This should be cited. Some other work relevant to the topic wasn't cited in the context of lncRNA-derived proteins, such as MALAT1 or TUG1.

Response:

We appreciate the reviewer's suggestion. The relevant studies have now been added to the Introduction, Results, and Discussion sections of the revised manuscript, as follows:

(1) The Results section now includes a reference to Biscarini *et al.* (Stem Cell Research, 2018). (Lines 129-133)

“Among the identified lncRNA candidates, *A730046J19Rik* was prominent as not only does it exhibit MN-specific expression (Fig. 1B, C) (Yen *et al.*, 2018), but it is also highly conserved across mammals (Fig. 2A). In line with our finding, a previous study also found that *A730046J19Rik* is enriched in differentiated MNs (Biscarini *et al.*, 2018).”

(2) The Introduction section now includes a reference to Xiao *et al.* (Genes Dev., 2024). (Lines 66-71)

“A recent study discovered that *MALAT1*, traditionally regarded as a nuclear lncRNA, may be exported into the cytoplasm in differentiating neurons and produce a micropeptide, M1. Synaptic stimulation was shown to enhance M1 expression, indicating that *MALAT1* could function as a cytoplasmic coding RNA that modulates synaptic activity, opening up a new avenue for investigating the roles of micropeptides in neural tissues (Xiao *et al.*, 2024).”

(3) The Discussion section now includes a reference to Lewandowski *et al.* (Genome Biology, 2020). (Lines 420-425)

“Additionally, a previous study showed that the *Tug1* locus exerts three distinct regulatory functions: acting in *cis* to regulate neighboring genes, in *trans* to modulate distant gene expression, and as a peptide to influence mitochondrial membrane potential, all contributing to its essential role in male fertility (Lewandowski *et al.*, 2020). It is tantalizing to probe the full spectrum of functions of the *A730046J19Rik* locus by deleting different regions and examining the physiological consequences.”

8. To make the article easily accessible and avoid confusions in the literature, I suggest naming the frequent synonyms used for this gene.

Response:

In line with the reviewer's suggestion, we agree that a lengthy name can be cumbersome and difficult to remember. To improve clarity and readability, we have replaced the original name (*A730046J19Rik*) with the commonly used synonym (*Sertm2*) in the revised manuscript, following confirmation of its coding potential.

Referee #2:

Fang-Yu et al describe the discovery of *Sertm2*, a novel micro protein encoded in a lncRNA. They show that it is expressed in motor neurons, and is important for motor neuron function. They use CRISPR KO and KI experiments to demonstrate that this micro protein is indeed expressed in human and mouse cells and is functional in vivo.

New microproteins and their function is a very interesting topic as they add new proteins to the catalog of known genes and they can teach us about how biology fine-tunes physiological functions.

The manuscript is well-written and a pleasure to read.

Response:

We thank the reviewer for their encouragement and have addressed each comment in detail below.

Here are my minor comments:

Figure 4 - RNAseq is missing details - how many replicates per condition, what does the PCA look like, and which GO categories are enriched in the differential genes.

Response:

We thank the reviewer for raising this query. To systematically evaluate our RNA-seq data, we have provided further information and assayed the quality of our RNA-seq data in detail (new Fig. EV5C, D). Specifically, we have included samples from three key time-points during hESC-MN differentiation (HuES MNX1::GFP), corresponding to Day 8 (pMNs), Day 16 (postmitotic MNs), and Day 31 (mature MNs). For each stage, we collected two biological replicates on Day 8, three on Day 16, and two on Day 31. We have now detailed the sampling information in the Methods section (Lines 550-557):

“For low-input stranded RNA-seq, 100 ng of total RNA was harvested from human ESC-derived MNs (HuES MNX1::GFP) at the indicated time-points (Day 8, Day 16, and Day 31) during MN

differentiation, using two, three, and two biological replicates, respectively, for cDNA library construction. cDNA libraries were prepared with a TruSeq Stranded Total RNA Ribo-zero Plus kit and TruSeq mRNA Stranded Library Prep kit. The seven libraries were sequenced on a Next-seq 500 system (Illumina) using a 75 pair-ended approach. Approximately, 80~100 million reads were generated from each library. The raw sequence reads in FASTQ format were generated for further analysis.”

Furthermore, we performed PCA to evaluate the characteristics of the samples (new Fig. EV5C). Each dot represents an individual sample. The PCA plot shows a clear separation between the three groups, indicating distinct differences in their gene expression profiles (Lines 1193-1195).

“The PCA plot of RNA-seq data from human ESC-derived MNs (MN_{X1}::GFP). Samples represent specific time-points during MN differentiation: Day 8 (n = 2), Day 16 (n = 3), and Day 31 (n = 2).”

Finally, to enable further downstream analyses, we identified differentially expressed genes across the stages of human ESC~MNs differentiation, allowing us to determine the primary sources of variation among the groups (new Fig. EV5D) (Lines 1197-1199).

“GO analysis was performed for the differentially expressed genes (DEGs) associated across stages of MN differentiation. The criteria for identifying DEGs are as follows: log₂ fold-change ≥ 1 and a false-discovery rate (FDR) < 0.05 .”

Figure 4e - statistics are missing.

Response:

We thank the reviewer for bringing this issue to our attention. To present the results more appropriately, we now display two columns representing the cytoplasmic and nuclear fractions and performed a two-way ANOVA to assess enrichment of the gene of interest. In this scenario, *SERTM2* was found to be enriched more significantly in the cytoplasmic fraction, using *GAPDH* and *UI* as markers for the cytoplasm and nucleus, respectively (new Fig. 7E).

(Lines 1076-1079)

“Subcellular fractionation of iPSC-derived MNs at Day 31 reveals significant *SERTM2* transcripts in the cytoplasm. *GAPDH* and *UI* serve as markers for the cytoplasmic and nuclear fractions, respectively. Data are presented as mean \pm SD from n = 3~4 independent experiments; multiple unpaired t-tests.”

Figure 4f - one CRISPR clone was used.

Response:

In light of this question, we generated a new human SERTM2 CRISPR KO line (KO2) (new Fig. 7G-I), verified the genotype (new Fig. EV5F), and can reveal that *ETV4* expression is compromised in both lines (new Fig. 7I) (Lines 1085-1090).

“qPCR analyses of Ctrl and KO iPSC-derived MNs at Day 31 reveal downregulation of *ETV4* in the absence of *SERTM2*, whereas *FOXP1* expression remains unaffected. Data are presented as fold-change (FC) relative to Ctrl, with mean \pm SD from $n = 3\sim 5$ independent experiments; Ordinary one-way ANOVA. Here, Ctrl refers to the MNX1-tdT iPSC line, whereas KO represents two CRISPR-mediated *SERTM2* mutant iPSCs (KO1 and KO2) used for qPCR analyses.”

Referee #3:

In this study, Hsu and colleagues elegantly demonstrate the role of an lncRNA-derived micropeptide Sertm2 in LMC ETV4+ MN differentiation and function during development. Mechanistically, they further show that peripheral, muscle-released GDNF induces the expression of ETV4 in a Sertm2-dependent manner in those specific MN pools contributing to their specification. Not only has this study exposed the relevance of investigating the potential functions of lncRNAs and their encoded micropeptides in neural development, but also interestingly discusses how lncRNA-derived micropeptides might be linked to neurodegenerative diseases and potentially also serve as biomarkers.

I enjoyed reading this study and have a few minor comments to improve the clarity of the message.

We appreciate the reviewer's encouragement and thank you for pointing out these issues, which have improved the clarity and accuracy of our work. We have addressed each comment in detail below.

- Probably the most important of them is: is the SERTM2 micropeptide expressed in human MNs? Unless I missed it because of the incomplete figure legend descriptions, all data shown in human cells seems to be transcript expression levels, but no protein. Since the authors seem to be proficient at generating genome edited human iPSC/ESC lines, FLAG-tagging one of these lines to try to detect the protein in human MNs would increase the relevance of SERTM2's role in human MN specification and function. Any alternative approach to prove it would also suffice.

Response:

We agree with the reviewer that verifying SERTM2 protein expression in human motor neurons is an important issue. Though mass spectrometry data indicate the presence of SERTM2 in other human tissues (Liu *et al*, 2022), its expression in motor neurons remains uncertain. To address this possibility, we have employed several approaches outlined below:

(1) **SERTM2 antibodies:** As noted in our manuscript, the two custom-made antibodies exhibited poor specificity for detecting endogenous SERTM2 protein. Consequently, we were unable to reliably distinguish signals between the control (Ctrl) and mutant (KO) cell lines when we performed either Western blotting or immunostaining (**Fig. R1**).

(2) **Mass spectrometry:** Confirming the presence of SERTM2 protein in human iPSC-derived motor neurons would be most confidently achieved through mass spectrometry. However, due to its small size and transmembrane nature, extraction and detection procedures for a microprotein likely requires extensive optimization to ensure reliable results, which we feel might be beyond the scope of the current study.

(3) **Epitope Tagging:** In line with the reviewer's suggestion, we employed a CRISPR-mediated HDR knock-in approach in human iPSCs. However, our attempts revealed that human iPSCs often demonstrate low HDR efficiency, which is critical for achieving precise knock-in. Although we are still actively troubleshooting this issue to improve the success rate, it might take a prohibitively long time to acquire the knock-in line and further characterize it.

Thus, although we have attempted several approaches, it remains an unsolved question in our revised manuscript. Nevertheless, we raise the issue in the Discussion as follows (**Line 410-414**):

“Moreover, using Sertm2-Flag to pull down the mass spectrometry-determined Sertm2 interactome in the spinal cord would enable identification of possible candidates responsible for how Sertm2 modulates the GDNF signaling pathway. Doing so would also clarify if Sertm2 is locally induced at neuromuscular junctions or within MN cell bodies, providing deeper insights into its regulatory role in GDNF-mediated MN function.”

- Often times the text does not explicitly state the approach/technique used to collect the results shown in a certain graph, nor does the figure legend. For example, it should be specified that whatever type of RNAseq was used in fig 1B. By the way, the authors claim that that panel also

shows that A730046J19Rik is "also highly conserved across mammals (Figure 1B)", however, I don't see how that information is revealed in that panel.

Along the same lines, to improve clarity, the Fig 1I legend should indicate what clusters are indicated in the violin plots (X axis), especially bc those clusters are not indicated in the UMAPs either.

Similarly, it is not clear how the authors determine SERTM2 cytoplasmic/nuclear expression in Fig. 4E, and therefore it's not clear whether they are referring to mRNA or the micropeptide, which is key information. One has to go to the M&M section to infer that it was done by qPCR upon cytoplasmic/nuclear RNA fractionation. The fig legend again omits this.

Generally, the figure legends are missing important information to fully explain what the figures show. Please, revise.

Response:

We apologize for the missing information in the original manuscript and thank the reviewer for very constructive comments to improve clarity. We have addressed these issues in the revised manuscript as follows:

(1) **Heatmap in Figure 1B:** The heatmap is derived from a strand-specific RNA-seq database reported by Yen *et al.* (*eLife*, 2018). This information has been added into the figure legend of **new Figure 1B**.

(Lines 855-858)

"Heatmap showing the gene expression profiling of protein-coding genes [pink] and lncRNAs [blue] during mouse ESC-derived MN differentiation. The green color gradient indicates the stage specificity score across stages from ESCs to MNs or non-MNs. The strand-specific RNA-seq data analyzed in this figure was sourced from Yen *et al.*, 2018."

(2) **Conservation of A730046J19Rik:** As suggested by reviewer#1, we have reorganized the figure layout to report A730046J19Rik (Sertm2) conservation earlier in the revised manuscript. In this scenario, conservation of A730046J19Rik across species is illustrated in new **Figure 2A**. Additionally, we have corrected the mislabeled annotation in the Results section.

(Lines 129-131)

"Among the identified lncRNA candidates, A730046J19Rik was prominent as not only does it exhibit MN-specific expression (Fig. 1B, C) (Yen *et al.*, 2018), but it is also highly conserved across mammals (Fig. 2A)."

(3) **Clarification of scRNA-seq results:** To enhance clarity, we have revised the statements in the figure legend related to our scRNA-seq data. In addition, as per the reviewer’s suggestion, we have added a new UMAP analysis as **new Figure 1G** (the leftmost panel) to indicate the 16 subclusters in the LMC MNs of the brachial limb region from our previous study (Liau *et al*, 2023), so that the distribution of *A730046J19Rik* in different subclusters in LMC MNs could be compared in UMAP (**new Fig. 1G**) and violin plots (**new Fig. 1H**), respectively. The figure legend of 1G and 1H has been revised as follows:

(Lines 879-890)

“(G) (Left) Uniform Manifold Approximation and Projection (UMAP) plot of 16 LMC subclusters from single-cell RNA-sequencing (scRNA-seq) of *Mnx1::GFP* labeled-MNs in E13.5 brachial spinal cord (Rostral C4-T3 segments). The four shaded colors represent distinct subtypes within limb MNs, including rLMCl (rostral lateral, orange), rLMCm (rostral medial, green), cLMCl (caudal lateral, blue), and cLMCm (caudal medial, pink) regions. (Right) UMAP plots displaying the expression patterns of *A730046J19Rik*, *Etv4 (Pea3)*, *Pou3f1(Scip)*, and *Isl1* in LMC MN subtypes. The green gradient represents the expression levels of the genes of interest. The dataset is derived from Liau *et al*, 2023.

(H) Violin plots reflecting the expression of selected genes (*A730046J19Rik*, *Etv4*, *Pou3f1*, and *Isl1*) within LMC MN clusters. The x-axis represents 16 LMC subclusters, categorized as follows: rl (rostral lateral), rm (rostral medial), cl (caudal lateral), and cm (caudal medial).”

(4) **Subcellular Fractionation of Human SERTM2:** Enrichment of human *SERTM2* in subcellular fractions was determined by qPCR, reflecting RNA levels. This clarification has been added to the figure legend to avoid confusion (**new Figure 7E**).

(Lines 1076-1079)

“Subcellular fractionation of iPSC-derived MNs at Day 31 reveals significant *SERTM2* transcripts in the cytoplasm. *GAPDH* and *UI* serve as markers for the cytoplasmic and nuclear fractions, respectively. Data are presented as mean \pm SD from n = 3-4 independent experiments; multiple unpaired t-tests.”

- The authors show that *A730046J19Rik* is not only expressed in the LMC MNs in mouse spinal cord, but also in some MMC populations (*Satb2*⁺, *NR2f2*⁺). While I understand that this study specifically focuses on the role of this lncRNA in LMC *ETV4*⁺ MNs, I would have welcomed some discussion on what role this lncRNA might be exerting in MMC MNs.

Response:

We thank the reviewer for this insightful comment. While we have shown that *Sertm2* is expressed in both the MMC and LMC, it appears to exert a greater functional impact on LMC ETV4⁺ motor neurons. We have now added a brief discussion of this as follows (Lines 427-432):

“Second, *Sertm2* is also expressed in MMC MNs, yet we did not observe a significant impact on MMC MN subtypes (i.e., *Nr2f2* and *Satb2*) in the spinal cords of *Sertm2* KO mice. Given that the protein size of *Sertm2* appears to vary in different cell contexts, it is likely that LMCs and MMCs might enforce distinct PTMs, thereby modifying *Sertm2* to have differential functions in MN subtypes, representing an uncharted domain for further research.”

- The authors seem to attribute the defective motor axon arborization in A730046J19Rik KO mice to MN degeneration (line 191). However, this could just be a developmental defect. The fig legend (3A-B) again does not specify the age of the mice studied, but to claim a degenerative phenotype, the authors should perform the same study across multiple times (prenatal, neonatal and different postnatal stages) and show that degeneration develops over time. Otherwise, the degeneration claim is only an unsubstantiated assumption.

Response:

We thank the reviewer for raising this important point. We acknowledge that attributing defective motor axon arborization in *Sertm2* KO mice could not exactly reflect MN degeneration, which would require further evidence, particularly a temporal analysis to demonstrate a progressive degenerative phenotype. In our experiment, we only examined motor axon arborization at the E12.5 embryonic stage, so we have revised the figure legend as follows (new Fig 5A, B):

(Lines 992-1000)

“(A and B) Neurite arborization in the latissimus dorsi muscle (A) and cutaneous maximus (CM) muscle (B) in E12.5 Ctrl; *Mnx1::GFP* and *Sertm2* KO; *Mnx1::GFP* (KO; *Mnx1::GFP*) mice, respectively. The yellow dashed squares in (A) mark regions that are enlarged, with axonal reconstructions displayed in the lower panels. The right panel provides quantification of axonal branches in the latissimus dorsi (LD) muscle. Similarly, axons in the CM muscle are reconstructed in the lower panels of (B), with the right panel showing quantification of axon innervation area, normalized to body length. Data represent mean \pm SD from $n = 2$ independent biological samples. Scale bars: 100 μ m.”

Moreover, in the revised manuscript, we have altered our statement to better reflect our finding.

(Lines 274-277)

“We reasoned that impaired specification of the *Etv4*⁺ motor pool and erosion of the associated motor nerves could elicit deficits in motor coordination, particularly affecting delicate, limb-driven behaviors.”

- It is unclear why the authors need to generate a HB9:TdT hiPSC reporter SERTM2 KO line when they already had a hESC SERTM2 KO line (Fig. 4F- L?). From that point on, the text seems to go back-and-forth between both human lines and a second layer of confusion is added when they refer to the KO ESC-derived MNs, as it's unclear whether they are referring to the human or the mouse embryonic line. Again, not specified in the figure legends. Please, revise the text in this regard, since, especially for this study, it is critical that the reader understands whether the text and referred figure pertains to human or mouse in vitro neurons and to transcripts or protein.

Response:

We apologize for any confusion caused by the mislabeled annotations in the original manuscript. To ensure clarity, we have updated the relevant sections of the manuscript, including the Results, Materials and Methods, and Figure Legends. To clarify the KO annotation for humans and mice, we have standardized the Ensembl nomenclature. All capitalized and italicized letters refer to human transcripts, whereas non-italicized capital letters denote human proteins. For example, human SERTM2 RNA is now labeled as *SERTM2* (all capitalized and italicized), whereas the protein is referred to as SERTM2 (all capitalized and non-italicized). Moreover, for consistency with mouse gene annotation, we have updated the gene name in the revised manuscript, renaming the originally annotated HB9 as MNX1. In this study, we utilized human ESC-derived motor neurons (HuES3 MNX1::GFP) to profile transcriptomic dynamics across developmental stages (new Fig. 7C). Human iPSCs (MNX1-tdT) were used to generate the *SERTM2* mutant cell line to perform functional studies (new Fig. 7D-L). We agree that switching between the human and mouse embryonic lines for comparison may cause some confusion. Consequently, in the revised manuscript, we have moved the human study to the last figure (new Fig. 7), with all characterizations therein performed on human cells. Specifically, we have revised the text as follows (Lines 327-332):

“Next, we examined the expression pattern of human *SERTM2* during MN differentiation. To do so, we differentiated a human ESC line harboring the MN reporter (HuES3 MNX1::GFP) into spinal MNs (Fig. 7B) (Di Giorgio *et al*, 2008), and then performed strand-specific RNA sequencing at three stages of differentiation, i.e., motor neuron progenitors (pMNs, Day 8), nascent postmitotic MNs (Day 16), and mature MNs with neurites (Day 31) (Figs. 7C and EV7C, D).”

(Lines 340-343)

“Moreover, to establish if the function of the GDNF-SERTM2-ETV4 axis is conserved in humans, we mutated the entire locus of *SERTM2* in the MNX1-tdT iPSC line (Fig. 7F and EV5E, F), and verified the absence of *SERTM2* expression from the resulting *SERTM2* KO iPSC~MNs (KO) (Fig. 7G).”

- Based on the results shown in Fig 4 on human iPSC/ESC-derived MNs, the authors conclude that *SERTM2* expression patterns recapitulate what they've previously shown in mouse embryos. However, the data shown in fig. 4 does not say anything about in which MN pool *SERTM2* is expressed in the human spinal cord. Since gaining access to human post mortem samples isn't easy, to make that claim, the authors should check whether recently published sc/snRNAseq studies on human embryonic and adult SC report the expression of this lncRNA and if it's specific to any MN subtype.

Response:

We thank the reviewer for these insightful comments. To perform this analysis, we obtained publicly available single-cell RNA-seq (scRNA-seq) datasets to characterize *SERTM2* expression in human spinal cord at various developmental stages. Specifically, we used the following two databases:

(1) **GSE171892:** (Rayon *et al*, 2021)

Rayon *et al.* used scRNA-seq to profile cervical and thoracic regions of the human neural tube across gestational weeks (GW) 4–7, identifying diverse progenitor and neuronal cell types and their differentiation pathways. Comparison with mouse models revealed conserved and human-specific features of neural tube development. The data provide a valuable resource for understanding sensory and motor system development.

(2) **GSE221692:** (Shi *et al*, 2024)

Shi *et al.* explored gene dynamics across time and space throughout human spinal cord development from GW7 to GW25 using TF-seqFISH and complementary spatial transcriptomic methods. A key finding highlights the spatial and molecular mechanisms underlying MN diversification, revealing how transcription factor combinations drive MN specification.

By analyzing these two independent databases, we consistently observed that human *SERTM2* is enriched in MNs within the developing spinal cord (**Fig. R2**). However, the datasets collectively reveal a scarcity of MN subtypes in human embryos, particularly with limited numbers of ETV4⁺ cells (< 23 barcodes found). This limitation makes it challenging to precisely map *SERTM2* expression to specific MN subtypes in human embryos. Consequently, we only present these scRNA-seq findings here for the reviewer's reference. Future efforts to collect larger datasets from human embryos with increased representation of MN subtypes will be essential to address this

question more definitively and provide a clearer understanding of *SERTM2*'s role in MN subtype specification in a human context.

- In Fig. 4J-L, to reliably show that human *SERTM2* KO MNs have shorter/less neurites, the area occupied by SMI-32+ neurites should be relative to the number of neurons present in the quantified images. Also, is this data collected from the hESC line or the HB9:TdT hiPSC line? Text and fig legend contradictory.

Response:

We appreciate the reviewer's comments. This data was collected from the MNX1-TdT iPSC line. We have now corrected the figure legend and text for consistency.

(Lines 340-343)

“Moreover, to establish if the function of the GDNF-*SERTM2*-*ETV4* axis is conserved in humans, we mutated the entire locus of *SERTM2* in the MNX1-tdT iPSC line (Fig. 7F and EV5E, F), and

verified the absence of *SERTM2* expression from the resulting *SERTM2* KO iPSC~MNs (KO) (Fig. 7G).”

Additionally, we analyzed the number of ISL1⁺ cells, together with SMI32 staining (new Fig 7L), to establish that the shorter/fewer neurites in the *SERTM2* KO line is not due to a reduction in MNs.

(Lines 1094-1098)

“(K and L) Validation of MN identity and morphology by immunostaining for ISL1 and SMI-32 at Day 31 of iPSC~MN differentiation (K). SMI-32 highlights neurite formation, while DAPI staining labels the nuclei. Scale bar, 50 μ m. Neurite complexity, quantified using ilastik, was normalized to the number of ISL1⁺ neurons and is shown in panel (L). Data are presented as mean \pm SD from n = 4 independent experiments; Ordinary one-way ANOVA.

- Fig 5C explores the subcellular distribution of A730046J19Rik in murine MNs, however, what does the area circled by the dotted line indicate?, a nucleus? The legend says "MNs" and that DAPI staining makes it look like it's multiple nuclei. Also not described what the white dots indicate. And what do the 4 points inside the bars in the quantification in 5D represent? The legend should indicate it too.

Response:

We appreciate the reviewer for noticing this confusing presentation. For clarity, we now display the DAPI signal as a real staining image in red, motor neurons are identified by the endogenous Mnx1::GFP signal depicted in blue, and the distribution of *A730046J19Rik* RNA in yellow (new Fig 2C).

For our quantification, we analyzed more than three batches of independent biological replicates to determine enrichment of *A730046J19Rik* in the subcellular distribution by means of smFISH, with each dot representing one experimental batch. The corresponding figure legend has also been updated accordingly (new Fig. 2D).

(Lines 907-913)

“(C and D) Single-molecule FISH (smFISH) experiments on mouse Mnx1::GFP ESC-derived MNs reveal the localization patterns of *A730046J19Rik* transcripts. A representative image is shown in (C). MNs were identified by the endogenous fluorescent Mnx1::GFP signal, and the DAPI staining labels the nucleus. Scale bar, 20 μ m. Additionally, quantitative measurements of *A730046J19Rik* expression across different subcellular compartments are shown in (D). Each dot represents one experimental batch, with results presented as mean \pm SD from n = 5 independent experiments; unpaired two-tailed t-test.”

- According to the text, Figs 2E and 2F seem to be swapped.

Response:

Apologies for this error, which has now been corrected. The figures previously labeled as Fig. 2E, F have been renumbered as Fig. 4E, F (Lines 251-254):

“Whereas GDNF significantly induced *Etv4* expression in the Ctrl MNs, *Etv4* expression was notably reduced in the *Sertm2* KO ESC-derived MNs (Fig. 4F). Interestingly, expression of the upstream regulator *Hoxc8* remained unchanged, indicating that *Sertm2* exerts a specific role in regulating *Etv4* pool identity through GDNF signaling (Fig. 4E).”

References:

- Biscarini S, Capauto D, Peruzzi G, Lu L, Colantoni A, Santini T, Shneider NA, Caffarelli E, Laneve P, Bozzoni I (2018) Characterization of the lncRNA transcriptome in mESC-derived motor neurons: Implications for FUS-ALS. *Stem Cell Res* 27: 172-179
- Di Giorgio FP, Boulting GL, Bobrowicz S, Eggan KC (2008) Human embryonic stem cell-derived motor neurons are sensitive to the toxic effect of glial cells carrying an ALS-causing mutation. *Cell Stem Cell* 3: 637-648
- Lewandowski JP, Dumbovic G, Watson AR, Hwang T, Jacobs-Palmer E, Chang N, Much C, Turner KM, Kirby C, Rubinstein ND *et al* (2020) The Tug1 lncRNA locus is essential for male fertility. *Genome Biol* 21: 237
- Liau ES, Jin S, Chen YC, Liu WS, Calon M, Nedelec S, Nie Q, Chen JA (2023) Single-cell transcriptomic analysis reveals diversity within mammalian spinal motor neurons. *Nat Commun* 14: 46
- Liu T, Wu J, Wu Y, Hu W, Fang Z, Wang Z, Jiang C, Li S (2022) LncPep: A Resource of Translational Evidences for lncRNAs. *Front Cell Dev Biol* 10: 795084
- Mattick JS, Amaral PP, Carninci P, Carpenter S, Chang HY, Chen LL, Chen R, Dean C, Dinger ME, Fitzgerald KA *et al* (2023) Long non-coding RNAs: definitions, functions, challenges and recommendations. *Nat Rev Mol Cell Biol* 24: 430-447
- Nam JW, Choi SW, You BH (2016) Incredible RNA: Dual Functions of Coding and Noncoding. *Mol Cells* 39: 367-374
- Rayon T, Maizels RJ, Barrington C, Briscoe J (2021) Single-cell transcriptome profiling of the human developing spinal cord reveals a conserved genetic programme with human-specific features. *Development* 148
- Shi Y, Huang L, Dong H, Yang M, Ding W, Zhou X, Lu T, Liu Z, Zhou X, Wang M *et al* (2024) Decoding the spatiotemporal regulation of transcription factors during human spinal cord development. *Cell Res* 34: 193-213
- Wright BW, Yi Z, Weissman JS, Chen J (2022) The dark proteome: translation from noncanonical open reading frames. *Trends Cell Biol* 32: 243-258
- Xiao W, Halabi R, Lin CH, Nazim M, Yeom KH, Black DL (2024) The lncRNA Malat1 is trafficked to the cytoplasm as a localized mRNA encoding a small peptide in neurons. *Genes Dev* 38: 294-307

Yen YP, Hsieh WF, Tsai YY, Lu YL, Liao ES, Hsu HC, Chen YC, Liu TC, Chang M, Li J *et al* (2018) Dlk1-Dio3 locus-derived lncRNAs perpetuate postmitotic motor neuron cell fate and subtype identity. *Elife* 7

Dear Dr. Chen,

Thank you for the submission of your revised manuscript. We have now received the enclosed report from referee 3, whom I asked to please assess your full reply to all referee comments. Referee 3 still has one more suggestion that is important and that I would like you to address and incorporate before we can proceed with the official acceptance of your manuscript.

A few editorial requests will also need to be addressed:

- Our systematic image analyses of ms to be accepted identified 4 image duplications: 2 images each in Fig 1F and 4G are very similar, possibly identical, and the upper 2 images in Fig 5A and B are also very similar. Can you please explain and clarify?
- All main and all EV figures need to be uploaded as individual, high resolution figure files.
- Please reduce the number of keywords to 5.
- Please remove the author credits from the manuscript file. All credits need to be entered during online ms submission.
- The videos need to be renamed Movie EV1 and Movie EV2, and both need legends zipped together with the corresponding movie files.
- The Appendix table could be called and uploaded as EV Table 1 so that no Appendix file is needed.
- FIGURE CALLOUTS: Fig 2A is called out before Fig 1D; Fig EV5B is called out before Fig EV2A; please correct.
- The reference to BioRender should be removed from the Acknowledgments and added to the end of the Methods under a separate "Graphics" section.
- The Figure legends need to be moved to after the References.
- Figure legends:
 1. Please note that the exact p values are not provided in the legends of figures 1C, 2D, 4C, F; 5G(a), 6E, H; 7E, G, I, L.
 2. Please note that n=2 in figures 5A, B
 3. Please note that the scale bar needs to be defined for figure 7J
 4. Please note that the dotted lines are not defined in the legend of figures 1I, J. This needs to be rectified.
- I have 2 comments regarding the abstract:
 1. By generating C-terminally Flag-tagged Sertm2 and expressing it from the A730046J19Rik locus, we demonstrate that the Sertm2 micropeptide localizes in spinal MNs of mice. [Is this in vivo?]
 2. The following sentence could start with:
The GDNF signaling-induced Etv4+ motor pool is impaired in Sertm2 knockout mice ...

I would like to suggest to change the title to:

Sertm2 is a Conserved Micropeptide that promotes GDNF-mediated Motor Neuron Subtype Specification

to make the title a little more specific.

I slightly modified the short summary and blurb. Are you OK with this and is all correct:

An evolutionarily conserved micropeptide, Sertm2, encoded by the lncRNA A730046J19Rik, promotes Etv4+ motor neuron subtype specification by GDNF, proper neurite development, and motor function.

- A730046J19Rik hosts a conserved small ORF that encodes Sertm2, a 89-amino acid micropeptide.
- Sertm2 is predominantly enriched in postmitotic MNs and plays a role in GDNF-mediated regulation of Etv4 expression.
- Sertm2 shows sequence and functional conservation across mice and humans.

Referee #3:

The revision is very thorough, well-structured and very clear, a pleasure to go over the PBP.

All answers to all reviewers were more than properly addressed, except for two points that actually did not require additional experiments but just (further) explanation in the text. I feel that the authors could tackle these two points in a more straightforward manner. I'm referring to:

* Reviewer #1 point #1: "The manuscript suggests dual functions for this RNA, but no supporting data or citations establish any non-coding roles for A730046J19Rik... If there are specific data or literature supporting any non-coding roles, it should be included. Otherwise, a clearer statement is needed on why it might be more appropriate to classify this gene as a protein-coding RNA."

I'm not totally sure that the authors' response to this very good point raised by the reviewer is directly and properly addressed. And the closest to an answer I see in their response, added now to this discussion, is "it is tantalizing to probe the full spectrum of functions of the A730046J19Rik locus by deleting different regions and examining the physiological consequences." In my opinion, the authors could more directly and fearlessly address the reviewer's point in the discussion, because if their answer was that indeed we can't yet rule out that this is protein-coding gene and not a lncRNA as originally thought, that would not take away the novelty, knowledge gained and robustness of their results. This can be easily clarified by a simple sentence in the discussion.

* Reviewer #3 first point: "is the SERTM2 micropeptide expressed in human MNs?"

The authors' attempts to fulfill the experimental request are totally valid, even if unsuccessful. However, the response added now in lines 410-414 could more directly state that it's still to be demonstrated that SERTM2 peptide is expressed in human MNs, something that technically has not yet been possible.

This comment goes along the lines the authors' response to reviewer's #1 first point, no reason to not face the limitations directly because the message, novelty and interest are still great.

In my view both points are important enough to be revised by just adding/editing a single sentence in the corresponding text sections or discussion. No further review to those changes would be necessary.

Responses to the editor and reviewers

We sincerely thank the reviewers for their appreciation of the importance of our study and for their insightful comments. Our point-by-point responses to their comments and descriptions of the new experiments and computational analyses that we have now conducted are provided in this letter. The original editor's and reviewers' comments are in **black**, followed by our responses in **blue**. We highlight changes made to the text and figures in **yellow**, whereas **cyan** indicates the paragraph line numbers in the revised text. All changes are highlighted in the revised text. We have made changes to the figure numbering to address the reviewers' comments. To avoid confusion about the numbering, we provide a conversion table here for the reviewers' and editor's reference.

A few editorial requests will also need to be addressed:

- Our systematic image analyses of ms to be accepted identified 4 image duplications: 2 images each in Fig 1F and 4G are very similar, possibly identical, and the upper 2 images in Fig 5A and B are also very similar. Can you please explain and clarify?

We sincerely appreciate the editor's careful review and for bringing the duplication concerns to our attention. In response, we have thoroughly reexamined our figures and have made the following revisions to clarify the presentation while maintaining the integrity of our findings:

1. Both figures were initially intended to illustrate the expression pattern of *A730046J19Rik* in **wild-type** embryos, which led to the use of identical images. To prevent any potential confusion, we have replaced the image in the revised Figure 1F with an alternative wild-type expression image, thereby eliminating redundancy.
2. The original Figure 5B was an enlarged version of Figure 5A, intended to highlight motor nerve erosions in different regions (e.g., *cutaneous maximus*). However, we recognize that this could cause confusion for readers. To improve clarity, we have now consolidated these figures by presenting a single image in Figure 5A. Additionally, we have adjusted the margins and added parentheses in Figure 5A' and 5A" to clearly indicate the quantification areas. All quantification results are now presented separately in the revised Figure 5B.

While we have adjusted the figure layouts to enhance clarity, we emphasize that all original conclusions remain unchanged. We sincerely appreciate the editor's valuable feedback in helping us refine the presentation of our data.

- All main and all EV figures need to be uploaded as individual, high resolution figure files.

Thank you for the reminder; we have now uploaded the high-resolution files individually.

- Please reduce the number of keywords to 5.

The number of keywords has been reduced to five: motor neurons, long noncoding RNA, micropeptide, Etv4, and GDNF.

- Please remove the author credits from the manuscript file. All credits need to be entered during online ms submission.

The Author Contributions section has been removed from the manuscript.

- The videos need to be renamed Movie EV1 and Movie EV2, and both need legends zipped together with the corresponding movie files.

The movies and their corresponding legends have been modified according to the instructions.

- The Appendix table could be called and uploaded as EV Table 1 so that no Appendix file is needed.

The original appendix table has been renamed Table EV1, following the nomenclature in the Author Guidelines, and the manuscript has been updated accordingly.

- FIGURE CALLOUTS: Fig 2A is called out before Fig 1D; Fig EV5B is called out before Fig EV2A; please correct.

To resolve this issue, we added a new Figure EV1A to prevent the callout of Figure 2A before Figure 1D, while the original Figure EV5B has been relocated to the new Figure EV1E. Additionally, we identified a similar issue with Figures EV5A and EV3H and have rearranged them as new Figures EV1F and EV5A, respectively. These adjustments correct the premature callout issue, and the main text and figure legends have been updated accordingly for consistency.

- The reference to BioRender should be removed from the Acknowledgments and added to the end of the Methods under a separate "Graphics" section.

The Graphics section has been added to the revised manuscript and can be found on Lines 805–807.

- The Figure legends need to be moved to after the References.

The Figure Legends section has been added after the References section.

- Figure legends:

1. Please note that the exact p values are not provided in the legends of figures 1C, 2D, 4C, F; 5G(a), 6E, H; 7E, G, I, L.

The exact P-values for the aforementioned comparisons have been updated in the figure legends.

2. Please note that n=2 in figures 5A, B

We conducted the experiments and added an N number of 3 in the last revision, updating the quantification results for the original Figures 5A and 5B. However, due to an error, the N number was recorded incorrectly. In this revised manuscript, the figure legend has been corrected on Line 1211.

3. Please note that the scale bar needs to be defined for figure 7J

Thank you for the reminder; the scale bar for Figure 7J has now been included. (Line 1317-1318)

4. Please note that the dotted lines are not defined in the legend of figures 1I, J. This needs to be rectified.

The description of the dotted lines has been added to the figure legend accordingly. (Line 1101)

- I have 2 comments regarding the abstract:

1. By generating C-terminally Flag-tagged Sertm2 and expressing it from the A730046J19Rik locus, we demonstrate that the Sertm2 micropeptide localizes in spinal MNs of mice. [Is this in vivo?]

Yes, we employed a precise tagging system to investigate and confirm the *in vivo* distribution of the Sertm2 micropeptide in the mouse spinal cord, demonstrating its localization within spinal motor neurons. Thus, the original sentence is correct.

2. The following sentence could start with:

The GDNF signaling-induced Etv4+ motor pool is impaired in Sertm2 knockout mice ...

I would like to suggest to change the title to:

Sertm2 is a Conserved Micropeptide that promotes GDNF-mediated Motor Neuron Subtype Specification

to make the title a little more specific.

Thank you for the suggestions; the abstract has been modified accordingly. (Line 24)

I slightly modified the short summary and blurb. Are you OK with this and is all correct:

An evolutionarily conserved micropeptide, Sertm2, encoded by the lncRNA A730046J19Rik, promotes Etv4+ motor neuron subtype specification by GDNF, proper neurite development, and motor function.

- A730046J19Rik hosts a conserved small ORF that encodes Sertm2, a 89-amino acid micropeptide.
- Sertm2 is predominantly enriched in postmitotic MNs and plays a role in GDNF-mediated regulation of Etv4 expression.
- Sertm2 shows sequence and functional conservation across mice and humans.

We are grateful for the editor's suggestions, and the short summary and blurb have been revised accordingly.

EMBO reports

Referee #3:

The revision is very thorough, well-structured and very clear, a pleasure to go over the PBP.

All answers to all reviewers were more than properly addressed, except for two points that actually did not require additional experiments but just (further) explanation in the text. I feel that the authors could tackle these two points in a more straightforward manner. I'm referring to:

* Reviewer #1 point #1: "The manuscript suggests dual functions for this RNA, but no supporting data or citations establish any non-coding roles for A730046J19Rik... If there are specific data or literature supporting any non-coding roles, it should be included. Otherwise, a clearer statement is needed on why it might be more appropriate to classify this gene as a protein-coding RNA."

I'm not totally sure that the authors' response to this very good point raised by the reviewer is directly and properly addressed. And the closest to an answer I see in their response, added now to this discussion, is "it is tantalizing to probe the full spectrum of functions of the A730046J19Rik locus by deleting different regions and examining the physiological consequences." In my opinion, the authors could more directly and fearlessly address the reviewer's point in the discussion, because if their answer was that indeed we can't yet rule out

that this is protein-coding gene and not a lncRNA as originally thought, that would not take away the novelty, knowledge gained and robustness of their results. This can be easily clarified by a simple sentence in the discussion.

We appreciate the reviewer's encouraging suggestions. We understand that the reviewer supports our characterization of *A730046J19Rik* as a micropeptide while downplaying its originally misannotated function as a lncRNA. This interpretation aligns with the conclusion we presented. However, we believe it is important to maintain an objective discussion regarding its potential lncRNA function, as our rescue experiments focused solely on GDNF/Etv4 and did not explore other aspects. In light of this, we have strengthened the tone of our conclusion while leaving room for open discussion, as outlined below: (Line 413-428)

“While our study demonstrates that *Sertm2* plays a crucial role in the specification of Etv4⁺ MN pools, two intriguing questions remain. First, while our findings demonstrate the bona fide protein-coding potential of *Sertm2* as a conserved micropeptide, we cannot entirely rule out the possibility that *Sertm2* may also have additional regulatory functions within motor neurons. Interestingly, given that the host lncRNA, *A730046J19Rik*, is also significantly expressed in MN nuclei, it is plausible that *A730046J19Rik* could possess both RNA- and protein-mediated functions (Mattick *et al.*, 2023; Wright *et al.*, 2022), which remain to be fully explored. Additionally, a previous study showed that the *Tug1* locus exerts three distinct regulatory functions: acting in *cis* to regulate neighboring genes; in *trans* to modulate distant gene expression; and as a peptide to influence mitochondrial membrane potential, all contributing to its essential role in male fertility (Lewandowski *et al.*, 2020). It is tantalizing to probe the full spectrum of functions of the *A730046J19Rik* locus by deleting specific regions and examining the resulting physiological consequences will provide critical insights. Future studies employing more systematic approaches are necessary to investigate these potential multifaceted roles and further elucidate the broader regulatory functions of *Sertm2* and *A730046J19Rik* in MNs.”

* Reviewer #3 first point: "is the SERTM2 micropeptide expressed in human MNs?" The authors' attempts to fulfill the experimental request are totally valid, even if unsuccessful. However, the response added now in lines 410-414 could more directly state that it's still to be demonstrated that SERTM2 peptide is expressed in human MNs, something that technically has not yet been possible.

This comment goes along the lines the authors' response to reviewer's #1 first point, no reason to not face the limitations directly because the message, novelty and interest are still great.

In my view both points are important enough to be revised by just adding/editing a single sentence in the corresponding text sections or discussion. No further review to those changes would be necessary.

(Line 433-439)

We thank the reviewer for his/her encouragement and feedback. In the revised manuscript, we have revised the discussion section to address the point more directly and enhance our statements.

“Our findings highlight the conserved function of human SERTM2 in iPSC-MNs, providing valuable insights into its evolutionary and functional conservation. However, the expression of the SERTM2 micropeptide in specific human MN subtypes *in vivo* has yet to be confirmed. Recent advancements in spinal cord organoid technology (Gribaudo *et al*, 2024), which may better capture MN diversity in a human context, could help address this question and facilitate functional studies of SERTM2 in human organoids.”

Dr. Jun-An Chen
Academia Sinica
Institute of Molecular Biology
NangKang
Taipei, Taipei 11529
Taiwan

Dear Jun-An,

I am very pleased to accept your manuscript for publication in the next available issue of EMBO reports. Thank you for your contribution to our journal.
